# SymmCetry breaking meets multisite modification

**Vaidhiswaran Ramesh, J Krishnan***

Department of Chemical Engineering, Centre for Process Systems Engineering, Imperial College London, London, United Kingdom

**Abstract** Multisite modification is a basic way of conferring functionality to proteins and a key component of post-translational modification networks. Additional interest in multisite modification stems from its capability of acting as complex information processors. In this paper, we connect two seemingly disparate themes: symmetry and multisite modification. We examine different classes of random modification networks of substrates involving separate or common enzymes. We demonstrate that under different instances of symmetry of the modification network (invoked explicitly or implicitly and discussed in the literature), the biochemistry of multisite modification can lead to the symmetry being broken. This is shown computationally and consolidated analytically, revealing parameter regions where this can (and in fact does) happen, and characteristics of the symmetry-broken state. We discuss the relevance of these results in situations where exact symmetry is not present. Overall, through our study we show how symmetry breaking (i) can confer new capabilities to protein networks, including concentration robustness of different combinations of species (in conjunction with multiple steady states); (ii) could have been the basis for ordering of multisite modification, which is widely observed in cells; (iii) can significantly impact information processing in multisite modification and in cell signalling networks/pathways where multisite modification is present; and (iv) can be a fruitful new angle for engineering in synthetic biology and chemistry. All in all, the emerging conceptual synthesis provides a new vantage point for the elucidation and the engineering of molecular systems at the junction of chemical and biological systems.

**\*For correspondence:**
j.krishnan@imperial.ac.uk

**Competing interest:** The authors declare that no competing interests exist.

## Introduction

Reversible phosphorylation, and post-translational modification (PTM) of proteins in general, constitutes a basic way of conferring functionality to proteins in cells. This basic unit (covalent modification) is built upon in many different ways to result in the complex biochemical pathways encountered in cells.

A particular elaboration of this mechanism, which is widely encountered, is the reversible multisite modification of proteins by enzymes. Here, a number of basic variations are possible depending on whether the enzymes involved in distinct modification steps are different or if a common enzyme effects multiple modifications. In the latter case, there are variations depending on whether the enzymatic mechanism is distributive (enzyme dissociating from substrate after every modification) or processive (enzyme remains attached). Finally, there are variations depending on whether a specific ordering of modifications occurs (ordered mechanism) or not (random mechanism).

In addition to being a widely encountered way in which substrates are reversibly modified to confer functionality (and consequently of broad interest), interest in multisite modification stems from the fact that the basic modification mechanisms are capable of acting as complex molecular information processors (*Conradi and Shiu, 2018*). Various studies have highlighted the possibilities of these mechanisms exhibiting switching and threshold behaviour (*Markevich et al., 2004*), bistability/multistability (*Thomson and Gunawardena, 2009*; *Conradi and Mincheva, 2014*), oscillations (*Rubinstein et al., 2016*; *Suwanmajo and Krishnan, 2015*; *Suwanmajo et al., 2020*), biphasic dose–response

curves (*Suwanmajo and Krishnan, 2013*), and other complex behaviour (*Suwanmajo and Krishnan, 2018*). A range of studies have delineated the ingredients required (from the above possible variations) to enable or prevent such behaviour (*Conradi et al., 2017*; *Eithun and Shiu, 2017*; *Tung, 2018*). We emphasize that this rich repertoire of behaviour emerges from the most basic considerations and aspects of enzymatic modification of substrates, and that this behaviour is a feature and a consequence of the modification network (rather than a single modification). Information processing capabilities are also at the heart of different strands of work in synthetic biology engineering multisite modification, and reaction networks more broadly (*Valk et al., 2014*; *Lyons et al., 2013*; *OShaughnessy et al., 2011*; *Maguire and Huck, 2019*). This paper focuses on a distinct aspect of information processing of multisite modification: symmetry and symmetry breaking.

Symmetry and symmetry breaking are themes encountered across different scales and levels in biology, ranging from the cell population, to the cellular, to the molecular level. A fundamental theme in developmental biology is the breaking of symmetry to generate patterns. The basic questions here centre around how an apparently homogeneous field of cells can differentiate to exhibit a basic pattern which serves as a precursor for subsequent development. Modelling, experiments, and concepts from self-organization have been used to probe this generation of form, which breaks spatial symmetry. The underlying mechanisms invoked involve many variations on the classical Turing mechanism or the interplay of mechanics and chemistry (*Green and Sharpe, 2015*; *Maini et al., 2012*). This can be significantly complicated by the presence of many layers of regulation. Strong experimental evidence for such mechanisms present at the core of developmental regulation has been demonstrated in multiple model systems (*Onimaru et al., 2016*). Symmetry breaking as a basis of generating form at the cellular level, for instance, polarization and polarized or other strongly inhomogeneous patterns of concentration of species, has been explored in a range of contexts. Examples include polarity generation in fungi and plant cells (*Khan et al., 2015*), and in neutrophil chemotaxis (*Wang, 2009*). Symmetry has also been invoked as a key ingredient in the development of the MWC model which has been used to explain allostery in biomolecular information processing (*Changeux, 2012*).

While the theme of symmetry in chemistry is well recognized especially at the molecular structure level (*Hargittai and Hargittai, 1994*), there are relatively few studies of symmetry breaking at the molecular reaction level. In chemical reaction systems, symmetry is encountered in the context of chirality in racemic mixtures. Racemic mixtures comprise equal amounts of the two enantiomeric forms of a chiral molecule with opposite chiralities, and a central question is how a dominant orientation (chirality) of the molecular mixture can emerge from this. Some studies explain this as an emergent behaviour of the reaction network system governing the two forms of the molecules: even if the network/reaction system is symmetric allowing for equal amounts of the two forms, this symmetry can break, giving rise to a dominant form. A recent study (*Hochberg et al., 2017*; *Ribó et al., 2017*) evaluated and demonstrated the feasibility of such symmetry breaking in a number of potential reaction systems. Chiral symmetry breaking has been experimentally observed in crystallization of nanoparticles (*Hananel et al., 2019*), fibril formation from racemic mixtures (*Kushida et al., 2017*), and in the Soai reaction (*Soai et al., 1995*). Such symmetry breaking is of particular importance in prebiotic evolution and biology, where biopolymers and biomolecules are characterized by a specific chirality and orientation, even though the original non-life chemical world was chirally symmetric (*Chen and Ma, 2020*). The establishment of such chirality has been postulated to be important in understanding the origins of life (*Blackmond, 2020*).

This paper focuses on a specific aspect of symmetry breaking at the junction of the biological and the chemical: the breaking of symmetry in basic multisite phosphorylation (MSP) systems.

The motivation for studying symmetry and symmetry breaking in the context of multisite modification stems from different sources: conceptual insights, relevance to systems biology, and potential application in synthetic biology. In this regard, we note that (i) many of the basic modification networks accommodate different types of symmetries, as we discuss below. (ii) Certain symmetries, for example, resulting in equal concentrations of different partial phosphoforms of a given level (Case 2 symmetry, discussed below) are not only plausible in vivo, but have also been assumed in multiple contexts sometimes implicitly. (iii) An asymmetric state currently observed may have its genesis traced back to a symmetric state in evolution, which broke symmetry. (iv) Other symmetries (Case 3 symmetry, discussed below) have been found to be particularly desirable in enabling oscillatory behaviour: in fact, a thorough parametric analysis of oscillatory behaviour in certain random double-site modification networks

reveals clusters in parameter space centred around parameter sets representing networks with this symmetry (*Jolley et al., 2012*). Case 3 symmetry involves a combination of two symmetries (Case 1 and Case 2) which we individually study as well. (v) Symmetry breaking can confer new functionality and information processing characteristics enriching the repertoire of MSP. (vi) Our study of symmetric systems allows us to draw important insights about multisite modification even when exact symmetry does not hold good. Thus it is also relevant to networks which are approximately symmetric. (vii) In this sense, the symmetric scenarios also serve as valuable (and sometimes non-obvious) vantage points from which to investigate important aspects of multisite modification. Furthermore, while studying modification networks of larger numbers of modifications, the symmetric networks may represent one of the few tractable vantage points from which to study and elucidate the behaviour of such networks. (viii) These serve as interesting candidates for engineering multisite modification in synthetic biology with desirable features.

We examine basic models of MSP and evaluate the possibility of spontaneous symmetry breaking in basic and canonical reaction pathways/circuits/networks of MSP. We discuss the consequences of the results which emerge for multisite modification networks which may not possess an exact symmetry. We then discuss the various consequences of such behaviour for biological systems, and cellular signalling pathways and networks which contain multisite modification. The ordering of multisite modification is a fundamental aspect of substrate modification and its regulation, and the deployment of modified substrates in various processes. It has been the focus of different studies (*Kocieniewski et al., 2012*; *Lyons et al., 2013*; *Lössl et al., 2016*; *Valk et al., 2014*) spanning canonical pathways, important cellular processes, basic principles, and engineering for synthetic biology. We show how symmetry breaking could provide a natural mechanism for the creation of ordered multisite modification systems from random multisite modification, which could in turn explain the various degrees of ordering encountered in cells.

## Results

We begin by discussing the basic aspects of the models we employ and the way they are analysed before proceeding to the results. We discuss the multisite modification networks we study and the possible symmetries they may exhibit (with further details in Appendix 1).

### Models of multisite modification

Our primary focus is on random mechanisms of multisite modification, and we study the case of double-site modification as a tractable, representative case. *Figure 1A* represents random mechanisms of modification (i.e. modifications can proceed in either order) and depicts cases where the kinases and phosphatases effecting individual modifications could either be the same or different. Taken together, these networks span a range of basic cases of multisite modification, including the possibility of an enzyme performing multiple modifications (seen in many biological contexts) and the possibility that this may be associated with one modification direction, but not the other (due to the fact that kinases significantly outnumber phosphatases, as seen in genome-wide studies (*Ghaemmaghami et al., 2003*)). When a common enzyme is involved in effecting multiple modifications, the modification mechanism is assumed to be distributive, unless otherwise stated. We note here that such modification circuits are encountered in multiple cellular contexts and can be viewed as building blocks of more complex multisite modification networks. Such networks have been the focus of detailed studies in contexts such as circadian oscillations (*Ode and Ueda, 2018*) (involving the common kinase common phosphatase network depicted: the substrate represents the Per proteins), with additional studies on temperature compensation in this context (*Shinohara et al., 2017*; *Hatakeyama and Kaneko, 2012*). They have also been used to evaluate design principles for both oscillatory and pattern forming behaviour more broadly (*Jolley et al., 2012*; *Sugai et al., 2017*). For purposes of contrast and elucidating basic effects, we also examine two related modification networks: (i) an ordered double-site modification mechanism (*Figure 1C*) mediated by a common kinase and common phosphatase. The specific ordering of modification involves the phosphorylation order being opposite to that of dephosphorylation resulting in one partial phosphoform. This has been extensively studied in the literature (e.g. (*Thomson and Gunawardena, 2009*; *Conradi and*

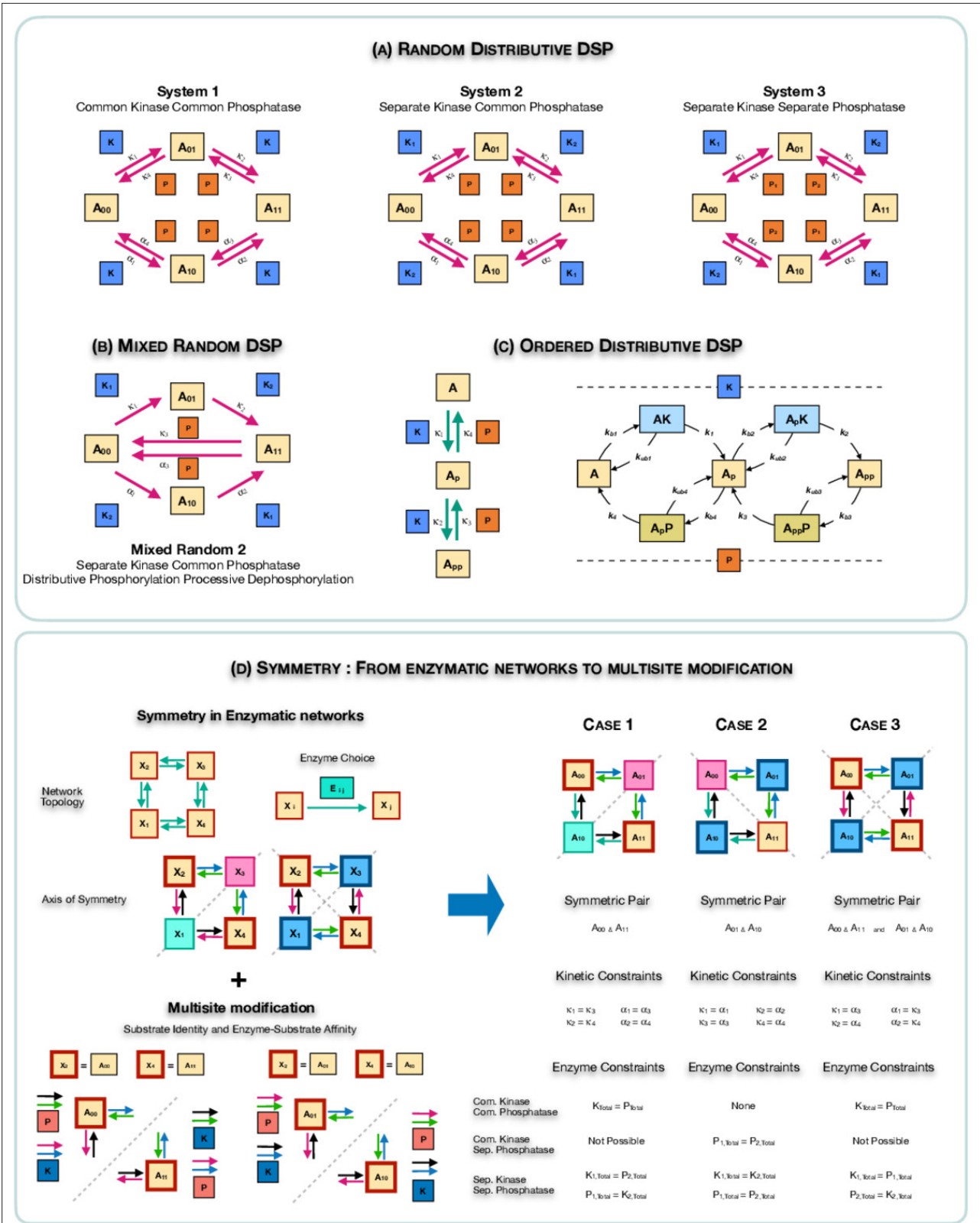

**Figure 1.** (A–C) Schematic representation of the various multisite phosphorylation networks considered in the paper. (A) depicts the core networks while (B,C) serve as suitable contrasts to illustrate basic points. The labels $\kappa_i$, $\alpha_i$ in the schematic represent the triplet of binding, unbinding, and catalytic rate constants involved in the enzyme modification for the $i^{th}$ modification (on each leg of the network). Detailed model description is provided in *Appendix 2—figure 10*. (A) shows the various random (distributive) double-site phosphorylation (DSP) networks (the focal point of the study) with

*Figure 1 continued on next page*

*Figure 1 continued*

different combinations of enzyme action (common kinase and common phosphatase, separate kinase and common phosphatase, and separate kinase and separate phosphatase). (**B**) shows the random DSP with distributive phosphorylation and processive dephosphorylation (depicted for simplicity as direct arrows from $A_{11} \rightarrow A_{00}$) with separate kinase and common phosphatase (model: mixed random 2). (**C**) shows the ordered distributive DSP with common kinase and common phosphatase, and an expanded description of reaction mechanism showing in detail the binding, unbinding, and catalytic action for each modification step. (**D**). Schematic representation of the three different classes of symmetries in the random DSP networks considered in this study. The different symmetries are depicted in the right-hand panel. In each case, the axis of symmetry is depicted, and nodes of the network on either side of this axis (enclosed in a boundary of the same colour) have equal concentration. Identically coloured arrows in schematics are indicative of equal kinetic rate constants (for the corresponding triplet of binding/unbinding/catalysis reaction) and equal total concentrations of enzymes involved. The associated kinetic and enzymatic requirements required for enforcing symmetries are also listed. These are key ingredient in establishing symmetry in the reaction network. The origins of these symmetries can be conceptualized and visualized by examining an enzymatic network with a 'square' topology (left-hand panel), where every reaction is mediated by an enzyme. Such networks can have symmetry about one axis or two axes. Now examining the single-axis reflection symmetry in multisite modification results in two possibilities of symmetric nodes (the pair $(A_{00}, A_{11})$ and $(A_{01}, A_{10})$). In each case, symmetry requires different pairs of reactions (depicted by identically coloured arrows) to have equivalent rates and enzyme amounts. Importantly, in the $A_{11} = A_{00}$ case these pairs of reactions are associated with enzymes of different kinds, while in the $A_{01} = A_{10}$ case they are associated with enzymes of the same kind. While this is depicted for a single kinase and a single phosphatase, it applies to any combination of common/separate kinase and phosphatase. This dichotomy underscores the difference between case 1 and case 2 symmetry. Overall this conceptualization allows us to obtain the three symmetries along with the kinetic and enzymatic requirements shown in the right-hand panel.

*Shiu, 2018*). (ii) A random mechanism where the dephosphorylation is processive (*Figure 1B* and *Appendix 2—figure Appendix 2—figure 1A*).

## From networks to models

The model for each of these networks (depicted in *Figure 1A–C*, *Appendix 2—figure Appendix 2—figure 1A*) is built up from widely used descriptions of individual substrate modification by an enzyme (involving reversible complex formation and irreversible modification to give the product; see *Appendix 2—figure 10*). Such a description makes no a priori assumptions about the kinetic regimes of modifications. Further details are presented in Appendix 1. Throughout the paper, we work with these canonical modification circuits, where the substrate forms are denoted by $A$, with subscripts denoting the type of modification. Depending on the context, these could represent different proteins.

## Associated network symmetries

In order to understand and visualize the types of symmetries we will examine, it is fruitful to examine a 'square' reaction network, which has the same network topology as that of the multisite modification networks above. Note that in this depiction the nodes of the network represent substrates, while the enzymes are implicitly present in the arrows: both substrates and enzymes together constitute an enzymatic reaction network of this type. As depicted in *Figure 1D* (left panel), there are two types of symmetries which can be encountered. (i) In the first case, the two 'legs' of the network are symmetric (about the axis of symmetry depicted), which means that the rates of reaction for corresponding reactions on either side of the axis are the same. The associated pair of species (nodes of this network representing substrates, on either side of the axis of symmetry) is expected to behave identically (assuming the same initial conditions). (ii) In the second case, the symmetry is associated with two pairs of species simultaneously and can be viewed as a simultaneous occurrence of two of the previous symmetries, along different axes (see *Figure 1D*). Viewed from a general network perspective, in each case the symmetry is a consequence of rates of different pairs of reactions (intrinsic reaction rate constants as well as total enzyme concentrations) being identical, thus giving rise to the symmetry. Thus enabling such symmetries establishes correspondences/constraints between different pairs of enzymes. Note that in this network we have not made any restrictions on which enzymes may be involved in specific steps. Establishing the structural requirements for symmetry then allows us to examine when and how the multisite modification networks we study exhibit different symmetries.

## Network symmetry meets multisite phosphorylation

We now focus on the network symmetries in the specific instance of the biochemistry of multisite modification. In so doing, we discuss different types of symmetries which multisite modification networks can exhibit. While some of these symmetries may appear to be more natural biologically,

it is useful to examine all these together to obtain a comprehensive systems understanding. Furthermore, some of these have been postulated explicitly or invoked implicitly in multiple different contexts.

Symmetries in such networks require basic conditions/constraints on the kinetics and enzyme amounts (refer *Figure 1D*, right panel). In particular, equivalence between two reactions (as represented in the schematic) requires that the rate constants of their constituent elementary reactions (binding, unbinding, and catalytic) remain equal. The first two cases of symmetries correspond to a scenario where the two 'legs' of the network are symmetric. The difference between them is what the symmetric nodes of the network correspond to in the context of multisite modification along with the fundamentally distinct pairings of enzymes in each case (discussed further below).

### Case 1 symmetry: $[A_{00}] = [A_{11}]$

In this case, the nodes involved in either leg of the symmetric network are $A_{00}$ and $A_{11}$. In such a case, the requirement of a symmetry implies that for these two phosphoforms the action of an enzyme (kinase) on one of these substrates ($A_{00}$) has the same rate as that of another enzyme (phosphatase) on the other substrate ($A_{11}$) (this is seen by the corresponding reaction arrows on the two legs of the network). Furthermore, an analogous requirement applies to the production of each of these species from the partial phosphoforms. With these requirements a symmetry between $A_{00}$ and $A_{11}$ is maintained. We further note that such a requirement (of having certain rates of kinase-mediated reactions being equal to that of other phosphate-based reactions) places a constraint on total enzyme amounts as well. Case 1 symmetry is of interest both as a basic independent symmetry and as a constituent of Case 3 symmetry discussed below.

### Accommodating the requirements for symmetry

(i) The above requirements can be accommodated both in the common kinase common phosphatase case and the separate kinase separate phosphatase case, but not in the separate kinase common phosphatase case (discussed in Appendix 1). (ii) We also note that a simpler network, which corresponds to ordered double-site modification, also accommodates a symmetry of this type (while only possessing a single partial phosphoform). Here too, this is accommodated in the common kinase common phosphatase and separate kinase separate phosphatase cases.

### Case 2 symmetry: $[A_{01}] = [A_{10}]$

In this case, the nodes involved in either leg of the symmetric network are $A_{01}$ and $A_{10}$. Such a symmetry is realized if the following pairs of reactions have the same rates: (i) phosphorylation of $A_{00}$ to produce the respective partial phosphoforms, (ii) dephosphorylation of $A_{11}$ to produce the respective partial phosphoforms, (iii) the phosphorylation of the respective partial phosphoforms, and (iv) the dephosphorylation of the two partial phosphoforms. Note that equal rates of reaction require the same intrinsic kinetic rate constants (for binding, unbinding, and catalysis of substrate by enzymes) as well as total enzyme amounts. This is characterized by saying that the rate of modification of all substrates of a given level of modification is the same (and likewise for demodification), and this ensures that progression in substrate modification is equally balanced between the pathways associated with each partial phosphoform (a feature explicitly/implicitly assumed in multiple instances in the literature). This symmetry can be accommodated in all cases of separate/common kinases and separate/common phosphatases.

### Difference between Case 1 and Case 2 symmetries

Case 1 and Case 2 symmetries involve different pairs of symmetric nodes. As noted earlier, the symmetries require both intrinsic rate constants and enzyme amounts to be equal for different pairs of enzymes. The essential difference between the two cases is the essentially different enzyme pairs associated with this. In Case 1 symmetry, the pairing is between enzymes of different types (a kinase and a phosphatase), while in Case 2 it is between enzymes of the same type (between kinases and between phosphatases). This is exactly why Case 1 symmetry is not possible in the separate kinase common phosphatase network, while Case 2 symmetry is.

## Case 3 symmetry: $[A_{00}] = [A_{11}]$ and $[A_{01}] = [A_{10}]$ simultaneously

This involves the combination of the earlier cases. Here the action of a kinase enzyme on a substrate occurs at the same rate as that of the phosphatase enzyme on its associated substrate in the diagonally opposite modification leg. Thus the action of the kinase on $A_{00}$ modifying it to $A_{01}$ is the same as that of the phosphatase action on $A_{11}$ modifying it to $A_{10}$. This applies to the modification of all substrates in the network. Such symmetries have been implicated in oscillatory networks of multisite modification underlying circadian oscillators (*Jolley et al., 2012*). Again, similar to Case 1 symmetry, the separate kinase common phosphatase case cannot accommodate this symmetry.

## The basic questions

The basic questions we address below are (i) Are these symmetries always maintained or can they be broken? (ii) What network features determine whether or not symmetry breaking is possible? (iii) What kind of capabilities does symmetry breaking contribute? (iv) When symmetry breaking is possible, can the parameter regimes for symmetry breaking be established?

## Methods of analysis

To address the above questions, we employ two approaches in tandem: (i) computational analysis, involving simulations and bifurcation analysis, where we demonstrate the possibility of such behaviour occurring. We note here that our bifurcation parameter is the total substrate concentration, though it could apply to other parameters; and (ii) analytical work which rules out the possibility of symmetry breaking in networks irrespective of kinetic parameters, bringing to the fore structural features which prevent the occurrence of the behaviour. Analytical work is also used to demonstrate necessary conditions for symmetry breaking (in terms of kinetic parameters and total enzyme and substrate amounts) and further that in these cases these conditions are sufficient to guarantee the presence of symmetry breaking. Additionally, analytical work also reveals important characteristics of symmetry-broken states.

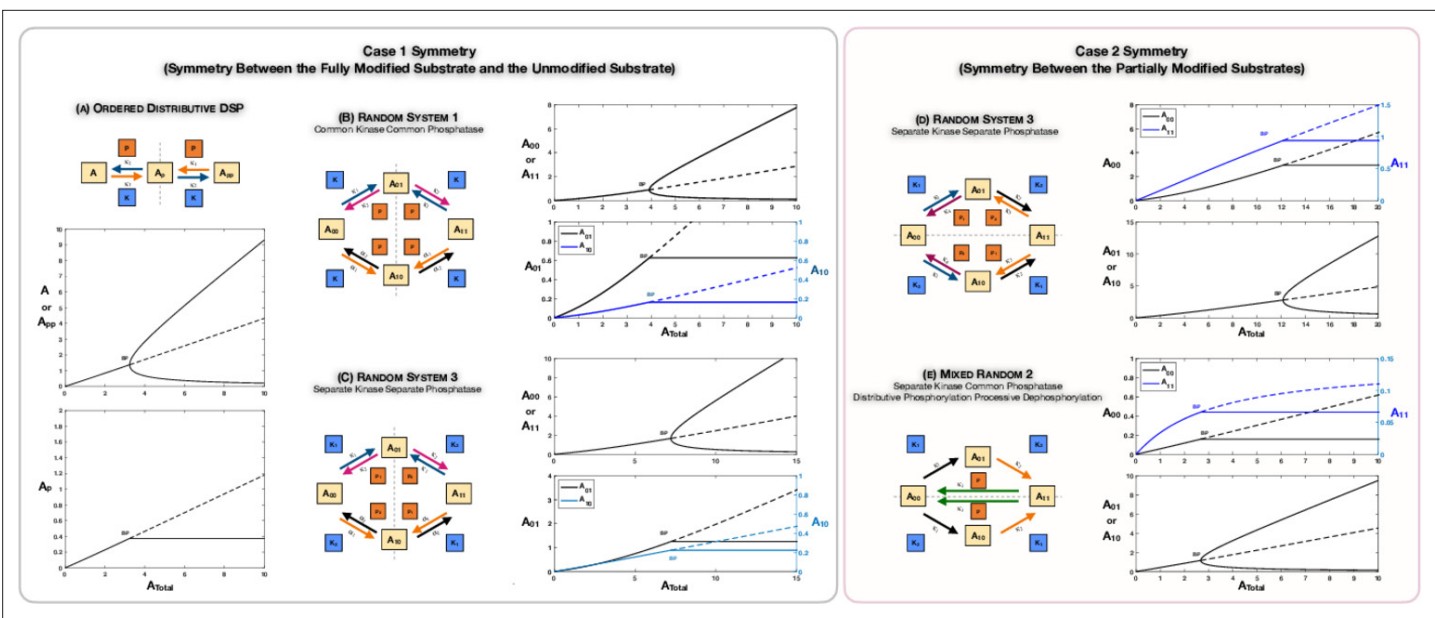

**Figure 2.** Case 1 and Case 2 symmetry breaking in various double-site phosphorylation (DSP) networks. (**A–C**). Case 1 symmetry breaking in distributive DSP: (**A**) ordered DSP with common kinase and common phosphatase, (**B**), random DSP with common kinase and common phosphatase, and (**C**) random DSP with separate kinase and separate phosphatase. Note that in these cases the concentrations of the partially modified substrates are fixed after symmetry breaking (in the symmetry-broken state) in the bifurcation diagrams. (**D, E**). Case 2 symmetry breaking: (**D**) random DSP with separate kinase and separate phosphatase and (**E**) mixed random DSP with distributive phosphorylation through separate kinases and processive dephosphorylation through common phosphatase. Note that in these cases the concentrations of the fully modified and unmodified substrate are fixed after symmetry breaking (in the symmetry-broken state) in the bifurcation diagrams. Dashed lines indicate unstable steady states, while solid lines represent stable steady states in the bifurcation diagram. Dashed lines in the schematic represent axis of symmetry of the network. BP: pitchfork bifurcation.

We present the results for Case 1, Case 2, and Case 3 for the different random modification networks below. The ordered double-site modification network can exhibit Case 1 symmetry, as noted above. Therefore, in presenting Case 1 symmetry, we start with this simpler network, before proceeding to the random modification networks.

## Analysis of a simpler ordered mechanism reveals the origins of Case 1 symmetry breaking

We first analyse the scenario of Case 1 Symmetry. It is instructive to examine an ordered mechanism (*Figure 2A*) in this regard as it is simpler while exhibiting the same behaviour encountered in random mechanisms. The system has a symmetric steady state which is characterized by (i) equal concentrations of unmodified ($A$) and fully modified ($A_{pp}$) phosphoforms and (ii) equal concentrations of free kinase and phosphatase (note that the total concentrations of these enzymes need to be the same for symmetry to be present). This steady state simply represents an absence of bias in the direction of modification (i.e. between the unmodified and fully modified phosphoforms). However as the total substrate concentration is varied, we find that this steady state loses stability via a supercritical pitchfork bifurcation (*Strogatz, 2001*). A pair of asymmetric steady states emerge which are stable. These correspond to either $[A_{pp}] > [A]$ or the other way around, and unequal free enzyme concentrations. Interestingly on each of these steady-state branches, the value of the intermediate phosphoform $A_{pp}$ remains fixed at the level at the bifurcation point (*Figure 2A*, lower panel). The presence of asymmetric states, as well as the fact that the partial phosphoform concentration is fixed on the branches of asymmetric steady states, is established analytically (see Appendix 1, *Source code 1* [Section 2.1], and *Supplementary file 1*).

### Conditions for symmetry breaking

The asymmetric steady states represent the establishment of overall directionality in the reaction network output, even in the absence of any a priori bias (in terms of reaction rates and enzyme concentrations). Analytical work also reveals the necessary conditions for symmetry breaking to occur in this system: $k_{21}$. In other words, the catalytic rate constant for the second phosphorylation is greater than that of the first phosphorylation step. Note that this does not involve binding or unbinding constants for enzyme/substrate interactions. Further analysis indicates that in such a parameter regime a symmetry-broken state is guaranteed to exist for some value of the bifurcation parameter $A_{Total}$ (see Appendix 1, *Source code 1* [Section 2.1], and *Supplementary file 1*). It is worth emphasizing here that (i) the nonlinearity responsible for the symmetry breaking arises purely from sequestration effects with no explicit feedback present and (ii) an analogous case of single-site modification is incapable of intrinsically exhibiting such behaviour.

### The random mechanism with common kinase and phosphatase

The symmetry breaking observed in this ordered mechanism is seen in the random modification with common kinase and phosphatase (*Figure 2B*). The random modification network can be thought of as two connected 'legs' of ordered modification networks, and consequently echoes of the behaviour seen previously are observed here. In this instance, beyond the bifurcation point, the concentrations of both partial phosphoforms remain fixed at their values at the bifurcation point. This is established analytically (see Appendix 1, *Source code 1* [Section 3.1], and *Supplementary file 1*).

### Conditions for symmetry breaking

An analytical necessary criterion for the presence of symmetry breaking in this system is presented in Appendix 1, *Source code 1* (Section 3.1), and *Supplementary file 1*. It takes the form $(k_2 - k_1) + \alpha(k_2/a_2)(a_2 - a_1) > 0$. Here $k_1, k_2$ are the catalytic rate constants associated with phosphorylation along one leg of modifications ($A_{00} \to A_{01} \to A_{11}$), while $a_1, a_2$ are the catalytic rate constants associated with phosphorylation along the other leg of modifications ($A_{00} \to A_{10} \to A_{11}$). Further $\alpha$ is a positive constant determined in terms by rate constants (including binding and unbinding). Further work in *Source code 1* (Section 3.2) and *Supplementary file 1* establishes the fact that this is a sufficient condition for the generation of asymmetric states for some value of $A_{Total}$.

## A comparison with ordered mechanisms reveals additional flexibility available for symmetry breaking in random mechanisms

We can make multiple inferences from this condition in relation to the corresponding condition for ordered mechanisms discussed above. (i) If both $k_{21}$ and $a_{21}$, then the requisite condition is satisfied.

This means that if each leg (viewed as an ordered mechanism) satisfies the conditions for symmetry breaking in ordered mechanisms, this guarantees the possibility of symmetry breaking in the random mechanism. (ii) For the same reason, if neither leg satisfies the condition, then symmetry breaking is precluded in the random modification network. (iii) Interestingly if only one leg satisfies the criterion for symmetry breaking, it is possible for the entire random network to break symmetry (an example of this is presented in *Appendix 2—figure 3*). In such a case, the random network can be viewed as being comprising two subnetworks, only one of which is the driver of this behaviour.

## Symmetry breaking is possible even if the enzymes performing each modification are different

Random mechanisms with different kinases and phosphatases can also exhibit the same type of symmetry (this places constraints on total concentrations of 'corresponding' enzymes, in addition to the kinetic constraints already discussed). As seen in *Figure 2C*, this system also exhibits a similar symmetry-breaking behaviour as encountered above, and here again the concentration of partial phosphoforms is fixed beyond the bifurcation. The case of different kinases and phosphatases represents a much more general case (not requiring any enzyme to perform more than one modification) with significantly reduced nonlinearity (for the same reason), which is nonetheless sufficient for symmetry breaking.

## Conditions for symmetry breaking reveal requirements on both modification legs

The necessary conditions for symmetry breaking here are established analytically in *Source code 1* (Section 3.3) and *Supplementary file 1*, where it is also shown that these conditions guarantee the existence of a symmetry-broken state. This takes the form $k_2/k_{12Total}/P_{1Total}$ and $a_2/a_{11Total}/P_{2Total}$ (the equation could also be written in terms of the total concentrations of kinases). The main difference here is that (i) there are two such conditions to be satisfied and (ii) they also involve total enzyme amounts. (iii) When $P_{2Total} = P_{1Total}$, this amounts to $k_{21}$ AND $a_{21}$, which is a requirement for each of the legs of the modification network. (iv) When $P_{2Total} \neq P_{1Total}$, this amounts to a tighter requirement on one of the legs (where enzyme $P_2$ is involved in the first dephosphorylation step) and a weaker requirement on the other (where enzyme $P_2$ is involved in the second dephosphorylation step).

## Symmetry breaking cannot be observed in an ordered mechanism constituting a single leg of the modification if all modifications are effected by different enzymes

Each leg of the modification corresponds to an ordered modification mechanism with different kinases and phosphatases, which is incapable of symmetry breaking (as shown in *Source code 1* [Section 2.2] and *Supplementary file 1*) and multistability in general. Thus the observed symmetry breaking is an emergent behaviour of the entire network with both modification legs.

### Implications

The implication of Case 1 symmetry breaking is that it is possible to establish a directionality to the modification even if none existed, resulting in an establishment of relative dominance of fully modified phosphoforms vis-a-vis fully unmodified forms or the other way round. Case 1 symmetry is also a constituent of Case 3 symmetry, and this has implications in that situation as well.

### Case 2 symmetry: when can it break?

Case 2 symmetry implies that there is no bias in the ordering of modifications and this is manifest in the equal steady-state concentrations of the partial phosphoforms $A_{01}$ and $A_{10}$. Examining all the cases of random networks together (*Figure 1A–C*) reveals the following insights: (i) the case of common kinase and common phosphatase will not lead to the breaking of this symmetry. (ii) The case of different kinase and common phosphatase will also not lead to the breaking of this symmetry. In both cases, this can be shown directly analytically by demonstrating that for any steady states (irrespective of parameters) the partial phosphoforms necessarily must have equal concentrations (see *Source code 1* [Sections 3.1 and 3.2] and *Supplementary file 1*). Incidentally, an identical conclusion can be drawn

for the common kinase common phosphatase case, irrespective of the number of modification sites. (iii) On the other hand, the case of different kinase and different phosphatase can indeed exhibit the breaking of this symmetry (*Figure 2D*) via a supercritical pitchfork bifurcation: here the asymmetric steady states are characterized by fixed values of concentrations of unmodified and fully modified phosphoforms. This is established analytically (see *Source code 1* [Section 3.3] and *Supplementary file 1*).

## Conditions for symmetry breaking

Analytical results provide further insights. The necessary requirements for an asymmetric state are $k_1/k_4 > K_{Total}/K_{Total}$ and $k_2/k_3 > K_{Total}/K_{Total}$. Note here that $K_{Total}$ denotes the total concentration of each of the kinase enzymes while $P_{Total}$ denotes the total concentration of each of the phosphatase enzymes (the equality, a requirement of symmetry). $k_1, k_2$ denote the catalytic constants for phosphorylation of $A$ and the partial phosphoforms (the constants being equal), while $k_3, k_4$ denote the catalytic constants of dephosphorylation of $A_{11}$ and the partial phosphoforms. Further analysis shows that this is sufficient for the existence of an asymmetric state. From this we can infer that (i) if $k_1/k_4 > k_2/k_3$ then this behaviour is precluded. (ii) On the other hand if $k_1/k_4 < k_2/k_3$, then by suitable choices of total enzyme concentrations (quantities easy to manipulate), these conditions can be satisfied. (iii) In such a case, there is only a finite range of ratio of total enzyme concentrations (between $k_2/k_3$ and $k_1/k_4$) which can result in symmetry breaking. (iv) The presence of multiple kinases and phosphatases proves to have a combination of both sufficient nonlinearity as well as sufficient flexibility (from the multiplicity of enzymes) to enable such behaviour. As seen previously with multiple kinases/phosphatases, the parameters need to satisfy two inequalities. A combination of the naturalness of the symmetry and the widely encountered modification scenario suggests that symmetry breaking here may be encountered quite broadly.

## Case 2 symmetry breaking in a separate kinase common phosphatase network with processive dephosphorylation

We aimed to get further insights into the factors driving such behaviour by comparing it with other related networks. To do this, we examined associated random modification networks where the dephosphorylation was processive (*Figure 1B*, *Appendix 2—figure 1A*). Note that this implies that the phosphatase has to be the same for all modifications. Here we find that if the kinase is common to different phosphorylation steps, the symmetry does not break; however, surprisingly when the kinases are different, symmetry breaking does indeed happen, as shown in *Figure 2E* (again reinforcing the flexibility provided by having different enzymes in enabling such behaviour). This is supported analytically (see *Source code 1* [Section 4.2] and *Supplementary file 1*). A direct comparison with the different kinase common phosphatase random mechanism above reveals a new dimension: having processive dephosphorylation while reducing the nonlinearity in the network actually enables this behaviour which was otherwise precluded. Elsewhere we have noted how having processive dephosphorylation could readily enable other behaviour (oscillations) which was not observed when the modification was distributive (*Suwanmajo and Krishnan, 2015*). We also note that different conditions in the cell (or stimuli) could effect a transition from distributive to processive mechanisms, as demonstrated (*Aoki et al., 2013*; *Aoki et al., 2011*; *Kocieniewski et al., 2012*).

### Implications

Case 2 symmetry breaking ultimately results in the biasing of one ordered sequence of modifications over another. The implications of this as a key step for generating ordering of modifications are discussed subsequently.

## Case 3 symmetry breaking reveals the simultaneous breaking of two symmetries

The scenario of Case 3 symmetry is examined in *Figure 3*. This involves a combination of the earlier cases. *Figure 3A* (*Appendix 2—figure 2*) focuses on the common kinase common phosphatase case. Here too, the symmetric steady state (characterized by $[A_{01}] = [A_{10}]$ and $[A_{00}] = [A_{11}]$) can lose symmetry to a pair of asymmetric steady states via a supercritical pitchfork bifurcation (*Appendix 2—figure 2*). The noteworthy point here is that both symmetries necessarily break together in general. In addition to previously observed bifurcation patterns, new possibilities arise. One is the possibility of a

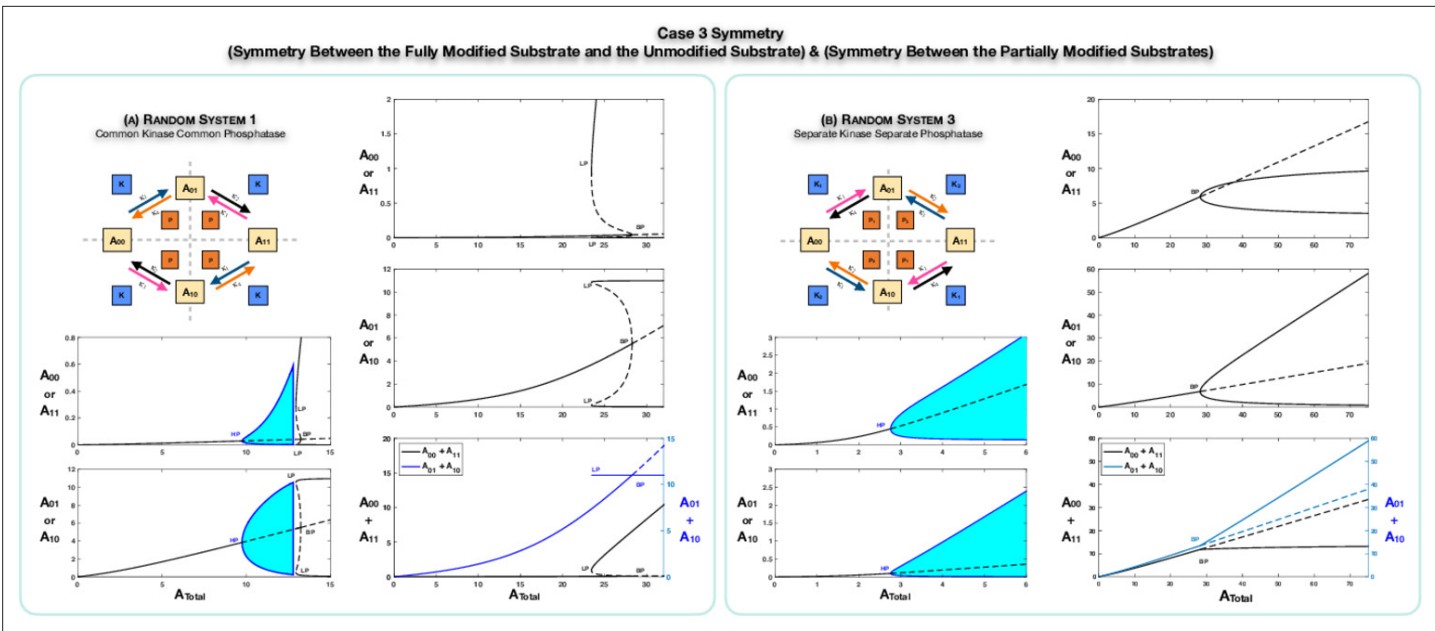

**Figure 3.** Case 3 symmetry breaking in various double-site phosphorylation (DSP) networks. (**A**) shows Case 3 symmetry breaking in random DSP with common kinase and common phosphatase. The symmetric steady state is capable of losing stability either through a Hopf bifurcation leading to oscillations, which is followed by symmetry breaking through a subcritical pitchfork bifurcation eventually leading to stable asymmetric steady states (column 1), or just by breaking symmetry leading to asymmetric branches through a subcritical pitchfork bifurcation, which eventually becomes stable (column 2). As seen in the plots, both symmetries simultaneously break. Note that the sum of the concentrations of the partially modified substrates is fixed in the asymmetric states in the bifurcation diagrams. Symmetry breaking via a supercritical pitchfork bifurcation, as seen previously in other cases, can also be seen (**Appendix 2—figure 2**). (**B**) shows Case 3 symmetry breaking in random DSP with separate kinase and separate phosphatase. The symmetric steady state is capable of losing stability either through a Hopf bifurcation leading to oscillations (column 1) or by breaking symmetry leading to two stable asymmetric branches through a supercritical pitchfork bifurcation (column 2). Note that the sum of the concentrations of the completely modified and completely unmodified substrates is approximately constant in the asymmetric steady states in the bifurcation diagram. (However, Case 3 symmetry breaking in this network is also capable of providing approximate concentration robustness in the sum of the concentrations of partial substrate forms; see main text and **Appendix 2—figure 4**.) Dotted lines indicate unstable steady states, while solid lines represent stable steady states in the bifurcation diagram. Shaded regions in the bifurcation diagram indicate regions of oscillations, and the blue lines indicate bounds on concentrations during such oscillations. BP: pitchfork bifurcation; HP: Hopf bifurcation; LP: saddle node bifurcation.

subcritical pitchfork bifurcation (**Figure 3A**). In such a case, the branches of asymmetric states are unstable at the point of inception, but following a saddle node bifurcation, they become stable. As a direct consequence of this, when the parameter (total concentration of substrate) is varied, a sudden change from a symmetric to asymmetric steady state is observed in simulations, the latter exhibiting a pronounced asymmetry, which is not a small perturbation of the symmetric state. The asymmetric steady states are characterized by high levels of $A_{11}$ and one of the partial phosphoforms, and low levels of $A_{00}$ and the other phosphoform, or the other way around. Furthermore, the grouping in which each partial phosphoform is present is determined by baseline parameters (and changing baseline parameters can alter the grouping).

Irrespective of the nature of the pitchfork bifurcation (supercritical or subcritical), we find that the sum of partial phosphoform concentrations remains fixed on the asymmetric branches, and this is established analytically (see Appendix 1, **Source code 1** [Section 3.3], and **Supplementary file 1**). The implications of this distinct type of invariant are discussed in the next section. It is interesting to contrast the invariant in this Case 3 symmetry breaking with invariants in symmetry breaking in the constituent symmetries (in this instance of common kinase common phosphatase, Case 1 symmetry, since Case 2 symmetry does not break). In Case 1 symmetry breaking, the invariants are the individual partial phosphoform concentrations while here it is the sum.

## Implications

This simultaneous breaking of symmetries and pairing of partial phosphoform and fully modified (or fully unmodified phosphoform) has a transparent interpretation. Symmetry breaking simultaneously

imposes directionality to the modification (i.e. relative dominance of fully modified vs. fully unmodified phosphoform) as well as a particular route of modification (via one of the two phosphoforms).

## The presence of oscillations

Another behaviour which is observed in a different parameter regime is the emergence of oscillations, via a Hopf bifurcation, and this precedes the subcritical pitchfork bifurcation (*Figure 3A*). The oscillations do not preserve the symmetry of the original system. Instead we see correlated changes between the corresponding pairs of substrates at different time intervals. As the total substrate concentration increases, the period of oscillations increases, as the periodic trajectory comes close to a steady state in the phase space (*Appendix 2—figure 2*). Oscillations in such networks can occur without symmetry breaking, and in fact oscillations emerging from such random modification networks have been the focus of earlier studies (*Jolley et al., 2012*), Here we show that in the presence of symmetry (a condition recognized as a desirable criterion for oscillations), oscillations may be present in conjunction with symmetry breaking, which affects the oscillatory range and characteristics of oscillations.

## Conditions for symmetry breaking

Analytical work in the case of common kinases and common phosphatases reveals a necessary condition for the realization of an asymmetric state (which is shown to be sufficient as well). This takes the form $c_3(1 - k_3/k_2) + c_1(1 - k_1/k_4) > 0$. Here $c_1$ and $c_3$ are known positive constants, which depend on kinetic parameters. As in the situation of Case 1 symmetry, this hinges on two ratios of catalytic constants, though in contrast to that case it is the phosphorylation and dephosphorylation rate constants associated with the interconversion between $A_{00}$ and $A_{01}$ ($k_1/k_4$) and dephosphorylation and phosphorylation rate constants for the interconversion between $A_{01}$ and $A_{11}$ ($k_3/k_2$). As before we can make a range of conclusions: (i) $k_3/k_2 > 1$ and $k_1/k_4 > 1$ will preclude an asymmetric state; (ii) $k_3/k_2 < 1$ and $k_1/k_4 < 1$ will ensure the possibility of an asymmetric state; and (iii) a combination of parts of the above conditions can allow for an asymmetric state depending on the constants $c_1, c_3$. In this regard, we also point out that these results show when symmetry breaking is precluded, and this combined with (*Jolley et al., 2012*) yield conditions under which oscillations can occur without interference from symmetry breaking (and in fact multistability).

## The case of different kinases and phosphatases

Here (*Figure 3B*), we again find symmetry breaking via supercritical pitchfork bifurcations (but not subcritical pitchfork bifurcations), and the symmetry-broken states (which necessarily have both symmetries broken: see *Source code 1* [Section 3.3] and *Supplementary file 1*) are characterized by having a higher level of one pair of substrates (one partial phosphoform and a completely modified/unmodified phosphoform) and a lower level for the other pair.

Here an exact invariant of the form examined previously does not hold: instead we find that (depending on parameters) either the sum of partial phosphoform concentrations or the sum of concentrations of $A_{00}$ and $A_{11}$ is approximately constant (see *Figure 3*, *Appendix 2—figure 4*, *Source code 1* [Section 3.3], and *Supplementary file 1*). It is worth noting as a contrast that fixed individual concentrations for pairs of species ($A_{01}, A_{10}$) and ($A_{00}, A_{11}$) are obtained in this case for symmetry breaking of the constituent Case 1 and Case 2 symmetries, respectively. The possibility of oscillations (stable over a broad range of parameters) emerging is also seen here (*Figure 3B*), though we have never found it occurring side-by-side with symmetry breaking (seen earlier in the common kinase common phosphatase network). The presence of oscillations expands on and complements (*Jolley et al., 2012*), by revealing oscillations in this network which is desirable as it has additional tuneable dials (multiple total enzyme amounts).

## Implications for inferences based on measurements

In Case 3 symmetry breaking in both the common kinase common phosphatase and separate kinase separate phosphatase, we find that in the symmetry-broken state the concentration of one partial phosphoform and either unmodified or fully modified form may be significantly elevated relative to its counterpart. This disparity between the two pairs can become pronounced and easily be misinterpreted as suggesting (i) the nonexistence of other active modifications or (ii) a significant disparity in intrinsic kinetic rates of modification in the two legs of the network.

## Discussion

This paper has focused on symmetry-breaking behaviour in MSP systems (summarized in *Figure 4*). The wide prevalence and relevance of MSP is well-established, but why focus on symmetry?

There are multiple reasons for this: (i) the structure of basic modification networks for MSP (the topology as well as positions of enzyme action therein) admits to the possibility of symmetries. Additionally, these have been sometimes implicitly or explicitly assumed in the literature (*Sadreev et al., 2014*; *Enciso and Ryerson, 2017*): for example, Case 2 symmetries where different partial phospho-forms behave in a similar way. In such instances, our results indicate that even if there is no biasing in the network interactions, the two phosphoforms in such a case may end up behaving very differently. Thus a simple, plausible assumption may have far-reaching and unexpected consequences. (ii) Certain symmetries may indeed naturally exist, for example, the possibility that modification/demodification can proceed at an equal rate, irrespective of the modification site under consideration (Case 2 symmetry). In other instances (Case 3 symmetry), exhaustive parametric analysis of random double-site modification networks reveals the fact that oscillatory behaviour occurs in clusters centred around these symmetric networks (i.e. parameter sets which enable Case 3 symmetry) (*Jolley et al., 2012*). Thus networks which possess these symmetries (or represent small to moderate deviations therefrom) represent those enabling oscillations, suggestive of a basic design principle. Case 3 symmetry involves Case 1 and Case 2 symmetries as basic building blocks. Our analysis in both instances indicates distinct, unexpected outcomes which can emerge in terms of system behaviour and information processing characteristics. (iii) The breaking of symmetries may have been exploited during the process of evolution: in such cases. the presence of observed asymmetric networks may have its origins in symmetric cases encountered in evolution. In particular, as discussed further below, we show how multisite modification possesses natural ingredients for creating ordering by (Case 2) symmetry breaking. (iv) The insights we obtain are also relevant to systems where the exact symmetry may not strictly hold, but which are not large deviations of the symmetric case. In the latter case, clear echoes of the behaviour we discuss may continue to be observed (*Appendix 2—figures 5 and 6*). In such cases, the symmetric scenario provides a key vantage point from which to understand the origins of the behaviour. This behaviour may manifest itself as multistability in these cases, but in contrast to multistability which may be more broadly seen in parameter space, the origins and characteristics of the steady states in these instances can be traced back to the symmetric case and symmetry breaking. This is also an example of how having clear-cut reference cases allows us to elucidate how and why certain behaviour may arise, going beyond parameter scanning-based model analysis and data analysis. We further emphasize that as the number of modifications increases the underlying modification networks become considerably more complex (with an exponential increase in the size of the network in the absence of ordering) and symmetric networks represent one of the few tractable vantage points from which to study such networks. (v) In addition to revealing distinct new information processing characteristics of multisite modification (for instance, exact absolute concentration robustness [ACR] of different types) of relevance in natural systems, this also serves as a potentially fruitful point of departure in engineering information processing circuits involving multisite modification in synthetic biology.

Our studies primarily focussed on different random mechanisms of double-site modification, with different combinations of common/separate kinase and common/separate phosphatase. These serve as a useful basis for investigation, noting that (i) these different types of enzyme combinations are widely encountered in cellular biology (*Stepanov et al., 2018*; *Lyons et al., 2013*; *Ramachandran et al., 1992*), and (ii) the double-site mechanism is among the simplest multisite modification system which both exhibits different types of symmetries and symmetry breaking. This provides a tractable case for understanding this behaviour transparently, serving as a basis for subsequent investigation of more complex modification scenarios. Furthermore, the insights obtained from our analysis suggest natural extensions and generalizations to modification networks with a greater number of modification sites. This is seen in preliminary studies of ordered triple-site modification systems (see *Source code 1* [Section 2.3], *Supplementary file 1*, and *Appendix 2—figure 8*).

### Symmetries

The symmetries which arise can be conceptualized and understood by examining symmetries of the underlying 'square' reaction network topology (refer *Figure 1D*). That viewpoint, relevant for a

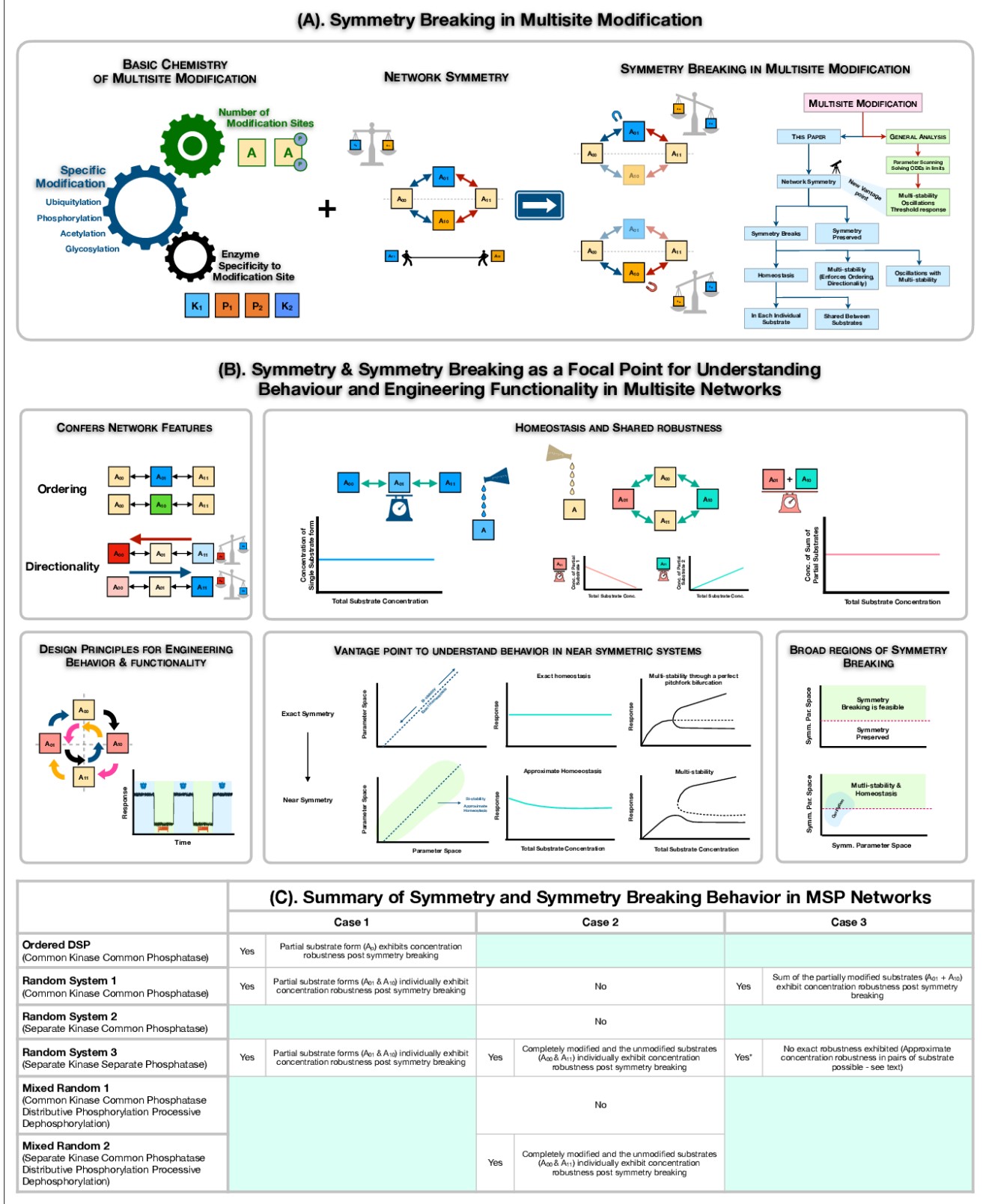

**Figure 4.** (**A**) Schematic representation of realization of symmetry and symmetry breaking in multisite modification networks through the interplay of basic biochemistry of post-translational protein modification and network symmetry. The analysis of the multisite network through symmetry and symmetry breaking reveals the underpinnings of new network features and serves as a rich and distinct vantage point to understand information processing behaviour in multisite modification networks. (**B**) Symmetry breaking as a new vista for understanding behaviour and engineering

*Figure 4 continued*
functionality in multisite modification networks: symmetry breaking can confer network features such as ordering and directionality to multisite phosphorylation (MSP) networks, limit the range of oscillations, and enable robust homeostasis for individual or combinations of substrates. It also provides key insights on the origin of behaviours in networks which are not far from symmetry. (**C**) A tabular summary of the presence and absence of symmetry and symmetry breaking in MSP networks, along with features of the symmetry-broken states (exact absolute concentration robustness).

broad class of chemical reaction networks, can then be applied to the specific instance of multisite substrate modification networks. Our studies involved a systematic analysis of different symmetries which emerge (noting the reasons above): symmetry in the modification direction (Case 1 symmetry: arising from a symmetry in action of the two enzymes on their corresponding substrates), symmetry between different branches of modification (Case 2 symmetry: arising from the same rates of modification to and from phosphoforms at a given level of modification: see text for distinction from Case 1) and combinations of the two (Case 3 symmetry: arising from a symmetry in the action of the two enzymes in the modification/demodification at a given site, but on opposite legs). The different types of symmetries are encountered among the different classes of multisite modification networks (with either common/separate kinase/phosphatase) though some networks may not exhibit all symmetries (summarized in *Figure 4C*).

## Which symmetries can be broken?

Case 1 symmetry can be broken in all the random modification networks where it exists. This symmetry breaking serves as a distinct mechanism for establishing directionality. Additionally, Case 1 symmetry breaking can also be observed in a simple *ordered* DSP network (with only a single partial phosphoform), though only in the common kinase, common phosphatase case. This ordered network serves as a simpler network for understanding this symmetry breaking transparently. Case 2 symmetry breaking is the basis of ordering of modifications. While Case 2 symmetry is possible for all combinations of common/separate kinase and common/separate phosphatase, the symmetry is broken only for the different kinase and different phosphatase case. A combination of the flexibility afforded by the different sets of enzymes, along with sufficient nonlinearity (due to enzymes participation in multiple complexes), enables this. Interestingly, if the dephosphorylation mechanism is processive (rather than distributive, as assumed throughout), the separate kinase common phosphatase network can also give rise to Case 2 symmetry breaking, reinforcing how the interplay of processive and distributive modification can enable new behaviour (see (*Suwanmajo and Krishnan, 2015*) for another example). In Case 3, the two symmetries necessarily break together. A distinct behaviour encountered here is the presence of symmetry breaking and oscillatory behaviour. In most cases, the symmetry-broken states are associated with transparent invariants which we analytically identify: these represent a behaviour reminiscent of absolute concentration robustness (discussed below). Additionally, the symmetry breaking manifests as a supercritical pitchfork bifurcation, while in Case 3 symmetry with common kinase/common phosphatase, a subcritical pitchfork bifurcation is observed, along with possible tristability. Our analytical work reveals how symmetry breaking (in all cases studied) may be accessed in large regions of (symmetric) parameter space by varying total enzyme/substrate concentrations, which represent easy to manipulate experimental factors (see *Appendix 2—figure 7*).

## Multisite modification and network symmetry breaking

It is worth viewing the above results from a different perspective: the breaking of symmetry in (potentially general) biochemical networks of the 'square' topology (discussed in *Figure 1D*). While symmetry simply imposes the restriction that the two legs of the network have kinetics which are identical, when can the symmetry actually break? Our study presents multiple insights: (i) firstly, a degree of nonlinearity is required, and this arises from conservation of species and sequestration of enzymes/substrates in complexes, a fundamental aspect of biochemical systems. All the modification networks we consider have enzymes shared between at least two enzyme-substrate complexes (this stems from the fact that a given modification is effected by only one enzyme, and without any ordering) and this provides the nonlinearity. (ii) On the other hand, for symmetry breaking to occur, a sufficient flexibility is required in the network to be able to allow for this. This is clearly seen in Case 2 symmetry in modification networks (with distributive enzyme mechanism), where reduced nonlinearity notwithstanding, it is only the separate kinase separate phosphatase modification network that allows symmetry to be

broken. In general, there is a trade-off between nonlinearity and flexibility (associated with distinct enzymes for different steps), but multisite modification provides many instances of sufficient combinations of both factors to realize symmetry breaking. These insights, bringing together basic (bio) chemistry and network features, are broadly relevant in biochemical networks.

## Enzyme sequestration

Enzyme sequestration (and competition) provides the key nonlinearity for generating symmetry breaking obviating the need for explicit feedback. Eliminating enzyme sequestration eliminates the possibility of symmetry breaking. Enzyme competition is a key ingredient in multisite modification, and in general this could combine with zeroth-order ultrasensitivity to generate new behaviour. However, the symmetry breaking we have found does not require any explicit assumption on the kinetic regime of enzymatic action (as seen from the sufficient conditions we have obtained) and so zeroth-order ultrasensitivity is neither necessary nor sufficient for this.

We now discuss the relevance of our results from different vantage points. All of these underscore the fact that information processing is a characteristic and consequence of the modification network (rather than an individual modification) and that symmetry and symmetry breaking provides distinct classes of insights therein (see *Figure 4*).

## Ordering and directionality in multisite modification

Multisite modification systems encountered in vivo often exhibit different degrees of ordering ranging from complete ordering of the sequence of modifications, to partial ordering, to a complete absence of ordering (symmetric scenario). Ordering is a fundamental aspect of substrate modification and its deployment in different pathways and processes. In fact, ordering of modifications is key to establishing a strictly sequential logic, which is likely to be an important aspect of information processing in those cellular contexts. A range of studies focus on these contexts, basic principles, and the potent role in engineering multisite modification (*Kocieniewski et al., 2012*; *Lyons et al., 2013*; *Ramachandran et al., 1992*; *Lössl et al., 2016*; *Valk et al., 2014*; *Kõivomägi et al., 2013*). How ordering has emerged is however unclear, and there could be multiple contributing factors. Our results indicate that the basic biochemistry of multisite modification by itself provides the basis for creating an ordering by breaking symmetry. The biasing which emerges can itself be very significant, and with possible additional refinements, gives rise to ordering. This demonstrates that a key driving factor could be at the modification network level rather than at the molecular level. Our analysis of the different symmetry cases allows us to explore the different ways in which both ordering and directionality may be determined. We determine explicit conditions for the occurrence of symmetry breaking, revealing broad ranges of parameter space where this can happen. In the context of ordering, this, along with the demonstration of sufficiency of the conditions for symmetry breaking, demonstrates the robustness of the mechanism. We further point out that even if the system is not exactly symmetric an echo of such symmetry breaking may be seen, which is indicated by multiple steady states which strongly bias one pathway over another, in a manner which is not commensurate with the (small) differences in kinetics of the pathways (*Appendix 2—figures 5; 6*).

Given a symmetric (Case 2 symmetry) or close to symmetric network where different phosphoforms behave (essentially) the same, there are different ways in which evolution could lead to biasing of one modification pathway over the other. One is by effecting local changes in one of the pathways. Symmetry breaking allows for a distinct mechanism whereby changing one easy to manipulate parameter (expression level of substrate), a significant biasing of one pathway over the other is established. This could be further reinforced (if this is a desirable outcome) by local changes in the pathway or increasing substrate amounts further (which further accentuates the biasing). This can lead to either partial or even complete ordering subsequently. Thus the mechanism could be seen as an efficient way of effecting a substantial change which could be reinforced and consolidated by further tinkering. It can also generate different robustness characteristics.

A similar comment applies to directionality. Case 3 symmetry breaking results in a combination of ordering and directionality, which ultimately manifests itself as elevated combinations of specific partial phosphoforms and unmodified/fully modified forms. Such a behaviour of the network (for instance, if observed experimentally) could easily be misinterpreted as suggestive of either some modification being inactive or there being a strong bias in the intrinsic kinetics, neither of which may

be correct. Our results also provide important insights in the cases of larger numbers of modification sites. For instance, analogues of Case 2 symmetry breaking could explain both ordering and partial ordering (some sequences of modifications ordered) in those systems.

## Absolute concentration robustness

Our analysis reveals the presence of (exact) ACR of different species in the symmetry-broken state. In this regard, we note that (i) the relevant species (in some cases, partial phosphoforms, and in others the fully modified or fully unmodified phosphoforms) exhibit concentration robustness to changes in total substrate concentration and are fixed at a level corresponding to the concentration of these species at the symmetry-breaking bifurcation (the inception of the asymmetric branch). (ii) Depending on the network and the type of symmetry broken, this can manifest itself as ACR for pairs of species (Case 1 and Case 2). (iii) In other cases (Case 3), the robustness is in the sum of concentrations of species, either exactly or approximately. From the above points, we see that multisite modification contains an in-built mechanism of creating robustness for clusters of species, either individually or collectively, something which represents an appealing characteristic for natural and engineered modification networks. It remains to be seen how this has been exploited in cells. (iv) There are different ways in which ACR may be obtained (for instance, in bifunctional enzymes (*Batchelor and Goulian, 2003*; *Krishnan et al., 2020*). The mechanism seen here shares a feature of ACR observed in auto-catalytic networks, arising from a transcritical bifurcation (*Shinar and Feinberg, 2011*; *Krishnan and Floros, 2019*) as being intimately tied to a nonlinear dynamic transition arising from the biochemistry. In both cases, there is more than one steady state possible, and one of the steady states exhibits the ACR.

## The origins of ACR

Based on the above, a natural question is which substrates could exhibit (exact) ACR and whether symmetry is a prerequisite. We note that in the ACR we have made no assumption/restriction or invoked any particular kinetic regime for enzymatic action. We answer the questions (based on analytical work: see Appendix 1 and *Appendix 2—figure 9*) relating to ACR in these terms in the ordered DSP network. (i) Only $A_p$ can exhibit ACR, and this occurs only in response to $A_{Total}$ (not $K_{Total}$ or $P_{Total}$). (ii) ACR necessarily requires multiple steady states, with two branches of steady states exhibiting ACR. There is another steady-state branch which does not exhibit ACR, but intersects one of the branches in what was computationally observed to be a transcritical bifurcation. (iii) There is a constraint on parameters to enable this, which is weaker than the symmetry condition. (iv) In the case of symmetry, the two ACR branches are symmetric and intersect with the other branch in a pitchfork bifurcation.

## Approximate ACR

As noted above, networks deviating from exact symmetry can exhibit approximate concentration robustness (refer *Appendix 2—figure 6*). Concentration robustness (approximate) could also be obtained in specific limiting kinetic parameter regimes. In ordered DSP with common kinase common phosphatase, we find that (i) $A_{pp}$ and $A$ could also exhibit concentration robustness. This can happen in a regime where the enzyme producing this (from $A_p$) acts in the saturated regime while the action of both enzymes on reactions not involving the species under consideration acts in the unsaturated limit (see Appendix 1). Here approximate ACR occurs without requiring multistability. (ii) Similarly approximate ACR can occur in $A_p$ without multistability by (for instance) having phosphorylation of A in the saturated regime and phosphorylation of $A_p$ and dephosphorylation of $A_{pp}$ in the unsaturated limit (or having phosphatases in excess). Similar insights can enable approximate ACR for one species in the corresponding random network (see Appendix 1). In contrast to such limiting regimes, absolute/approximate concentration robustness via symmetry breaking is present along with a rich repertoire of information processing characteristics.

## Pathways and modularity

How does symmetry breaking in multisite modification both affect and be affected by the behaviour of a signalling network of which it is a part? (i) The importance of multisite modification stems from the fact that it confers functionality to proteins which can regulate other processes. Here the symmetry breaking allows for both regulation of downstream pathways as well as insulation of some downstream

pathways from the effect of total upstream substrate and other upstream perturbations (via ACR for specific substrates in the modification network). This ability to insulate some parts of a network, while not the others, is a desirable feature which can be exploited: symmetry breaking in multisite modification provides a way of realizing this exactly purely from chemistry without requiring elaborate network structures (incorporating adaptation, feedback, etc.). (ii) Additionally, we find examples of 'shared ACR' where the sum of two species concentrations is fixed. This represents a case where robustness is applied to a combination of pathways if the two species regulate different pathways. This may be relevant in multiple cell signalling contexts by directly incorporating an inbuilt trade-off between the activation of two pathways, for instance, for efficient resource allocation. If the two species regulate the same pathway in the same way, this translates into robustness in regulating the pathway, while allowing flexibility through the redundancy. (iii) The effect of sequestration of a substrate species can be to either facilitate or make difficult the possibility of symmetry breaking. The sequestration of a substrate species in a downstream complex is the basis of a retroactive effect in a signalling pathway (*Ventura et al., 2010*; *Del Vecchio et al., 2008*). In the current case, this retroactive effect can help facilitate the possibility of symmetry breaking, and further that this happens in a context-dependent way. (iv) Other factors associated with the network, for instance, feedback, may also significantly affect the possibility of this happening. These aspects need to be assessed systematically and will be studied in the future. Interestingly an existing study (*Krishnamurthy et al., 2007*) examines sequential multisite modification with two explicit feedbacks: one from the maximally modified phosphoform increasing the probability of (every) modification and the other from the unmodified form increasing the probability of every demodification. In a stochastic setting, this has been shown to result in breaking a symmetry between phosphorylation and dephosphorylation even with no enzyme sequestration. In contrast to this, all our studies are on the intrinsic behaviour of multisite modification and in a deterministic setting.

## Relevance to oscillatory enzymatic networks

Studies of multisite networks have focused on their capacity of generating oscillations (*Rust et al., 2007*; *Van Zon et al., 2007*), including random networks with common kinase and common phosphatase with a view to their relevance in circadian oscillators (*Jolley et al., 2012*). A detailed computational study (*Jolley et al., 2012*) reveals regions of parameter space which facilitate the presence of oscillations, and the prominent regions are clustered around a symmetric network (the Case 3 network that we have studied). What is the relevance of our analysis here? Our study shows how oscillations occur in such cases, and also how (by changing substrate amounts) both oscillations and symmetry breaking may occur. We can identify different regimes based on our analysis. (i) A regime where symmetry breaking is ruled out. Here our analysis indicates regimes where oscillations can occur without any potential interference from symmetry breaking. (ii) A regime where symmetry breaking is possible, and in fact guaranteed, for some total substrate amount. In the latter case, we demonstrate that by varying total enzyme amounts (easily tuneable dials) it is possible to obtain, multistability, oscillations or a combination of such behaviour (see *Appendix 2—figure 7*). It indicates how in certain cases symmetry breaking may occur, limiting/preventing a range of oscillations. In particular, it indicates that oscillations do not have to be present for an indefinitely large range (suggested in the computations of (*Jolley et al., 2012*)). We provide further insights with regard to oscillations. (iii) The existence of long period oscillations which hover between different symmetric states is also seen (*Appendix 2—figure 2*). (iv) We demonstrate the possibility of oscillations in networks with different kinases and phosphatases, which potentially benefit from a greater tuneability than the common kinase common phosphatase case. (v) By contrast, we find that the other symmetries do not readily yield oscillatory behaviour, though further work needs to be done to study this exhaustively. The above points sharpen our understanding of oscillations emerging in random modification mechanisms and reinforce the theme of multisite modification as a complex information processor.

## Experimental signatures and testable predictions

Our analysis reveals the key features associated with symmetry breaking, which suggest multiple non-trivial signatures which could be seen experimentally: for instance, a considerable disparity in partial phosphoform behaviour, which may be incommensurate with the (minor) differences in kinetics in the legs of modification, characteristic patterns of ACR for specific species or groups.

These signatures even if approximate could suggest the presence of symmetry breaking. On the other hand, experiments could be developed to realize this behaviour by constructing underlying modification circuits either for synthetic purposes or to probe and test the behaviour itself. The multiplicity of enzymes involved allows for the deployment of a broad experimental tool kit for these purposes. Systems mimicking Case 2 symmetry could be created by engineering different modification sites (of similar properties), with modification by different isoforms of an enzyme. If this is also done for the demodification, then an approximate realization of a Case 2 symmetry (different kinase different phosphatase) can be realized. Alternatively, in a similar vein it may be possible to engineer a different kinase common phosphatase system with processive dephosphorylation: here the dephosphorylation could be induced to be processive (as seen elsewhere in cellular contexts). Other approaches could involve reconstitution of components of existing systems, such as circadian oscillators.

It is worth examining the implications and extensions of our study to a larger number of modification sites. Random networks lead to an exponential increase in the number of states. Additionally, the modifications/demodifications can be effected by common enzymes (for all modifications), distinct enzymes (for every modification), or a combination thereof, leading to a further combinatorial explosion in possibilities. Clearly direct analogues of the symmetry breaking seen here (e.g. Case 1 and Case 2) can be encountered here. In addition, new possibilities can emerge. In Case 2, for instance, in addition to the situation where all modification legs behave the same, we can have a situation where some modification legs (or parts thereof) are the same. Furthermore, not surprisingly, new behavioural characteristics can emerge. For instance, in the ordered triple-site phosphorylation network (Case 1 symmetry: common kinase common phosphatase), we find shared robustness (see *Appendix 2—figure 8*), not seen in the ordered double-site modification. These aspects need a dedicated study of their own and will be studied in the future. Viewed from the perspective of information storage, symmetry breaking suggests that a symmetric double-site modification network contains a bit of information. We emphasize, however, that symmetric network encodes a richer set of information, such as simultaneously presenting homeostasis and multiple steady states, an observation relevant to networks of any number of modification sites.

All in all, we have shown how basic biochemistry of multisite modification even within simple modification networks can be the basis of symmetry breaking. Symmetry breaking in turn can confer ordering, directionality, exact concentration robustness, and can significantly enhance the repertoire of information processing in multisite modification and regulation in signalling networks where they are present. The insights which arise from a structured systems study are of relevance in multiple contexts spanning the chemical and the biological, from systems biology to systems chemistry with potential in synthetic biology and the engineering of chemical systems.

## Acknowledgements

Funding to VR through a Presidential PhD scholarship at Imperial College London is gratefully acknowledged.

## Additional information

### Funding
The authors declare that no external funding was received for this work

### Author contributions
Vaidhiswaran Ramesh, formal-analysis, investigation, methodology, software, writing-original-draft, writing-review-and-editing, writing-methodology, investigation; J Krishnan, conceptualization, formal-analysis, investigation, methodology, supervision, writing-original-draft, writing-review-and-editing, writing-main-text-input-to-analysis, investigation

### Author ORCIDs
J Krishnan http://orcid.org/0000-0001-6196-2033

Decision letter and Author response
Decision letter https://doi.org/10.7554/65358.sa1
Author response https://doi.org/10.7554/65358.sa2

## Additional files

### Supplementary files
• Supplementary file 1. PDF version of the entire source code document (Maple file).
• Appendix 2—figure 1—source data 1.
• Appendix 2—figure 2—source data 1.
• Appendix 2—figure 3—source data 1.
• Appendix 2—figure 4—source data 1.
• Appendix 2—figure 5—source data 1.
• Appendix 2—figure 6—source data 1.
• Appendix 2—figure 7—source data 1.
• Appendix 2—figure 8—source data 1.
• Appendix 2—figure 9—source data 1.
• Source code 1. Maple document (Containing detailed proofs and parameter values used for the figures).

### Data availability
All data generated or analyzed during this study are included in this manuscript and supporting files. Relevant source code is also provided.

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

# Appendix 1

This paper analyses the propensity for symmetry and symmetry-breaking behaviour in networks involving multisite PTM of proteins. The study is conducted from the vantage point of a proto-typical example of PTMs, protein phosphorylation. MSP is a common PTM that confers functionality to proteins and is responsible for regulation of multiple cellular pathways. In this paper, multiple variations of double-site phosphorylation (DSP), realized through different combinations of common/separate kinases/phosphatases, random/ordered modifications, and the possibility of processivity in modification (refer to *Figures 1 and 4* and *Appendix 2— figure 1*), have been considered. By using the DSP as a suitable example, we have explored the phenomenon of symmetry and symmetry breaking in basic PTM networks. We present essential information about the methodology and analysis below, with additional details in *Source code 1* and *Supplementary file 1*. The information in this section is presented in the following order.

- Model descriptions
- Methodology
- Analytical proofs for arguments made in the main text
- Parameter values

## Model descriptions

We now present the kinetic descriptions of our modification networks. In all cases, we employ ODE-based kinetic descriptions as these focus on the central aspects of interest (the modifications and their sequence) for the purposes of our study. Spatial and stochastic aspects can be studied as extensions of this.

Consider enzyme action on a protein substrate with two modification sites and a common enzyme acting on both sites. The mechanism of modification could be either distributive or processive manner as described below:

Enzyme action: double-site modification (processive)

| Substrate | + | Enzyme | $\rightleftharpoons$ | Complex | $\rightleftharpoons$ | Mod. Com.1 | $\rightarrow$ | Mod. Sub. | + | Enzyme |
|---|---|---|---|---|---|---|---|---|---|---|
| $[A_{00}]$ | | $[K]$ | | $[A_{00}K]$ | | $[A_{01}K]$ | | $[A_{11}]$ | | $[K]$ |
| | | | | | OR | | | | | |
| $[A_{00}]$ | | $[K]$ | | $[A_{10}K]$ | | $[A_{10}K]$ | | $[A_{11}]$ | | $[K]$ |

Enzyme action: double-site modification (distributive)

| Substrate | + | Enzyme | $\rightleftharpoons$ | Complex1 | $\rightarrow$ | Mod. Sub. | + | Enzyme |
|---|---|---|---|---|---|---|---|---|
| $[A_{00}]$ | | $[K]$ | | $[(A_{00}K)_2]$ | | $[A_{01}]$ | | $[K]$ |
| $[A_{01}]$ | | $[K]$ | | $[A_{01}K]$ | | $[A_{11}]$ | | $[K]$ |
| | | | | OR | | | | |
| $[A_{00}]$ | | $[K]$ | | $[(A_{00}K)_2]$ | | $[A_{10}]$ | | $[K]$ |
| $[A_{10}]$ | | $[K]$ | | $[A_{10}K]$ | | $[A_{11}]$ | | $[K]$ |

In each case (processive and distributive), the enzyme (in the context of a phosphorylation and dephosphorylation – kinase and phosphatase, respectively) binds reversibly to the protein substrate to form an enzyme-substrate complex. If the enzyme action is processive, the enzyme is bound to the substrate (in multiple complexes) until all modifications are effected, after which it irreversibly detaches to give the completely modified protein and the free enzyme. In the context of a random double-site modification as shown, there are two possible ways in which the modifications can be (processively) effected, corresponding to the ordering in modifications, that is, first site being modified followed by the second site and vice versa. If the enzyme action is distributive, the enzyme detaches from the substrate after effecting each modification before reversibly binding again with the partially modified substrate to form a new complex corresponding to the additional modification. Similar to the processive enzyme action, there are two pathways (corresponding to different ordering) in which modifications can be effected in the double-site module as shown, each leading to a unique partially modified substrate. Combinations of processive and distributive enzyme action are also possible. Furthermore, additional variations in multisite modifications are possible depending on enzyme specificity. In the example above, we considered a case where both modification sites were effected by a common enzyme (kinase/ phosphatase). However,(de)modification of each modification site can be effected by a different

enzyme, and thus the possibility of having common/separate enzymes performing different modifications represents a suite of basic modification scenarios.

These complexities (for the double-site modification of substrate) are captured in the models depicted in *Figure 1*. The networks are modelled as a system of ODEs generated by describing the individual elementary reactions (as shown above in *Figure 1* and *Appendix 2—figure 10*) using generic mass action kinetic descriptions. This makes no assumptions on the kinetic regime of modification.

The nomenclature used for modelling is as follows. The kinetic constants of the binding and unbinding reactions are denoted by the letters $k_{bi}$ and $k_{ubi}$ while that of the irreversible catalytic reaction of the complex is denoted by $k_i$; where   is an index which stands for the reaction number in a given model. For the sake of clarity and brevity, models represented in the figures in the main text (refer *Figure 1*) use $\kappa_i$ and $\alpha_i$ to concisely represent binding, unbinding, and catalytic reaction rate constants involved in a particular modification. An individual modification/demodification step involves the enzyme reversibly binding to the substrate to form a complex, which then involves an irreversible catalytic reaction to produce the modified form (and release the enzyme): this represents a widely employed model of substrate modification. The detailed model description for the networks considered in the paper is constructed by employing such mass action kinetic descriptions of the elementary reactions for all modification/demodification steps (refer to *Appendix 2—figure 10* for a detailed schematic denoting independent reactions and their kinetic nomenclature). All models are presented in dimensionless form.

## Ordered distributive multisite phosphorylation

### DSP: common kinase common phosphatase

The ordered distributive DSP with common kinase common phosphatase acting on both modification sites (refer *Figure 1C*) is realized when the order of phosphorylation and order of dephosphorylation are opposite. This results in a single partially modified (phosphorylated) substrate. In the model description, $[A]$ represents the substrate while its subscript designation represents the number of modifications effected, that is, $[A_p]$ represents the partially modified substrate and $[A_{pp}]$ represents the completely modified substrate. As shown in the schematic, the substrate ($[A]$) first reversibly binds to the kinase ($[K]$) to form an enzyme-substrate complex ($[AK]$). The modification is then effected with the complex irreversibly producing the modified substrate ($[A_p]$) and the free enzyme ($[A_{pp}]$). Phosphorylation and dephosphorylation proceed in a distributive manner at both modification sites. Using mass action kinetics and the kinetic nomenclature described earlier, we can model the system as a set of ODEs as follows:

$$
\begin{aligned}
\frac{d[A]}{dt} &= -k_{b1}[A][K] + k_{ub1}[AK] + k_4[A_pP] \\
\frac{d[A_p]}{dt} &= -k_{b2}[A_p][K] - k_{b4}[A_p][P] + k_1[AK] + k_{ub2}[A_pK] + k_{ub4}[A_pP] + k_3[A_{pp}P] \\
\frac{d[A_{pp}]}{dt} &= -k_{b3}[A_{pp}][P] + k_2[A_pK] + k_{ub3}[A_{pp}P] \\
\frac{d[AK]}{dt} &= k_{b1}[A][K] - (k_{ub1} + k_1)[AK] \\
\frac{d[A_pK]}{dt} &= k_{b2}[A_p][K] - (k_{ub2} + k_2)[A_pK] \\
\frac{d[A_{pp}P]}{dt} &= k_{b3}[A_{pp}][P] - (k_{ub3} + k_3)[A_{pp}P] \\
\frac{d[A_pP]}{dt} &= k_{b4}[A_p][P] - (k_{ub4} + k_4)[A_pP] \\
\frac{d[K]}{dt} &= -k_{b1}[A][K] + (k_{ub1} + k_1)[AK] - k_{b2}[A_p][K] + (k_{ub2} + k_2)[A_pK] \\
\frac{d[P]}{dt} &= -k_{b3}[A_{pp}][P] + (k_{ub3} + k_3)[A_{pp}P] - k_{b4}[A_p][P] + (k_{ub4} + k_4)[A_pP]
\end{aligned}
\tag{1}
$$

The total substrate and enzyme concentrations are also conserved in this system. This is represented mathematically as follows:

$$
\begin{aligned}
A_{Total} &= [A] + [A_p] + [A_{pp}] + [AK] + [A_pK] + [A_{pp}P] + [A_pP] \\
K_{Total} &= [K] + [A_pK] + [AK] \\
P_{Total} &= [P] + [A_{pp}P] + [A_pP]
\end{aligned}
\tag{2}
$$

## Triple-site phosphorylation (TSP) – common kinase common phosphatase

While DSP networks are the primary focal point of the study, we use the ordered distributive TSP to make some specific points. The ordered distributive TSP with common kinase and common phosphatase acting on all modification sites is realized when the order of phosphorylation and dephosphorylation is reversed (see *Appendix 2—figure 8*). The network is modelled as a system of ODEs using mass action kinetic description of the elementary reactions involved. Similar to the ordered distributive DSP, the substrate is represented by $[A]$, while the subscript represents the number of phosphorylated modification sites. The equations are presented in *Source code 1* (Section 2.3) and *Supplementary file 1*.

## Random double-site modification

The random DSP network is realized when there is no preferential ordering for either the modification/demodification (phosphorylation/dephosphorylation). In this paper, we have considered multiple variations of random DSP, depending on whether the enzymes (kinases/phosphatases) effecting the modifications are the same or different (namely, systems 1–3, refer *Figure 1*).

The general nomenclature used in these models is as follows. The substrate is denoted by the letter $A$, while the subscript with binary digits is used to denote the nature of the specific modification sites. '1' represents a phosphorylated modification site, while '0' represents that the specific site is unphosphorylated. Thus $[A_{00}]$ and $[A_{11}]$ represent the completely unmodified and completely modified substrates, while $[A_{01}]$ and $[A_{10}]$ represent partially modified substrates with only the second and first site modified, respectively. Where required, the enzyme-substrate complexes are further designated with subscript numbers (outside curved brackets) to differentiate between complexes where the enzymes are associated with different modification (e.g. $[(A_{00}K)_1]$ and $[A_{11}]$).

The ODE models are constructed in a similar manner to those of the ordered mechanism by combining mass action kinetic descriptions of the elementary steps. For a detailed schematic including kinetic representation of the elementary binding and unbinding reactions, please refer to *Appendix 2—figure 10*.

## System 1 – common kinase common phosphatase:

System 1 has a single kinase and phosphatase that effects phosphorylation and dephosphorylation, respectively, on both the modification sites. Using the above nomenclature, this system can be represented as a system of ODEs as follows:

$$\frac{d[A_{00}]}{dt} = -k_{b1}[A_{00}][K] - a_{b1}[A_{00}][K] + k_{ub1}[(A_{00}K)_1] + a_{ub1}[A_{00}K_2] + a_4[A_{10}P] + k_4[A_{01}P]$$

$$\frac{d[A_{10}]}{dt} = -a_{b2}[A_{10}][K] - a_{b4}[A_{10}][P] + a_1[(A_{00}K)_2] + a_{ub2}[A_{10}K] + a_{ub4}[A_{10}P] + a_3[(A_{11}P)_2]$$

$$\frac{d[A_{01}]}{dt} = -k_{b2}[A_{01}][K] - k_{b4}[A_{01}][P] + k_1[(A_{00}K)_1] + k_{ub2}[A_{01}K] + k_{ub4}[A_{01}P] + k_3[(A_{11}P)_1]$$

$$\frac{d[A_{11}]}{dt} = -k_{b3}[A_{11}][P] - a_{b3}[A_{11}][P] + a_2[A_{10}K] + k_2[A_{01}K] + k_{ub3}[(A_{11}P)_1] + a_{ub3}[(A_{11}P)_2]$$

$$\frac{d[(A_{00}K)_1]}{dt} = k_{b1}[A_{00}][K] - (k_{ub1} + k_1)[(A_{00}K)_1]$$

$$\frac{d[A_{01}K]}{dt} = k_{b2}[A_{01}][K] - (k_{ub2} + k_2)[A_{01}K]$$

$$\frac{d[(A_{00}K)_2]}{dt} = a_{b1}[A_{00}][K] - (a_{ub1} + a_1)[(A_{00}K)_2]$$

$$\frac{d[A_{10}K]}{dt} = a_{b2}[A_{10}][K] - (a_{ub2} + a_2)[A_{10}K]$$

$$\frac{d[(A_{11}P)_1]}{dt} = k_{b3}[A_{11}][P] - (k_{ub3} + k_3)[(A_{11}P)_1]$$

$$\frac{d[A_{01}P]}{dt} = k_{b4}[A_{01}][P] - (k_{ub4} + k_4)[A_{01}P]$$

$$\frac{d[(A_{11}P)_2]}{dt} = a_{b3}[A_{11}][P] - (a_{ub3} + a_3)[(A_{11}P)_2]$$

$$\frac{d[A_{10}P]}{dt} = a_{b4}[A_{10}][P] - (a_{ub4} + a_4)[A_{10}P]$$

$$\frac{d[K]}{dt} = -k_{b1}[A_{00}][K] + (k_1 + k_{ub1})[(A_{00}K)_1] - a_{b1}[A_{00}][K]$$
$$+ (a_{ub1} + a_1)[(A_{00}K)_2] - a_{b2}[A_{10}][K] + (a_{ub2} + a_2)[A_{10}K]$$
$$- k_{b2}[A_{01}][K] + (k_2 + k_{ub2})[A_{01}K]$$

$$\frac{d[P]}{dt} = -k_{b3}[A_{11}][P] + (k_{ub3} + k_3)[(A_{11}P)_1] - a_{b3}[A_{11}][P]$$
$$+ (a_{ub3} + a_3)[(A_{11}P)_2] - a_{b4}[A_{10}][P] + (a_{ub4} + a_4)[A_{10}P]$$
$$- k_{b4}[A_{01}][P] + (k_{ub4} + k_4)[A_{01}P]$$

(3)

The total substrate and enzyme concentrations are conserved in the system. This is represented mathematically as follows:

$$A_{Total} = [A_{00}] + [A_{10}] + [A_{01}] + [A_{11}] + [(A_{00}K)_1] + [A_{01}K] + [(A_{00}K)_2]$$
$$+ [A_{10}K] + [(A_{11}P)_1] + [A_{10}P] + [(A_{11}P)_2] + [A_{01}P]$$

$$K_{Total} = [K] + [(A_{00}K)_1] + [A_{10}K] + [(A_{00}K)_2] + [A_{01}K]$$

$$P_{Total} = [P] + [(A_{11}P)_1] + [A_{10}P] + [(A_{11}P)_2] + [A_{01}P]$$

(4)

## System 2 – separate kinase common phosphatase:

System 2 has a distinct kinase acting on each modification site, while a common phosphatase effects dephosphorylation on both sites. Using the nomenclature discussed earlier, the system is modelled as a set of ODEs as shown below:

$$\frac{d[A_{00}]}{dt} = -k_{b1}[A_{00}][K_1] - a_{b1}[A_{00}][K_2] + k_{ub1}[A_{00}K_1] + a_{ub1}[A_{00}K_2] + k_4[A_{01}P] + a_4[A_{10}P]$$

$$\frac{d[A_{01}]}{dt} = -k_{b2}[A_{01}][K_2] - k_{b4}[A_{01}][P] + k_1[A_{00}K_1] + k_{ub2}[A_{01}K_2] + k_{ub4}[A_{01}P] + k_3[(A_{11}P)_1]$$

$$\frac{d[A_{10}]}{dt} = -a_{b2}[A_{10}][K_1] - a_{b4}[A_{10}][P] + a_1[A_{00}K_2] + a_{ub2}[A_{10}K_1] + a_{ub4}[A_{10}P] + a_3[(A_{11}P)_2]$$

$$\frac{d[A_{11}]}{dt} = -k_{b3}[A_{11}][P] - a_{b3}[A_{11}][P] + k_2[A_{01}K_2] + a_2[A_{10}K_1] + k_{ub3}[(A_{11}P)_1] + a_{ub3}[(A_{11}P)_2]$$

$$\frac{d[A_{00}K_1]}{dt} = k_{b1}[A_{00}][K_1] - (k_1 + k_{ub1})[A_{00}K_1]$$

$$\frac{d[A_{10}K_1]}{dt} = a_{b2}[A_{10}][K_1] - (a_2 + a_{ub2})[A_{10}K_1]$$

$$\frac{d[A_{00}K_2]}{dt} = a_{b1}[A_{00}][K_2] - (a_{ub1} + a_1)[A_{00}K_2]$$

$$\frac{d[A_{01}K_2]}{dt} = k_{b2}[A_{01}][K_2] - (k_{ub2} + k_2)[A_{01}K_2]$$

$$\frac{d[(A_{11}P)_1]}{dt} = k_{b3}[A_{11}][P] - (k_{ub3} + k_3)[(A_{11}P)_1] \tag{5}$$

$$\frac{d[A_{10}P]}{dt} = a_{b4}[A_{10}][P] - (a_{ub4} + a_4)[A_{10}P]$$

$$\frac{d[(A_{11}P)_2]}{dt} = a_{b3}[A_{11}][P] - (a_{ub3} + a_3)[(A_{11}P)_2]$$

$$\frac{d[A_{01}P]}{dt} = k_{b4}[A_{01}][P] - (k_{ub4} + k_4)[A_{01}P]$$

$$\frac{d[K_1]}{dt} = -k_{b1}[A_{00}][K_1] + (k_1 + k_{ub1})[A_{00}K_1] - a_{b2}[A_{10}][K_1] + (a_2 + a_{ub2})[A_{10}K_1]$$

$$\frac{d[K_2]}{dt} = -a_{b1}[A_{00}][K_2] + (a_{ub1} + a_1)[A_{00}K_2] - k_{b2}[A_{01}][K_2] + (k_{ub2} + k_2)[A_{01}K_2]$$

$$\frac{d[P]}{dt} = -k_{b3}[A_{11}][P] + (k_{ub3} + k_3)[(A_{11}P)_1] - a_{b4}[A_{10}][P] + (a_{ub4} + a_4)[A_{10}P]$$

$$-a_{b3}[A_{11}][P] + (a_{ub3} + a_3)[(A_{11}P)_2] - k_{b4}[A_{01}][P] + (k_{ub4} + k_4)[A_{01}P]$$

The total substrate and enzyme concentrations are conserved in this system. Note that each distinct kinase is associated with a conservation condition.

$$
\begin{aligned}
A_{Total} &= [A_{00}] + [A_{10}] + [A_{01}] + [A_{11}] + [A_{00}K_1] + [A_{01}K_2] + [A_{00}K_2] \\
&\quad + [A_{10}K_1] + [(A_{11}P)_1] + [A_{10}P] + [(A_{11}P)_2] + [A_{01}P] \\
K1_{Total} &= [K_1] + [A_{00}K_1] + [A_{10}K_1] \\
K2_{Total} &= [K_2] + [A_{00}K_2] + [A_{01}K_2] \\
P_{Total} &= [P] + [(A_{11}P)_1] + [A_{10}P] + [(A_{11}P)_2] + [A_{01}P]
\end{aligned}
\tag{6}
$$

## System 3 – separate kinase separate phosphatase:

System 3 has distinct kinases and phosphatases effecting phosphorylation and dephosphorylation on each modification site. The system is modelled as a set of ODEs as shown below:

$$\frac{d[A_{00}]}{dt} = -k_{b1}[A_{00}][K_1] - a_{b1}[A_{00}][K_2] + k_{ub1}[A_{00}K_1] + a_{ub1}[A_{00}K_2] + k_4[A_{01}P_1] + a_4[A_{10}P_2]$$

$$\frac{d[A_{01}]}{dt} = -k_{b2}[A_{01}][K_2] - k_{b4}[A_{01}][P_1] + k_1[A_{00}K_1] + k_{ub2}[A_{01}K_2] + k_{ub4}[A_{01}P_1] + k_3[A_{11}P_2]$$

$$\frac{d[A_{10}]}{dt} = -a_{b2}[A_{10}][K_1] - a_{b4}[A_{10}][P_2] + a_1[A_{00}K_2] + a_{ub2}[A_{10}K_1] + a_{ub4}[A_{10}P_2] + a_3[(A_{11}P_1]$$

$$\frac{d[A_{11}]}{dt} = -a_{b3}[A_{11}][P_1] - k_{b3}[A_{11}][P_2] + k_2[A_{01}K_2] + a_2[A_{10}K_1] + a_{ub3}[A_{11}P_1] + k_{ub3}[A_{11}P_2]$$

$$\frac{d[A_{00}K_1]}{dt} = k_{b1}[A_{00}][K_1] - (k_1 + k_{ub1})[A_{00}K_1]$$

$$\frac{d[A_{01}K_2]}{dt} = k_{b2}[A_{01}][K_2] - (k_2 + k_{ub2})[A_{01}K_2]$$

$$\frac{d[A_{11}P_2]}{dt} = k_{b3}[A_{11}][P_2] - (k_3 + k_{ub3})[A_{11}P_2]$$

$$\frac{d[A_{01}P_1]}{dt} = k_{b4}[A_{01}][P_1] - (k_4 + k_{ub4})[A_{01}P_1]$$

$$\frac{d[A_{00}K_2]}{dt} = a_{b1}[A_{00}][K_2] - (a_1 + a_{ub1})[A_{00}K_2] \tag{7}$$

$$\frac{d[A_{10}K_1]}{dt} = a_{b2}[A_{10}][K_1] - (a_2 + a_{ub2})[A_{10}K_1]$$

$$\frac{d[A_{11}P_1]}{dt} = a_{b3}[A_{11}][P_1] - (a_3 + a_{ub3})[A_{11}P_1]$$

$$\frac{d[A_{10}P_2]}{dt} = a_{b4}[A_{10}][P_2] - (a_4 + a_{ub4})[A_{10}P_2]$$

$$\frac{d[K_1]}{dt} = k_{b1}[A_{00}][K_1] - (k_1 + k_{ub1})[A_{00}K_1] + a_{b2}[A_{10}][K_1] - (a_2 + a_{ub2})[A_{10}K_1]$$

$$\frac{d[K_2]}{dt} = a_{b1}[A_{00}][K_2] - (a_1 + a_{ub1})[A_{00}K_2] + k_{b2}[A_{01}][K_2] - (k_2 + k_{ub2})[A_{01}K_2]$$

$$\frac{d[P_1]}{dt} = a_{b3}[A_{11}][P_1] - (a_3 + a_{ub3})[A_{11}P_1] + k_{b4}[A_{01}][P_1] - (k_4 + k_{ub4})[A_{01}P_1]$$

$$\frac{d[P_2]}{dt} = k_{b3}[A_{11}][P_2] - (k_3 + k_{ub3})[A_{11}P_2] + a_{b4}[A_{10}][P_2] - (a_4 + a_{ub4})[A_{10}P_2]$$

The conservation condition for the substrate and each of the enzymes is shown below:

$$
\begin{aligned}
A_{Total} &= [A_{00}] + [A_{10}] + [A_{01}] + [A_{11}] + [A_{00}K_1] + [A_{01}K_2] + [A_{11}P_2] + [A_{01}P_1] \\
&\quad + [A_{00}K_2] + [A_{10}K_1] + [A_{11}P_1] + [A_{10}P_2] \\
K1_{Total} &= [A_{00}K_1] + [A_{10}K_1] + [K_1] \\
K2_{Total} &= [A_{01}K_2] + [A_{00}K_2] + [K_2] \\
P1_{Total} &= [A_{11}P_1] + [A_{01}P_1] + [P_1] \\
P2_{Total} &= [A_{10}P_2] + [A_{11}P_2] + [P_2]
\end{aligned}
\tag{8}
$$

## Mixed random DSP

The 'mixed random' DSP which we study involves a combination of distributive phosphorylation and processive dephosphorylation, without any ordering. In this paper, multiple such mixed random modification networks (mixed random 1, 2, and 3; refer *Figure 1* and *Appendix 2—figure 1*) are analysed from a specific viewpoint: their individual capacity to exhibit Case 2 symmetry breaking. The goal there is to establish whether having processive modification results in any new insights. The model description for these systems is as follows.

### Mixed random 1 – common kinase common phosphatase

Mixed random 1 has a common kinase that effects phosphorylation on both the modification sites. Using the above nomenclature, this system can be represented as a system of ODEs as follows:

$$\frac{d[A_{00}]}{dt} = -k_{b1}[A_{00}][K] - a_{b1}[A_{00}][K] + k_{ub1}[A_{00}K_1] + a_{ub1}[A_{00}K_2] + k_4[A_{01}P] + a_4[A_{10}P]$$

$$\frac{d[A_{01}]}{dt} = -k_{b2}[A_{01}][K] + k_1[(A_{00}K)_1] + k_{ub2}[A_{01}K]$$

$$\frac{d[A_{10}]}{dt} = -a_{b2}[A_{10}][K] + a_1[(A_{00}K)_2] + a_{ub2}[A_{10}K]$$

$$\frac{d[A_{11}]}{dt} = -k_{b3}[A_{11}][P] - a_{b3}[A_{11}][P] + k_2[A_{01}K] + a_2[A_{10}K] + k_{ub3}[(A_{11}P)_1] + a_{ub3}[A_{11}P_2]$$

$$\frac{d[(A_{00}K)_1]}{dt} = k_{b1}[A_{00}][K] - (k_1 + k_{ub1})[(A_{00}K)_1]$$

$$\frac{d[A_{10}K]}{dt} = a_{b2}[A_{10}][K] - (a_2 + a_{ub2})[A_{10}K]$$

$$\frac{d[(A_{00}K)_2]}{dt} = a_{b1}[A_{00}][K] - (a_{ub1} + a_1)[(A_{00}K)_2]$$

$$\frac{d[A_{01}K]}{dt} = k_{b2}[A_{01}][K] - (k_{ub2} + k_2)[A_{01}K] \tag{9}$$

$$\frac{d[(A_{11}P)_1]}{dt} = k_{b3}[A_{11}]P - (k_{ub3} + k_3)[(A_{11}P)_1]$$

$$\frac{d[A_{10}P]}{dt} = -[A_{10}P]a_4 + [A_{11}P_2]a_3$$

$$\frac{d[A_{11}P_2]}{dt} = a_{b3}[A_{11}][P] - (a_{ub3} + a_3)[A_{11}P_2]$$

$$\frac{d[A_{01}P]}{dt} = -[A_{01}P]k_4 + [(A_{11}P)_1]k_3$$

$$\frac{d[K]}{dt} = -k_{b1}[A_{00}][K] + (k_1 + k_{ub1})[(A_{00}K)_1] - a_{b2}[A_{10}][K] + (a_2 + a_{ub2})[A_{10}K] - a_{b1}[A_{00}][K]$$
$$+ (a_{ub1} + a_1)[(A_{00}K)_2] - k_{b2}[A_{01}][K] + (k_{ub2} + k_2)[A_{01}K]$$

$$\frac{d[P]}{dt} = -k_{b3}[A_{11}][P] + (k_{ub3} + k_3)[(A_{11}P)_1] + a_4[A_{10}P] - a_3[A_{11}P_2]$$
$$- a_{b3}[A_{11}][P] + (a_{ub3} + a_3)[A_{11}P_2] + k_4[A_{01}P] - k_3[(A_{11}P)_1]$$

The total concentrations of substrates and enzymes are conserved, and this is depicted below:

$$A_{Total} = [A_{00}] + [A_{10}] + [A_{01}] + [A_{11}] + [A_{00}K_1] + [A_{01}K] + [A_{00}K_2]$$
$$+ [A_{10}K] + [(A_{11}P)_1] + [A_{10}P] + [(A_{11}P)_2] + [A_{01}P]$$

$$K_{Total} = [K_1] + [A_{00}K_1] + [A_{10}K] + [A_{01}K] + [(A_{00}K)_2] \tag{10}$$

$$P_{Total} = [P] + [(A_{11}P)_1] + [A_{10}P] + [(A_{11}P)_2] + [A_{01}P]$$

## Mixed random 2 – separate kinase common phosphatase: mixed random 2

This modification network differs from the previous one in one respect: distinct kinases effect phosphorylation on each modification site, while a common phosphatase performs the dephosphorylation in a processive manner, just as described above. This system can be represented as a system of ODEs as follows:

$$\frac{d[A_{00}]}{dt} = -k_{b1}[A_{00}][K_1] - a_{b1}[A_{00}][K_2] + k_{ub1}[A_{00}K_1] + a_{ub1}[A_{00}K_2] + k_4[A_{01}P] + a_4[A_{10}P]$$

$$\frac{d[A_{01}]}{dt} = -k_{b2}[A_{01}][K_2] + k_1[A_{00}K_1] + k_{ub2}[A_{01}K_2]$$

$$\frac{d[A_{10}]}{dt} = -a_{b2}[A_{10}][K_1] + a_1[A_{00}K_2] + a_{ub2}[A_{10}K_1]$$

$$\frac{d[A_{11}]}{dt} = -k_{b3}[A_{11}][P] - a_{b3}[A_{11}][P] + k_2[A_{01}K_2] + a_2[A_{10}K_1] + k_{ub3}[(A_{11}P)_1] + a_{ub3}[(A_{11}P)_2]$$

$$\frac{d[A_{00}K_1]}{dt} = k_{b1}[A_{00}][K_1] - (k_1 + k_{ub1})[A_{00}K_1]$$

$$\frac{d[A_{10}K_1]}{dt} = a_{b2}[A_{10}][K_1] - (a_2 + a_{ub2})[A_{10}K_1]$$

$$\frac{d[A_{00}K_2]}{dt} = a_{b1}[A_{00}][K_2] - (a_{ub1} + a_1)[A_{00}K_2]$$

$$\frac{d[A_{01}K_2]}{dt} = k_{b2}[A_{01}][K_2] - (k_{ub2} + k_2)[A_{01}K_2] \tag{11}$$

$$\frac{d[A_{11}P_1]}{dt} = k_{b3}[A_{11}][P] - (k_{ub3} + k_3)[(A_{11}P)_1]$$

$$\frac{d[A_{10}P]}{dt} = -a_4[A_{10}P] + a_3[(A_{11}P)_2]$$

$$\frac{d[A_{11}P_2]}{dt} = a_{b3}[A_{11}][P] - (a_{ub3} + a_3)[(A_{11}P)_2]$$

$$\frac{d[A_{01}P]}{dt} = -k_4[A_{01}P] + k_3[(A_{11}P)_1]$$

$$\frac{d[K_1]}{dt} = -k_{b1}[A_{00}][K_1] + (k_1 + k_{ub1})[A_{00}K_1] - a_{b2}[A_{10}][K_1] + (a_2 + a_{ub2})[A_{10}K_1]$$

$$\frac{d[K_2]}{dt} = -a_{b1}[A_{00}][K_2] + (a_{ub1} + a_1)[A_{00}K_2] - k_{b2}[A_{01}][K_2] + (k_{ub2} + k_2)[A_{01}K_2]$$

The total concentrations of the substrate and respective enzymes are conserved as shown below:

$$
\begin{aligned}
A_{Total} &= [A_{00}] + [A_{10}] + [A_{01}] + [A_{11}] + [A_{00}K_1] + [A_{01}K_2] + [A_{00}K_2] \\
&\quad + [A_{10}K_1] + [(A_{11}P)_1] + [[A_{10}]P] + [(A_{11}P)_2] + [A_{01}P] \\
K1_{Total} &= [K_1] + [A_{00}K_1] + [A_{10}K_1] \\
K2_{Total} &= [K_2] + [A_{00}K_2] + [A_{01}K_2] \\
P_{Total} &= [P] + [(A_{11}P)_1] + [[A_{10}]P] + [(A_{11}P)_2] + [A_{01}P]
\end{aligned}
\tag{12}
$$

## Mixed random 3 – separate kinase common phosphatase (unsaturated phosphorylation)

The model of mixed random 3 is similar to that of the mixed random 2 network, except that the dephosphorylation of the fully modified to unmodified form is described by a pair of linear reaction. This can happen when the catalytic constants for the dephosphorylation are significantly higher than the binding/unbinding of substrate to the phosphatase. The model is depicted below:

$$
\begin{aligned}
\frac{d[A_{00}]}{dt} &= -k_{b1}[A_{00}][K_1] - a_{b1}[A_{00}][K_2] + k_{ub1}[A_{00}K_1] + a_{ub1}[A_{00}K_2] + k_3[A_{11}] + a_3[A_{11}] \\
\frac{d[A_{11}]}{dt} &= k_2[A_{01}K_2] + a_2[A_{10}K_1] - k_3[A_{11}] - a_3[A_{11}]
\end{aligned}
\tag{13}
$$

## Methodology

Our approach to analyse the different networks in this paper relies on a careful balance of analytical and computational work. Through these two strands, we clearly isolate the presence or absence of classes of symmetry breaking, and where possible elucidate the necessary and sufficient conditions for this to occur. The networks described above as system of ODEs were simulated using the ode15s solver in MATLAB *Shampine and Reichelt, 1997*. The results of the simulations were complemented and cross-verified using the computational software COPASI (*Hoops et al., 2006*). COPASI automatically generates the system of ODEs based on provision of the network schematic and thus is used to cross-validate the MATLAB models. Bifurcation analysis was carried out using the computational package 'MatCont' *Dhooge et al., 2003* in MATLAB, and the symbolic software package *Maple Inc, 2018* was used to cross-verify the analytical results.

The bifurcation analysis presented in the paper is performed by varying the total substrate concentration $A_{Total}$ is a natural choice for a bifurcation parameter in this context as varying this parameter still accommodates the different classes of symmetries (Cases 1–3). However, the symmetry breaking in all cases reported can also be isolated from bifurcation along any kinetic parameter (or parameter pair) or total enzyme concentrations as long as the original symmetric structure is maintained.

We comment here that echoes of the symmetry breaking observed here are also seen when exact symmetry is not maintained (as shown in *Appendix 2—figures 5; 6*). Further discussion on approximately symmetric systems and echoes of 'symmetry breaking' in such contexts is presented in the main text.

## Parameters

The parameters used to generate the figures in the paper are presented in Appendix 2. Our results focus on instances of symmetry breaking and associated behaviour. The parameters used are generic and are in ranges commensurate with values typically reported in literature. We emphasize that we use computation here primarily as a tool to complement analysis from analytical work and to show the presence of symmetry breaking and associated features in a model. Thus, in this spirit, the values used are only representative and the behaviour is seen in a well-defined region of the parameter space (as we demonstrate in the analytical work). Analytical work also explicitly reveals basic features about the symmetry-broken state seen computationally. In networks where symmetry breaking does not occur, this fact is established analytically. The analytical results concerning features of symmetry breaking and the asymmetric states are cross-validated by computational analysis (bifurcation analysis). This cross-validation is presented along with the parameters used in the Maple file *Source code 1* and *Supplementary file 1*.

Mass action kinetic descriptions are used for the elementary reactions involved as this does not make any assumptions regarding regimes of enzyme activity and complex formation (unlike other reaction descriptions such as Hill kinetics and Michaelis–Menten kinetics). Thus the results

and model behaviour shown are not artefacts of the choice of kinetic description used and are representative of the true functionality of these networks.

## Computational resources:

A Maple file with the model descriptions and detailed analytical proofs is presented along with this text (see *Source code 1*). A PDF copy of the entire Maple document is also provided (see *Supplementary file 1*), with the files printed in the same order as they appear within the Maple document, for easy accessibility.

## Analytical proofs for arguments made in the main text

This section contains a summary of the basic analytical arguments used to explore the feasibility of symmetry breaking and its associated features in the various MSP networks considered in this paper. This analysis is expanded on in detail in *Source code 1* and *Supplementary file 1*.

The symmetry in the context of our models is established through a strict kinetic structure and enforcement of total enzyme concentrations which ensures that certain pairs of kinetic terms are equal (refer *Figure 1*). Note that this requires, in general, the binding, unbinding, and catalytic constants to be the same. This ensures that starting with symmetric initial conditions for the appropriate variables (substrates and associated enzymes and complexes), the system evolves maintaining this symmetry. In this context, we point out that all three constants could affect the modification rate, and in fact studies *Hatakeyama and Kaneko, 2014*; *Hatakeyama and Kaneko, 2020* show how even unbinding constants significantly affect efficacy of modification.

The equality of reaction rates results in symmetries in terms of concentrations of substrates at (one of) the steady state(s) of the system. The phenomenon of symmetry breaking allows for this symmetric state to lose stability, giving rise to stable asymmetric steady states. In this section, we show the infeasibility of asymmetric states existing in some networks, thereby ruling out any symmetry breaking; in other instances, we complement computational results showing symmetry breaking in other networks with analytical results, including necessary and sufficient conditions for symmetry breaking. This is done by solving the system of ODEs of the associated model at steady state and isolating asymmetric steady states therein (if they exist).

In each of the networks considered, we first isolate the substrate and enzyme pairs that share symmetry (symmetry breaking leading to asymmetric states thus naturally implies that this symmetric pairing is no longer maintained). For example, in Case 1 symmetry of an ordered distributive DSP with common kinase and common phosphatase, the enzyme pairs $[K]$ & $[P]$ and substrate pairs $[A]$ & $[A_{pp}]$ are equal. Thus an asymmetric steady state is characterized by $[K] \neq [P]$ and $[A] \neq [A_{pp}]$. This allows us to characterize the asymmetric state and focus on that in the analysis.

Thus in this way, through a series of algebraic manipulations (to obtain reduced equations characterizing steady states, and asymmetric ones in particular), we isolate the necessary and sufficient conditions for the asymmetric state. However, only a direct steady-state determination is performed, and the stability of these states is not discussed in the analytical work presented here. The following notation is used hereon for the sake of brevity in analytical expressions:

$$c_i = \frac{k_{bi}}{k_i + k_{ubi}} \quad \text{and} \quad d_i = \frac{a_{bi}}{a_i + a_{ubi}}$$

## Ordered distributive DSP – common kinase and common phosphatase

The ordered distributive DSP with common kinase and common phosphatase acting on both modification sites is capable of exhibiting Case 1 symmetry. The kinetic constraints necessary to impose this symmetry are given in the schematic in *Figure 2A* ($k_3 = k_1$, $k_4 = k_2$, $k_{b3} = k_{b1}$, $k_{b4} = k_{b2}$, $k_{ub3} = k_{ub1}$, $k_{ub4} = k_{ub2}$ and $P_{Total} = K_{Total}$). Under these constraints, Case 1 symmetry is established, resulting in equal concentrations of the substrates $[A]$ and $[A_{pp}]$ and equal concentrations of the enzymes $[K]$ and $[P]$. Imposing these constraints and evaluating the steady states of the system (by successively solving for the steady states of the ODEs), we get the following equation involving the concentrations at steady state:

$$([K] - [P]) \left( \frac{[A_p](k_1 - k_2)c_2 + k_1}{k_1} \right) = 0 \tag{14}$$

From this expression, we can clearly see that the system accommodates asymmetric solutions (i.e. $[K] \neq [P]$) when the term $([A_p](k_1 - k_2)c_2 + k_1)$ is 0. Using this information, we identify features of the asymmetric steady state. We show that at a given asymmetric state the concentration of $[A_p]$ is fixed at a certain value (invariant), given only by key kinetic constants. Further, since the concentration of a substrate is always positive, this asymmetric state can only exist when $k_{21}$, which gives us a necessary condition for the symmetry to break. The sufficiency of this condition follows by evaluating all species concentrations at this invariant value of $A_p$ and proving that (i) they are positive when the necessary condition is satisfied, and (ii) that they satisfy the system of ODEs at steady state and the conservation conditions. Thereby, we show the presence of asymmetric states for some suitable value of $A_{Total}$ (which is determined from the resulting steady-state substrate concentrations). This is carried out in *Source code 1* (Section 2.1) and *Supplementary file 1*.

- *Necessary and sufficient conditions for symmetry breaking:*

$k_2 > k_1$

- *Invariant:*

$$[A_p] = \frac{1}{c_2}\left[\frac{k_1}{k_2 - k_1}\right]$$

## Ordered distributive DSP – separate kinase and separate phosphatase

The ordered distributive DSP with separate kinase and separate phosphatase acting on each modification site can present Case 1 symmetry similar to the case of ordered distributive DSP with common kinase and common phosphatase acting on both modification sites. The symmetry is established with similar kinetic constraints (though now involving different pairs of enzymes; $K1_{Total} = P2_{Total}$ and $K2_{Total} = P1_{Total}$). At symmetric steady states, this network can be shown to necessarily have the following symmetric enzyme pairing, $[K_1] = [P_2]$ and $[K_2] = [P_1]$, in addition to symmetry in the substrate pair $[A] = [A_{pp}]$. Through a simple algebraic analysis following solving for steady states, we can establish that an asymmetric state violating these symmetric pairings of variables can never exist, thus ruling out symmetry breaking in the model (see *Source code 1* (Maple file) for more details).

Specifically by successively solving for the steady states of the ODEs, we can ascertain that the following equation is always true at steady state for the model, indicating that an asymmetric steady state is infeasible.

$$[A] = [A_{pp}] = [A_p]\left(\frac{[P1]}{[P2]}\right)\left(\frac{c_2 k_2}{c_1 k_1}\right)$$

## Ordered distributive TSP – common kinase and common phosphatase

The ordered distributive TSP with common kinase common phosphatase is capable of exhibiting Case 1 symmetry with $[A] = [A_{ppp}]$ and $[A_p] = [A_{pp}]$. The kinetic constraints necessary to impose this symmetry are given in the schematic in *Appendix 2—figure 8*; $k_4 = k_1$, $k_5 = k_2$, $k_6 = k_3$, $k_{b4} = k_{b1}$, $k_{b5} = k_{b2}$, $k_{b6} = k_{b3}$, $k_{ub4} = k_{ub1}$, $k_{ub5} = k_{ub2}$, $k_{ub6} = k_{ub3}$ and $P_{Total} = K_{Total}$. Under these constraints, Case 1 symmetry is established with $[A] = [A_{ppp}]$, $[A_p] = [A_{pp}]$ along with $[K] = [P]$. Solving the resulting system of equations for steady states, we arrive at the following equation:

$$([K] - [P])\left(\frac{[A_p][[K]+[P]](k_1 - k_3)c_3 + k_1[P]}{[P]k_1}\right) = 0 \qquad (15)$$

Hence for an asymmetric solution ($[K] \neq [P]$) to exist, the second term has to be 0, and thus with this we find the features and necessary conditions of the asymmetric state. From this, we show that at an asymmetric steady state the sum of the concentration of the partially modified substrates ($[A_p] + [A_{pp}]$) is fixed at a certain value (invariant), given only by key kinetic constants. This is an example (seen elsewhere) of the sum of concentrations of two species fixed at steady state. Further, since the concentrations of substrates are strictly positive, we can show that the asymmetric state can only exist when $k_{31}$, which gives us the necessary conditions for the symmetry to break. The sufficiency of this condition follows by evaluating all species concentrations at values of $A_p$ and $A_{pp}$ obtained from this invariant and proving that (i) they are positive when the necessary condition is satisfied, and (ii) that they satisfy the system of ODEs at steady state and the conservation conditions. We show the presence of asymmetric states for some suitable value of $A_{Total}$. This is carried out in *Source code 1* (Section 2.3) and *Supplementary file 1*.

- *Necessary and sufficient conditions for symmetry breaking:*

$k_3 > k_1$

- *Invariant:*

$$[A_p] + [A_{pp}] = \frac{1}{c_3}\left[\frac{k_1}{k_3 - k_1}\right]$$

## Random system 1 – common kinase and common phosphatase

The random distributive DSP with common kinase and common phosphatase acting on both modification sites (system 1) is capable of Case 1, Case 2, and Case 3 symmetries. However, only Case 1 and Case 3 symmetries can be broken. Here we present analytical arguments elucidating the presence of Case 1 and Case 3 symmetry breaking and its associated features. We also present the analytical arguments precluding Case 2 symmetry breaking.

### Case 1 symmetry

Case 1 symmetry is established in the random DSP through the kinetic structure provided in ***Figure 1*** ($k_3 = k_1$, $k_4 = k_2$, $k_{b3} = k_{b1}$, $k_{b4} = k_{b2}$, $k_{ub3} = k_{ub1}$, $k_{ub4} = k_{ub2}$, $a_3 = a_1$, $a_4 = a_2$, $a_{b3} = a_{b1}$, $a_{b4} = a_{b2}$, $a_{ub3} = a_{ub1}$, and $a_{ub4} = a_{ub2}$). In addition, the following constraint on enzyme total amounts also needs to be satisfied for exact symmetry to be present: $K_{Total} = P_{Total}$. Superimposing these kinetic constraints results in Case 1 symmetry with $[A_{00}] = [A_{11}]$ along with equality of the free enzyme concentrations $[K] = [P]$. Similar to the analysis earlier on the ordered distributive DSP models, we solve the system for steady states in terms of fewer variables. This results in the following equation:

$$([K] - [P])\left(\frac{[A_{10}]((k_1 - k_2)c_1 - d_1 k_2)c_2 + c_1 k_1)a_2 + [A_{10}]c_2 d_1 k_2 a_1)}{(c_1 k_1 a_2)}\right) = 0 \tag{16}$$

Thus for an asymmetric state ($[K] \neq [P]$) to exist, the second term needs to be equal to 0. Setting this second term to zero reveals the features of the asymmetric steady state. Using this information, we can establish that the concentrations of the partially modified substrates in the asymmetric steady states are both fixed at a constant value (invariants), determined by a few key kinetic constants. Since the concentrations are always positive, we get the necessary conditions for symmetry breaking as shown below. The sufficiency of this condition follows by evaluating all species concentrations at this invariant value of $A_{01}$ and $A_{10}$ (inferring the other species concentrations) and proving that (i) they are positive when the necessary condition is satisfied, and (ii) that they satisfy the system of ODEs at steady state and the conservation conditions. From this, we can deduce that asymmetric states exist beyond a critical value of $A_{Total}$. This is carried out in ***Source code 1*** (Section 3.1) and ***Supplementary file 1***.

- *Necessary and sufficient conditions for symmetry breaking:*

$$c_1 a_2(k_2 - k_1) + d_1 k_2(a_2 - a_1) > 0$$

- *Invariants:*

$$[A_{01}] = \frac{-d_1 k_2 a_1}{d_2((d_1(a_1 - a_2) - c_1 a_2)k_2 + c_1 k_1 a_2)}$$

$$[A_{10}] = \frac{-c_1 k_1 a_2}{c_2(((k_1 - k_2)c_1 - d_1 k_2)a_2 + d_1 k_2 a_1)}$$

### Case 2 symmetry

Case 2 symmetry is likewise established in the random DSP through the kinetic structure provided in ***Figure 1*** with constraints on pairs of parameters ($a_1 = k_1$, $a_2 = k_2$, $a_{b1} = k_{b1}$, $a_{b2} = k_{b2}$, $a_{ub1} = k_{ub1}$, $a_{ub2} = k_{ub2}$, $a_3 = k_3$, $a_4 = k_4$, $a_{b3} = k_{b3}$, $a_{b4} = k_{b4}$, $a_{ub3} = k_{ub3}$, and $a_{ub4} = k_{ub4}$). However unlike other symmetries, it needs no additional constraint on total enzyme amounts for exact symmetry to be present. Under these conditions, Case 2 symmetry is established with $[A_{01}] = [A_{10}]$. Hence should symmetry break and an asymmetric steady state exist, it would be characterized by $[A_{01}] \neq [A_{10}]$. Solving for the steady states of these substrates in terms of the free enzyme concentrations and the fully unmodified substrate, we can ascertain that the concentrations of $[A_{01}]$ will always equal $[A_{10}]$, and thus that no asymmetric state can exist (see ***Source code 1*** for more details). In particular, we can see that for any given steady state, $[A_{10}] = [A_{01}] = [A_{00}]\left(\frac{[K]}{[P]}\right)\left(\frac{c_1 k_1}{c_4 k_4}\right)$, making any asymmetric steady state impossible.

### Case 3

Case 3 symmetry is established in the random DSP through the kinetic structure provided in ***Figure 1*** with constraints $a_1 = k_3$, $a_2 = k_4$, $a_{b1} = k_{b3}$, $a_{b2} = k_{b4}$, $a_{ub1} = k_{ub3}$, $a_{ub2} = k_{ub4}$, $a_3 = k_1$, $a_4 = k_2$, $a_{b3} = k_{b1}$, $a_{b4} = k_{b2}$, $a_{ub3} = k_{ub1}$, and $a_{ub4} = k_{ub2}$. In addition, the following constraint on total enzyme

concentrations needs to be satisfied for exact symmetry, $K_{Total} = P_{Total}$. Under these conditions, Case 3 symmetry is established with $[A_{00}] = [A_{11}]$ and $[A_{01}] = [A_{10}]$ along with equality of the free enzyme concentrations $[K] = [P]$. Imposing these conditions and solving for the steady states of the system, we get the following correlation involving concentrations of the variables:

$$\begin{aligned}
(\epsilon - 1)(c_1 c_2^2 \epsilon k_1 k_2^2 + (c_1 k_1 + c_3 k_3)c_4(((\epsilon^2 - [A_{11}](c_1 + c_3)\epsilon - [A_{11}] \\
(c_1 + c_3))k_4 + [A_{11}]c_1 k_1(\epsilon + 1))k_2 + k_4[A_{11}]c_3 k_3(\epsilon + 1))c_2 + c_3 c_4^2 \epsilon k_3 k_4^2) \quad &= \quad 0
\end{aligned} \tag{17}$$

where $\epsilon = \frac{[K]}{[P]}$. Thus for an asymmetric state to exist ($[K] \neq [P]$, or $\epsilon \neq 1$), the second term needs to be equal to 0. Using this information to further solve for the steady states of the ODEs reveals features of the asymmetric steady states. We find that the sum of the concentrations of the partially modified substrates is fixed at a constant value (invariant), determined by a few key kinetic constants. Since the concentrations are always positive, we also get the necessary conditions for symmetry breaking as shown below. The sufficiency of this condition follows by evaluating all species concentrations at this invariant value of $A_{01} + A_{10}$ and proving that (i) they are positive when the necessary condition is satisfied, and (ii) that they satisfy the system of ODEs at steady state and the conservation conditions. This demonstrates that there exist values of $A_{Total}$ beyond which asymmetric steady states exist. This is carried out in *Source code 1* (Section 3.1) and *Supplementary file 1*.

- *Necessary and sufficient conditions for symmetry breaking:*

$c_3 k_4(k_2 - k_3) - c_1 k_2(k_1 - k_4) > 0$

- *Invariant:*

$[A_{01}] + [A_{10}] = \frac{-c_1 c_2 k_1 k_2 - c_3 c_4 k_3 k_4}{(((-c_1 - c_3)k_4 + c_1 k_1)k_2 + c_3 k_3 k_4)c_4 c_2}$

## Random system 2 – separate kinase and common phosphatase

The random distributive DSP with separate kinase acting on each modification site and a common phosphatase effecting dephosphorylation (system 2) can only permit Case 2 symmetry ($[A_{01}] = [A_{10}]$). This is due to the nature of independent enzymes effecting phosphorylation on each modification site while a common enzyme is responsible for dephosphorylation, which precludes Case 1 and Case 3 symmetry. In this subsection, we present the analytical arguments precluding Case 2 symmetry breaking in the network.

### Case 2

Case 2 symmetry is established in the random ordered distributive DSP network through the kinetic structure provided in *Figure 1* with constraints $a_1 = k_1$, $a_2 = k_2$, $a_{b1} = k_{b1}$, $a_{b2} = k_{b2}$, $a_{ub1} = k_{ub1}$, $a_{ub2} = k_{ub2}$, $a_3 = k_3$, $a_4 = k_4$, $a_{b3} = k_{b3}$, $a_{b4} = k_{b4}$, $a_{ub3} = k_{ub3}$, and $a_{ub4} = k_{ub4}$. In addition, the following constraint on total enzyme amounts needs to be satisfied for exact symmetry: $K1_{Total} = K2_{Total}$. Under these kinetic constraints, Case 2 symmetry is established with $[A_{01}] = [A_{10}]$. Imposing these constraints and solving for the steady states of these substrates, we get the following correlation representing permissible steady states:

$$([K_1] - [K_2]) \left( \frac{[A_{00}]k_{b1} + k_1 + k_{ub1}}{k_1 + k_{ub1}} \right) \quad = \quad 0 \tag{18}$$

Since the concentrations of the substrates and the kinetic constants are always positive, we can ascertain that the only steady state permitted by the system is when $[K_1] = [K_2]$, or $[A_{01}] = [A_{10}]$. Thus Case 2 symmetry breaking is not possible in random system 2 with separate kinase and common phosphatase. See *Source code 1* (Section 3.2) and *Supplementary file 1*.

## Random system 3 – separate kinase and separate phosphatase

The random distributive DSP with separate kinase and separate phosphatase effecting modifications in each modification site (system 3) is capable of Case 1, Case 2, and Case 3 symmetries, and each of these symmetries is capable of breaking. Here we derive the necessary and sufficient conditions for symmetry breaking, and the features of the symmetry-broken state in that process.

## Case 1 symmetry

Case 1 symmetry is established in the random DSP through the kinetic structure provided in **Figure 1** with constraints $k_3 = k_1$, $k_4 = k_2$, $k_{b3} = k_{b1}$, $k_{b4} = k_{b2}$, $k_{ub3} = k_{ub1}$, $k_{ub4} = k_{ub2}$, $a_3 = a_1$, $a_4 = a_2$, $a_{b3} = a_{b1}$, $a_{b4} = a_{b2}$, $a_{ub3} = a_{ub1}$, and $a_{ub4} = a_{ub2}$. In addition, the following constraint on total enzyme concentrations needs to be satisfied for exact symmetry, $K1_{Total} = P2_{Total}$ and $K2_{Total} = P1_{Total}$. Under these conditions, Case 1 symmetry is established with $[A_{00}] = [A_{11}]$ along with equality of the free enzyme concentrations $[K_1]$ and $[P_2]$ and $[K_2]$ and $[P_1]$. Imposing these kinetic constraints and evaluating the steady states of the system (by successively solving the steady states of ODEs for variables in terms of each other), we can get the following equation representing permissible steady states of the model:

$$(\epsilon - 1)\frac{[K1]^2\epsilon a_2 c_1([A_{01}]c_2+1)k_1^2+k_2[P1]^2[A_{01}]a_1c_2d_1([A_{01}]c_2+1)k_1-[A_{01}]^2[P1]^2a_2c_2^2d_1k_2^2}{([K1]c_1\epsilon k_1^2 a_2)} = 0 \qquad (19)$$

where $\epsilon = \frac{[A_{00}]}{[A_{11}]}$. Thus for asymmetric solutions ($\epsilon \neq 1$) to exist, the second term in the expression needs to be equal to zero. Using this information and rearranging the terms, we can establish the following correlation between concentrations of the partially modified substrates $[A_{01}]$ and $[A_{10}]$:

$$\lambda_{[A_{01}]}\lambda_{[A_{10}]} = 1$$

where

$$\lambda_{[A_{01}]} = \frac{k_2 c_2 [A_{01}]}{k_1(c_2[A_{01}]+1)} \qquad \text{and} \qquad \lambda_{[A_{10}]} = \frac{a_2 d_2 [A_{10}]}{a_1(d_2[A_{10}]+1)} \qquad (20)$$

This correlation represents asymmetric steady-state solutions to the system of ODEs (for additional details, see Source Code 1 [Section 3.3] and **Supplementary file 1**). Simultaneously, by solving the steady state of the system, using the total individual enzyme conservation equations we get an alternative correlation between the partial substrate concentrations $[A_{01}]$ and $[A_{10}]$ as shown below, which is valid for all steady states:

$$\frac{P2_{Total}}{P1_{Total}} = \left[\frac{D\alpha\lambda_{[A_{01}]}+1}{D\alpha+\lambda_{[A_{10}]}}\right] \qquad (21)$$

where

$$D = \frac{c_2[A_{01}]+1}{d_2[A_{10}]+1} \qquad \text{and} \qquad \alpha = \frac{[K2]+[P1]}{[K1]+[P2]} \qquad (22)$$

Using the above correlations between concentrations of $[A_{00}]$ and $[A_{11}]$ (eq. (20) representing the asymmetric solutions and eq. (21) representing all feasible solutions) we can ascertain additional features of the asymmetric state. We find that in a symmetry-broken state the concentrations of the partially modified forms ($[A_{01}]$ and $[A_{10}]$) are fixed at constant values determined by key kinetic constants and total enzyme concentrations. Since the concentrations cannot be negative, we can also thus isolate the necessary constraints for symmetry breaking in this network as shown below. The sufficiency of this condition follows by evaluating all species concentrations based on the invariant values of $A_{01}$ and $A_{10}$, and proving that (i) they are positive when the necessary condition is satisfied, and (ii) that they satisfy the system of ODEs at steady state and the conservation conditions. This indicates that there are values of $A_{Total}$ beyond which asymmetric steady states are guaranteed to exist. This is carried out in **Source code 1** (Section 3.3) and **Supplementary file 1**.

- *Necessary and sufficient conditions for symmetry breaking:*

$$k_2 P1_{Total} > k_1 P2_{Total}$$

$$a_2 P2_{Total} > a_1 P1_{Total}$$

- *Invariants:*

$$[A_{01}] = \frac{-k_1 P2_{Total}}{c_2(k_1 P2_{Total}-k_2 P1_{Total})}$$

$$[A_{10}] = \frac{-a_1 P1_{Total}}{d_2(a_1 P1_{Total}-a_2 P2_{Total})}$$

## Case 2 symmetry

Case 2 symmetry is likewise established in the random DSP through the kinetic structure provided in **Figure 1** with constraints $a_1 = k_1$, $a_2 = k_2$, $a_{b1} = k_{b1}$, $a_{b2} = k_{b2}$, $a_{ub1} = k_{ub1}$, $a_{ub2} = k_{ub2}$, $a_3 = k_3$, $a_4 = k_4$, $a_{b3} = k_{b3}$, $a_{b4} = k_{b4}$, $a_{ub3} = k_{ub3}$, and $a_{ub4} = k_{ub4}$. In addition, the following constraints on total enzyme

concentrations are also needed for exact symmetry, $K1_{Total} = K2_{Total} = K_{Total}$ and $P2_{Total} = P1_{Total} = P_{Total}$. Under these conditions, Case 2 symmetry is established with $[A_{01}] = [A_{10}]$ along with equality of the free enzyme concentrations $[K_1] = [K_2]$ and $[P_1] = [P_2]$. Imposing these kinetic constraints and evaluating the steady states of the system, we can get the following equation representing permissible steady states of the model:

$$(\epsilon - 1)\frac{(-k_3\epsilon[P1]^2 c_4([A_{00}]c_1+1)k_4^2 - [K1]^2[A_{00}]c_1 c_2 k_1 k_2([A_{00}]c_1+1)k_4 + [A_{00}]^2[K1]^2 c_1^2 c_2 k_1^2 k_3)[K2]}{([P1]c_4 k_4^2 k_3 \epsilon)} = 0 \qquad (23)$$

where $\epsilon = \frac{[A_{00}]}{[A_{11}]}$. Thus for asymmetric solutions ($\epsilon \neq 1$) to exist, the second term in the expression needs to be equal to zero. Using this information and rearranging the terms, we can establish the following correlation between concentrations of the fully modified and fully unmodified substrates $[A_{00}]$ and $[A_{11}]$:

$$\lambda_{[A_{00}]}\lambda_{[A_{11}]} = 1$$

where

$$\lambda_{[A_{00}]} = \frac{k_1 c_1 [A_{00}]}{k_4(c_1[A_{00}]+1)} \qquad \text{and} \qquad \lambda_{[A_{11}]} = \frac{k_3 c_3 [A_{11}]}{k_2(c_3[A_{11}]+1)} \qquad (24)$$

This correlation represents a requirement for asymmetric steady-state solutions to the system of ODEs. Simultaneously by solving the steady state of the system, using the total individual enzyme conservation equations, we get an alternative correlation between the partial substrate concentrations as shown below:

$$\frac{P_{Total}}{K_{Total}} = \alpha D\left[\frac{D\alpha\lambda_{[A_{00}]}+1}{D\alpha+\lambda_{[A_{11}]}}\right] \qquad (25)$$

where

$$D = \frac{c_1[A_{00}]+1}{c_3[A_{11}]+1} \qquad \text{and} \qquad \alpha = \frac{[K2]+[K1]}{[P1]+[P2]} \qquad (26)$$

Using the above correlations between concentrations of $[A_{00}]$ and $[A_{11}]$ together (eq. (24) representing the asymmetric solutions and eq. (25) representing all feasible solutions), we can ascertain additional features of the asymmetric state. We find that in a symmetry-broken state the concentrations of the completely modified and unmodified substrate forms ($[A_{00}]$ and $[A_{11}]$) are fixed at constant concentrations (invariant) given by key kinetic constants and total enzyme concentrations. Since the concentrations cannot be negative, we can also thus isolate the necessary constraints for symmetry breaking in this network as shown below. The sufficiency of this condition follows by evaluating all species concentrations at this invariant value of $A_{01}$ and $A_{10}$ and proving that (i) they are positive when the necessary condition is satisfied, and (ii) that they satisfy the system of ODEs at steady state and the conservation conditions. This then indicates that there exist finite positive values of $A_{Total}$ for which asymmetric states exist. This is carried out in **Source code 1** (Section 3.3) and **Supplementary file 1**.

- *Necessary and sufficient conditions for symmetry breaking:*

$k_1 K_{Total} > k_4 P_{Total}$

$k_3 P_{Total} > k_2 K_{Total}$

- *Invariants:*

$[A_{00}] = \frac{k_4 P_{Total}}{c_1(k_1 K_{Total} - k_4 P_{Total})}$

$[A_{11}] = \frac{-k_2 K_{Total}}{c_3(k_2 K_{Total} - k_3 P_{Total})}$

## Case 3

Case 3 symmetry is established in the random DSP through the kinetic structure provided in **Figure 1** with constraints $a_1 = k_3$, $a_2 = k_4$, $a_{b1} = k_{b3}$, $a_{b2} = k_{b4}$, $a_{ub1} = k_{ub3}$, $a_{ub2} = k_{ub4}$, $a_3 = k_1$, $a_4 = k_2$, $a_{b3} = k_{b1}$, $a_{b4} = k_{b2}$, $a_{ub3} = k_{ub1}$, and $a_{ub4} = k_{ub2}$. In addition, the following constraints on total enzyme concentrations are also needed for exact symmetry, $K1_{Total} = P1_{Total}$ and $K2_{Total} = P2_{Total}$. Under these conditions, Case 3 symmetry is established with $[A_{01}] = [A_{10}]$ and $[A_{00}] = [A_{11}]$, along with equality of the free enzyme concentrations $[K_1] = [P_1]$ and $[P_2] = [K_2]$.

This symmetry can indeed break as shown computationally in the main text (*Figure 3* and *Appendix 2—figure 4*). Similar to the approach used earlier, in order to obtain the necessary and sufficient conditions for symmetry breaking, we solve for the steady states of the system of ODEs successively to obtain steady-state correlations of variables in terms of each other. In so doing and by isolating asymmetric solutions of the type $[A_{00}] \neq [A_{11}]$ and $[A_{01}] \neq [A_{10}]$, we obtain the necessary conditions (see below) for symmetry breaking (analysis not shown here; please refer to *Source code 1* [Section 3.3] and *Supplementary file 1*).

However unlike other classes of symmetries, Case 3 symmetry breaking in this model is not associated with a simple linear invariant in terms of concentrations of species. Moreover, the symmetry breaking in this instance can be of qualitatively different types, (i) where the asymmetry in the symmetry-broken states is more pronounced in the partial phosphoforms $A_{01}$ and $A_{10}$ or (ii) where the asymmetry is more pronounced in the fully modified/unmodified forms $A_{00}$ and $A_{11}$.

These come with contrasting qualitative implications with regard to the symmetry-broken steady states. As shown earlier in *Figure 3B* (panel 2) and *Appendix 2—figure 4*, depending on the nature of symmetry breaking outlined above, either the sum of the partially modified substrates, or the sum of the completely modified and unmodified substrates, can exhibit an approximate robustness in concentration relative to the other pair. This qualitative difference can be traced to the underlying kinetics. In particular, in our analytical work (*Source code 1* [Section 3.3] and *Supplementary file 1*), we have shown that the choice of kinetics can dictate the nature of symmetry breaking.

This approximate concentration robustness exhibited by either pair of substrates can be traced to the fact that in an asymmetric steady state one pair of substrate concentrations ($[A_{01}]$ and $[A_{10}]$ or $[A_{00}]$ and $[A_{11}]$) is bounded and asymptotically reaches a certain value as the total substrate concentration approaches infinity, while the other pair can vary in an unbounded manner with varying total substrate concentration. This is summarized below.

- *Necessary and sufficient conditions for symmetry breaking*

$k_1 > k_4$ and $k_3 > k_2$

OR

$k_4 > k_1$ and $k_2 > k_3$

- *Features of asymmetric steady states*

IF $k_1 > k_4$ and $k_3 > k_2$

Concentrations of $[A_{00}]$ and $[A_{11}]$ exhibit approximate robustness.

Asymptotic concentration of $[A_{00}]$ at infinite $A_{Total} = \frac{k_4}{c_1(k_1-k_4)}$ or $\frac{k_2}{c_3(k_3-k_2)}$

Asymptotic concentration of $[A_{11}]$ at infinite $A_{Total} = \frac{k_2}{c_3(k_3-k_2)}$ or $\frac{k_4}{c_1(k_1-k_4)}$

IF $k_4 > k_1$ and $k_2 > k_3$

Concentration of $[A_{01}]$ and $[A_{10}]$ exhibit approximate robustness.

Asymptotic concentration of $[A_{01}]$ at infinite $A_{Total} = \frac{k_3}{c_2(k_2-k_3)}$ or $\frac{k_1}{c_4(k_4-k_1)}$

Asymptotic concentration of $[A_{10}]$ at infinite $A_{Total} = \frac{k_1}{c_4(k_4-k_1)}$ or $\frac{k_3}{c_2(k_2-k_3)}$

These asymptotes obtained analytically have been cross-validated with bifurcation analysis, and the specific cross-validation for the figures in this paper are provided in the file (Read_Me.mw of *Source code 1* and *Supplementary file 1*).

## Mixed random 1 – common kinase and common phosphatase

The mixed random ordered DSP with a common kinase effecting distributive phosphorylation and a common phosphatase effecting processive dephosphorylation is capable of Case 2 symmetry. However, this symmetry cannot be broken. In this subsection, we present the analytical arguments precluding Case 2 symmetry breaking in this network.

### Case 2 symmetry

Case 2 symmetry is established in the mixed random DSP through the kinetic structure provided in *Figure 2E* with constraints $a_1 = k_1$, $a_{b1} = k_{b1}$, $a_{ub1} = k_{ub1}$, $a_2 = k_2$, $a_{b2} = k_{b2}$, $a_{ub2} = k_{ub2}$, $a_3 = k_3$, $a_{b3} = k_{b3}$

, $a_{ub3} = k_{ub3}$ and $k_4 = a_4$. Under these conditions, Case 2 symmetry is established with $[A_{01}] = [A_{10}]$. Imposing these constraints and solving for the steady states of these substrates in terms of the free enzyme concentrations and the fully unmodified substrate, we can ascertain that $[A_{01}]$ always equals $[A_{10}]$. In particular, $[A_{01}] = [A_{10}] = [A_{00}] \left( \frac{c_1 k_1}{c_2 k_2} \right)$ is always true at steady state, making any asymmetric steady state infeasible (see *Source code 1* [Section 4.1] and *Supplementary file 1* for more details).

## Mixed random 2 – separate kinase and common phosphatase

The mixed random DSP with separate kinases acting on each site effecting distributive phosphorylation and a common phosphatase effecting processive dephosphorylation is capable of Case 2 symmetry, and it is possible for this symmetry to break. In this subsection, we show the necessary conditions for symmetry breaking and features of the asymmetric state.

### Case 2

Case 2 symmetry is established in the mixed random DSP through the kinetic structure provided in *Figure 2E* with constraints $a_1 = k_1$, $a_{b1} = k_{b1}$, $a_{ub1} = k_{ub1}$, $a_2 = k_2$, $a_{b2} = k_{b2}$, $a_{ub2} = k_{ub2}$, $a_3 = k_3$, $a_{b3} = k_{b3}$ and $a_{ub3} = k_{ub3}$ and $k_4 = a_4$. In addition, the following constraints on total enzyme concentrations need to be satisfied for exact symmetry: $K1_{Total} = K2_{Total}$. Under these conditions, Case 2 symmetry is established with $[A_{01}] = [A_{10}]$ along with equality of the free enzyme concentrations $[K_1] = [K_2]$. Imposing these kinetic constraints and evaluating the steady states of the system, we get the following equation representing permissible steady states of the model:

$$(\epsilon - 1)(-k_1 \epsilon + [A_{01}] c_2 (k_1 - k_2)) \quad = \quad 0 \tag{27}$$

where $\epsilon = \frac{[K_1]}{[K_2]}$. Thus we can see that the system accommodates an asymmetric steady state where $\epsilon \neq 1$, provided the second term in the equation is 0. From this, we can ascertain the features of asymmetric state by isolating correlations for $A_{01}$ from the above equation and using it to simplify the steady states of system of ODEs. We find that in a symmetry-broken state the concentrations of the completely modified and unmodified substrate forms ($[A_{00}]$ and $[A_{11}]$) are fixed at constant values (invariant) given by key kinetic constants and total enzyme concentrations. Since the concentrations cannot be negative, we can also thus isolate the necessary constraints for symmetry breaking in this network as shown below. The sufficiency of this condition follows by evaluating all species concentrations at this invariant value of $A_{00}$ and $A_{11}$ and proving that (i) they are positive when the necessary condition is satisfied, and (ii) that they satisfy the system of ODEs at steady state and the conservation conditions. This ensures that there exist values of $A_{Total}$ for which asymmetric states exist. This is carried out in *Source code 1* (Section 4.2) and *Supplementary file 1*.

- *Necessary and sufficient conditions for symmetry breaking:*

$k_2 < k_1$

$k_2 K1_{Total} (k_3 + k_4) < P_{Total} k_3 k_4$

- *Invariant(s):*

$[A_{00}] = \frac{1}{c_1} \left[ \frac{k_2}{(k_1 - k_2)} \right]$

$[A_{11}] = \frac{-k_2 k_4 K1_{Total}}{(2(k_2 K1_{Total}(k_3 + k_4) - P_{Total} k_3 k_4)) c_3}$

In the limit where dephosphorylation is acting in the unsaturated limit (mixed random 2A), as described in the models section earlier, the dephosphorylation is replaced by a single linear reaction from $A_{11}$ to $A_{00}$ (with a rate constant which is denoted by $k_3$ which implicitly contains the effect of the total phosphatase concentration: in fact, it corresponds to $k_{b3} P_{Total}$ in the previous model). In this case, the symmetry established can still break. The necessary and sufficient conditions, and the features of the asymmetric state, are as follows (the analysis is the same as above; see *Source code 1* [Section 4.3] and *Supplementary file 1* for more details):

- *Necessary and sufficient condition for symmetry breaking:*

$k_2 < k_1$

- *Invariant(s):*

$[A_{00}] = \frac{1}{c_1} \left[ \frac{k_2}{k_1 - k_2} \right]$

$[A_{11}] = K1_{Total} \left[ \frac{k_2}{2 k_3} \right]$

## Sufficiency of necessary conditions

We have established necessary conditions for the presence of symmetry breaking in various classes of symmetries and networks. We have also further shown that these necessary conditions are also sufficient for symmetry breaking to occur at some positive total substrate concentration. The sufficiency argument has been briefly discussed above at the appropriate sections, and the approach is consolidated below. For more details, please refer to *Source code 1* and *Supplementary file 1*. The feasibility of any steady state for the models described above relies on the concentrations of variables at steady state satisfying three separate constraints.

1. Variable values satisfy ODE description of the model (for a given set of kinetic parameters).
2. Conservation equations (for the enzymes and total substrate concentrations) should be satisfied.
3. All variable values (concentrations) must be positive.

We obtained the necessary conditions for symmetry breaking by suitably leveraging points 1–3. By solving for the steady states of the system of ODEs, we obtained correlations between concentrations of various variables in terms of kinetic parameters and total enzyme concentrations. Further, by isolating asymmetric solutions, we established the features of the asymmetric state. Then by requiring that the concentrations of key substrates (invariants) be positive, we obtained the necessary conditions.

We extended this argument to ensure sufficiency of these conditions by evaluating concentrations of all variables in the asymmetric state and showing that they are indeed positive if the necessary conditions (in terms of kinetic constants and total enzyme concentrations) are satisfied for some total substrate concentration (note here that the bifurcation diagram is along $A_{Total}$). We also verify that in each case the total enzyme conservations are satisfied. In this manner, we have shown that the necessary conditions are indeed sufficient for symmetry breaking to occur for some total enzyme and substrate concentration.

In addition, the analysis provided in *Source code 1* (and *Supplementary file 1*) also evaluates and predicts the position of symmetry breaking along $A_{Total}$ (not shown here in Appendix 2). Note that the intersection of the symmetric steady state and the asymmetric steady state denotes a pitchfork bifurcation and thus the position of symmetry breaking. In each case, along with the invariants we also evaluate the total substrate concentration at which the pitchfork bifurcation occurs by imposing features of symmetry in terms of concentration of various variables in the asymmetric steady-state invariants and correlations. This is presented in *Source code 1* (Maple file) and has been cross-validated with bifurcation analysis for all plots provided in this paper. This prediction and cross-validation can be found in (Read_Me.mw) and *Supplementary file 1*.

## Origins of ACR

In each instance of symmetry breaking encountered in the above models, we observe that either the concentration of specific individual substrates or sum of concentrations of specific substrates exhibits strict ACR in the asymmetric branches (with Case 3 symmetry breaking in the separate kinase separate phosphatase network being an exception, where approximate concentration robustness is exhibited by sums of concentrations of pairs of substrates). Overall, following symmetry breaking in these networks, the asymmetric steady states are characterized by the concentration of the specific substrates (or sums thereof) being (exactly) fixed, and in fact this remains so indefinitely along increasing $A_{Total}$ concentrations.

To understand the origins of ACR and its dependence on symmetry and symmetry breaking, we use the ordered DSP model as a basis of exploration.

Note that ACR in this instance is defined to mean the concentration of some substrate form being exactly maintained at some fixed concentration (along a steady-state branch) for a range of total concentrations of either the substrate or the enzymes. In this definition, we make no assumptions on the kinetic regime of modification/demodification of substrates. Note that it is possible for the concentration of a substrate species to be approximately constant (something encountered in limiting regimes of enzymatic action), and this will be described subsequently.

We begin by analysing the ordered distributive DSP network with common kinase and common phosphatase (refer *Figure 1* for the schematic) in the absence of any symmetry in either the kinetic parameters or the total concentrations of enzymes/substrate. This network has three substrate forms, $A$, $A_p$ and $A_{pp}$ that can potentially exhibit ACR. Note that earlier in our analysis of Case 1

symmetry breaking in the model, we observed $A_p$ exhibiting ACR with increasing concentration of $A_{Total}$, in the asymmetric branches following symmetry breaking.

The analysis carried out here is structured to answer four key questions regarding the phenomenon of ACR.

- Which substrates in this network are capable of exhibiting ACR?
- Is ACR (where possible) only exhibited with respect to the (change of) total substrate concentration? Is ACR possible with changing total enzyme concentration?
- Are there any additional constraints on the kinetic parameters required to observe ACR (where possible)?
- What associated features (if any) are exhibited when ACR is observed in this network?

The mathematical analysis is described in detail in the attached Maple document (Section 5.1). The results are summarized here following which the key analytical arguments used are briefly presented.

- Our analysis reveals that ACR can only be exhibited by the partially modified substrate form ($A_p$)
- ACR in $A_p$ is only possible with changing concentration of total substrate.
- Further analysis of the ACR in $A_p$ reveals necessary (and sufficient) constraints on the kinetics and total concentrations of enzymes. $\frac{k_3 P_{Total}}{c_2 \left(k_2 K_{Total} - k_3 P_{Total}\right)} = \frac{k_1 K_{Total}}{c_4 \left(k_4 P_{Total} - k_1 K_{Total}\right)} > 0$
- Our analysis was able to further characterize associated features of this network when $A_p$ exhibits ACR. In particular,

If this system exhibits ACR in $A_p$ for a range of total substrate concentration, it is necessarily multistable within that range with two steady states exhibiting ACR.

There necessarily exists another steady-state branch for all positive total substrate concentrations. This branch intersects one of the two ACR branches which at some $A_{Total}$ value (computationally found to be a transcritical bifurcation, refer *Appendix 2—figure 9*)

This non-ACR branch state is characterized by a fixed ratio of free (unbound) kinase to free (unbound) phosphatase concentration for all positive total substrate concentrations, given by

$$\frac{[K]}{[P]} = \frac{K_{Total}}{P_{Total}}$$

This analysis allows us to summarize a number of key insights, specifically in the context of symmetry and symmetry breaking.

Complemented by *Appendix 2—figure 9*, our analysis reveals that a strict Case 1 symmetry is not a prerequisite for encountering ACR in the network. However, there is a kinetic requirement (which is sufficient) for the network to exhibit ACR (mathematically exact) (see above and in eq. (31)). It is noteworthy to observe that this requirement is satisfied trivially by an assumption of Case 1 symmetry in the network, but is a weaker condition. Similarly, the non-ACR branch of steady states becomes the symmetric branch with the assumption of Case 1 symmetry, and the transcritical bifurcation (at the intersection of the non-ACR branch and an ACR branch) as seen in *Appendix 2—figure 9* becomes a pitchfork bifurcation.

Thus while Case 1 symmetry is not a strict prerequisite for obtaining ACR in the model, it is a suitable vantage point to analyse the phenomenon, additionally serving to reduce the parametric complexity of the network and highlighting multisite-specific characteristics such as directionality of modifications.

In the following subsection, we provide the key insight used (refer to attached Maple document; *Source code 1*, Section 5.1) to obtain proofs for the above conclusions. We also provide a sketch of the proof establishing the presence of ACR in $A_p$ with increasing (changing) $A_{Total}$ here.

## Approach used for determining absence or presence of ACR in the ordered DSP with common kinase and common phosphatase

In the absence of any imposed symmetry, the DSP model is a set of nine ODEs involving 15 parameters, 9 variables, and 3 conservation conditions. However at steady state, the ODEs reduce to a system of nonlinear equations for the variable concentrations. The solution of these equations in sequence allows for eliminating many of the variables (by writing them in terms of a smaller set of variables). This systematic algebraic reduction of the system of equations allows for the system at steady state to be represented as a set of two coupled polynomial equations in two variables. In each proof, this reduction is in terms of the following two variables, namely, variable of the species being investigated for exhibiting the ACR and the variable, which is the ratio of free kinase to free phosphatase concentrations (denoted by $\varepsilon$). Note that the (feasible) solutions of these coupled polynomials determine the steady-state concentrations as the concentration of all other variables of the system can be obtained as functions of these variables.

As a consequence of ACR in the variable of interest, with changing total concentration (of substrate/enzyme), both equations need to be satisfied with only $\varepsilon$ being allowed to vary. Thus the resulting two polynomials should accommodate a common root for $\varepsilon$ for changing total amounts of either substrate/the enzymes (as pertinent to the proof) if ACR is to be exhibited in the substrate form of interest. This insight allows us to rule out ACR by contradiction in cases where the resulting polynomials do not provide this flexibility. Further in the case of ACR in $A_p$ with total amount of substrate, it allows us to elucidate the necessary and sufficient conditions for obtaining ACR in the network.

In the next section, we show exactly how this is pursued by providing a sketch of the proof for the presence of ACR in $A_p$ with changing total amounts of substrate.

## Proof: partially modified substrate exhibits ACR with changing total substrate concentration

As mentioned earlier, at steady the DSP model can be simplified as a set of coupled polynomials in two variables; namely, the substrate form being investigated for exhibiting ACR and the ratio of free (unbound) kinase and the free (unbound) phosphatase. In the context of this specific proof, the two variables are $A_p$ and $\varepsilon$. This results in the following two coupled polynomials whose feasible solutions determine the steady-state concentrations of the system:

$$0 = \left[A_p c_2 (k_2 K_{Total} - k_3 P_{Total}) - k_3 P_{Total}\right] k_1 \epsilon + \left[A_p c_4 (k_1 K_{Total} - k_4 P_{Total}) + k_1 K_{Total}\right] k_3 \tag{28}$$

$$
\begin{aligned}
0 = &\left[-(c_3(c_1 \epsilon * k_1 + c_4 k_4)k_3 + c_1 c_2 \epsilon^2 k_1 k_2)(c_2 \epsilon k_2 + c_4 k_3)\right] A_p^2 + \\
&\left[k_3(c_2 k_1 c_1((-P_{Total}k_3 + k_2(A_{Total} - P_{Total}))c_3 - k_2)\epsilon^2 + k_3 c_1 c_3((c_4 A_{Total} - c_4 P_{Total} - 1)k_1 - P_{Total}c_4 k_4)\epsilon - c_3\ldots\right. \\
&c_1 c_3 \epsilon k_1 k_3^2 A_{Total}
\end{aligned}
\tag{29}
$$

where the concentrations of the other variables can be obtained from these as shown below:

$$
\begin{aligned}
[A] &= \left[\frac{A_p}{\epsilon}\right]\left[\frac{c_4 k_4}{c_1 k_1}\right] & [A_{pp}] &= [A_p][\epsilon]\left[\frac{c_2 k_2}{c_3 k_3}\right] \\
[AK] &= \frac{k_3 c_4 k_4 P_{Total}[A_p]}{k_1([\epsilon][A_p]c_2 k_2 + k_3 c_4[A_p] + k_3)} & [A_p K] &= \frac{\epsilon P_{Total}k_3 c_2[A_p]}{k_3 + (c_2 k_2 \epsilon + c_4 k_3)[A_p]} \\
[A_{pp}P] &= \frac{k_2 c_2 P_{Total}[A_p][\epsilon]}{k_3 + k_3 c_4[A_p] + k_2 c_2[A_p][\epsilon]} & [A_p P] &= \frac{k_3 c_4 P_{Total}[A_p]}{k_3 + (k_3 c_4 + k_2 c_2)[A_p]} \\
[P] &= \frac{P_{Total}k_3}{k_3 + k_3 c_4[A_p] + c_2 k_2[A_p][\epsilon]}
\end{aligned}
\tag{30}
$$

Now assuming $A_p$ exhibits ACR for changing $A_{Total}$, both equations need to be satisfied with only epsilon allowed to vary.

The two polynomials have the following structure. (i) The polynomial in eq. (29) is a cubic polynomial in $\varepsilon$ whose coefficients include the parameter $A_{Total}$ (and $[A_p]$). (ii) The polynomial in eq. (28) is a linear expression in $\varepsilon$ and its coefficients do not involve total substrate concentration parameter $A_{Total}$ (but includes $[A_p]$).

It then follows that as $A_{Total}$ changes the roots of the cubic polynomial eq. (29) (for $\varepsilon$) change. However, the polynomial in eq. (28) cannot accommodate a changing $\varepsilon$ (being independent of

$A_{Total}$) unless the presence of $\varepsilon$ in this equation is eliminated by a suitable parameter choice. In this instance, eq. (28) can be satisfied independent of $\varepsilon$ (and $A_{Total}$).

This can only be accomplished if the coefficient of $\varepsilon$ and the constant term in the polynomial (eq. (28)) are both simultaneously zero. Thus we get kinetic constraints that permit the possibility of ACR. Since these expressions both include $A_p$, this also provides the ACR concentration of $A_p$ as shown below:

$$[A_p] \quad = \quad \frac{k_3 P_{Total}}{c_2\left(k_2 K_{Total} - k_3 P_{Total}\right)} \quad = \quad \frac{k_1 K_{Total}}{c_4\left(k_4 P_{Total} - k_1 K_{Total}\right)} \quad > \quad 0 \tag{31}$$

Hence, the above simultaneously establishes the ACR concentration of $A_p$, while providing us the kinetic constraints required for ACR. As mentioned earlier, it is also noteworthy to observe that a Case 1 symmetry in kinetics and enzyme concentrations trivially transforms this kinetic constraint to that observed earlier in the symmetric instance.

Thus, under conditions in, eq. (31), the polynomial, eq. (28) is satisfied independent of $A_{Total}$ or $\varepsilon$. The other polynomial (eq. (29)) can be rewritten as a polynomial in $\varepsilon$ (with $A_p$ assuming the fixed value given in eq. (31)) as shown below in eq. (32)

$$0 = \left[c_1 c_2^2 k_1 k_2^2 [A_p]^2\right]\epsilon^3 + \left[c_1(((([A_p] + P_{Total} - A_{Total})c_3 + [A_p]c_4 + 1)k_2 + P_{Total}c_3 k_3)[A_p]k_3 c_2 k_1\right]\epsilon^2 + $$
$$\left[(c_1([A_p]^2 c_4 k_1 + (((-A_{Total} + P_{Total})k_1 + P_{Total}k_4)c_4 + k_1)[A_p] - A_{Total}k_1)k_3 + [A_p]^2 c_2 c_4 k_2 k_4)k_3 c_3\right]\epsilon + \left[[A_p]c\right]$$

$$\tag{32}$$

Note that the coefficient of $\epsilon^3$ and the constant term in this polynomial are both positive, indicating the presence of one negative real root. This also implies that the product of the three roots is negative. Thus, if there exists an ACR branch (i.e. positive real roots in $\varepsilon$), it necessarily implies that there exists another positive real root for $\varepsilon$, indicating the presence of another second ACR branch.

## Sufficiency

For large enough $A_{Total}$, the sum of the roots (the ratio of the coefficient of the $\epsilon^2$ and the leading coefficient) is positive, and it can be shown that the discriminant of this polynomial is also positive (refer *Source code 1* [Section 5.1] and *Supplementary file 1*). This together guarantees the presence of two positive real roots in $\varepsilon$, proving the sufficiency of kinetic constraints in eq. (32) to obtain ACR in $A_p$ at some finite range of concentrations of $A_{Total}$.

Extension of this proof to ascertain associated features of ACR networks in DSP is provided in detail in *Source code 1* (Section 5.1) and *Supplementary file 1*.

## **Approximate concentration robustness**

We now turn to the case of approximate ACR wherein a substrate may exhibit concentration robustness approximately in some limiting regime of enzymatic action. We will show that such behaviour can be readily obtained in different scenarios.

We begin our analysis by focusing on the ordered DSP network. We assume that the concentration of the phosphatase is much higher than the concentration of the kinase or the substrate. Thus the dephosphorylation of substrates, $A_p$ and $A_{pp}$, can be approximated by simple first-order reactions.

### Approximate concentration robustness in the partial substrate ($A_p$)

We assume that the phosphorylation of $A_p$ is in the unsaturated limit and the phosphorylation of $A$ is saturating. In this case, the concentration of the $A_pK$ complex is negligible and further $K \approx \frac{K_{Total}}{1+\alpha A} \approx \frac{K_{Total}}{\alpha A}$. This implies that the flux of phosphorylation of $A_p$ is a zeroth-order reaction, $\frac{K_{Total}\beta}{\alpha}$. Since steady state in this network involves pairwise equilibrium, we have

$$Rate_{Phosphorylation-of-A} = Rate_{DePhosphorylation-of-A_p} \implies \frac{\beta K_{Total}}{\alpha} \approx A_p \gamma$$

where $\alpha$, $\beta$, $\gamma$ are kinetic constants of the network.

Thus the concentration of $A_p$ is approximately given by $A_p \approx \frac{\beta K_{Total}}{\alpha\gamma}$ for a range of $A_{Total}$.

**Approximate concentration robustness in the fully modified substrate ($A_{pp}$)**

Here similar to the logic above, if the phosphorylation of $A_p$ is in the saturated limit and the phosphorylation of $A$ is in the unsaturated limit, then (as a consequence of equilibrium between $A_p$ and $A_{pp}$) we find that $A_{pp}$ exhibits approximate ACR.

A similar proof assuming an excess of kinase in the network can be used to show the feasibility of approximate concentration robustness in $A$.

## Comment about different kinases and different phosphatases

The above logic can readily be used in the case of different kinases or phosphatases. In fact, having different enzymes only removes constraints on kinetic regimes/enzyme amounts.

## Random modification networks

The above approach can be employed to establish approximate ACR for one species in the random modification network. This is facilitated by having separate kinases and separate phosphatases. Here we simply need to ensure that the two production reactions for a substrate are acting in the saturated regime (these are associated with different enzymes) which can be assumed to perform other modifications in the unsaturated limit. Further enzymes involved in the removal of the substrate are assumed to be in large amounts relative to the substrate concentration.

## Appendix 2

### Parameter values

*Figure 2*

- A. $k_1 = 0.1$; $k_2 = 0.5$; $k_3 = 0.1$; $k_4 = 0.5$; $K_{Total} = 0.1$; $P_{Total} = 0.1$;
- B. $k_1 = 0.1$; $k_3 = 0.1$; $a_1 = 0.25$; $a_2 = 0.4$; $a_3 = 0.25$; $a_4 = 0.4$; $K_{Total} = 1$; $k_1 = 1$;
- C. $k_1 = 0.1$; $k_3 = 0.1$; $a_1 = 0.5$; $a_2 = 1.5$; $a_3 = 0.5$; $a_4 = 1.5$; $K1_{Total} = 1$; $P1_{Total} = 1$; $K2_{Total} = 1$; $P2_{Total} = 1$;
- D. $k_1 = 2.35$; $k_2 = 0.46$; $k_3 = 1.86$; $k_4 = 1.1$; $a_1 = 2.35$; $a_2 = 0.46$; $a_3 = 1.86$; $a_4 = 1.1$; $K1_{Total} = 1$; $P1_{Total} = 1$; $K2_{Total} = 1$; $P2_{Total} = 1$;
- E. $k_1 = 2$; $k_2 = 0.1$; $k_3 = 0.75$; $a_1 = 2$; $a_2 = 0.1$; $a_3 = 0.75$; $K1_{Total} = 0.1$; $K1_{Total} = 0.1$; $P_{Total} = 0.2$;

*Figure 3*

- A. (Panel 1 – Hopf and pitchfork bifurcation): $k_1 = 100$; $k_2 = 2$; $k_3 = 0.01$; $k_4 = 20$; $a_1 = 0.01$; $a_2 = 20$; $a_3 = 100$; $a_4 = 2$; $k_{b1} = 100$; $k_{b3} = 100$; $k_{b4} = 0.1$; $a_{b1} = 100$; $a_{b3} = 100$; $a_{b2} = 0.1$; $K_{Total} = 1.25$; $P_{Total} = 1.25$;
- A. (Panel 2 – pitchfork bifurcation): $k_1 = 100$; $k_2 = 2$; $k_3 = 0.01$; $k_4 = 20$; $a_1 = 0.01$; $a_2 = 20$; $a_3 = 100$; $a_4 = 2$; $k_{b1} = 100$; $k_{b3} = 100$; $k_{b4} = 0.1$; $a_{b1} = 100$; $a_{b2} = 0.1$; $a_{b3} = 100$; $K_{Total} = 10$; $P_{Total} = 10$;
- B. (Panel 1 – Hopf bifurcation): $k_1 = 150$; $k_2 = 50$; $k_3 = 1$; $k_4 = 10$; $a_1 = 1$; $a_2 = 10$; $a_3 = 150$; $a_4 = 50$; $k_{b1} = 100$; $k_{b3} = 0.01$; $k_{b4} = 500$; $a_{b1} = 0.01$; $a_{b2} = 500$; $a_{b3} = 100$; $K1_{Total} = 1$; $P1_{Total} = 1$; $K2_{Total} = 1$; $P2_{Total} = 1$;
- B. (Panel 2 – pitchfork bifurcation): $k_1 = 10$; $k_2 = 1$; $k_3 = 2$; $k_4 = 5$; $a_1 = 2$; $a_2 = 5$; $a_3 = 10$; $a_4 = 1$; $K1_{Total} = 1$; $P1_{Total} = 1$; $K2_{Total} = 1$; $P2_{Total} = 1$;

*Appendix 2—figure 1*

- $k_1 = 0.9$; $k_2 = 0.8$; $k_3 = 2$; $a_1 = 0.9$; $a_2 = 0.8$; $a_3 = 2$; $K1_{Total} = 0.1$; $K2_{Total} = 0.1$;

*Appendix 2—figure 2*

- A. (Panel 1 – Hopf bifurcation): $k_1 = 100$; $k_2 = 20$; $k_4 = 2$; $a_2 = 2$; $a_3 = 100$; $a_4 = 20$; $k_{b1} = 100$; $k_{b3} = 100$; $a_{b1} = 100$; $a_{b3} = 100$; $K_{Total} = 1$; $P_{Total} = 1$;
- B. (Panel 2 – dynamic simulation): parameter set in *Figure 3A* where $A_{Total} = 12.95$;
- C. (Panel 3 – pitchfork bifurcation): $k_1 = 0.1$; $k_3 = 2$; $k_4 = 5$; $a_1 = 2$; $a_2 = 5$; $a_3 = 0.1$; $K_{Total} = 1$; $P_{Total} = 1$;

*Appendix 2—figure 3*

- $k_1 = 0.1$; $k_3 = 0.1$; $K_{Total} = 1$; $P_{Total} = 1$;

*Appendix 2—figure 4*

- $k_2 = 2$; $k_4 = 20$; $a_2 = 20$; $a_4 = 2$; $k_{b1} = 10$; $k_{b3} = 10$; $a_{b1} = 10$; $a_{b3} = 10$; $K1_{Total} = 5$; $P1_{Total} = 5$; $K2_{Total} = 5$; $P2_{Total} = 5$;

*Appendix 2—figure 5*

- $k_1 = 1.25$; $k_2 = 1.1$; $k_3 = 2.5$; $k_4 = 0.4$; $a_1 = 1.25$; $a_3 = 2.5$; $a_4 = 0.4$; $K1_{Total} = 1$; $P1_{Total} = 1$; $K2_{Total} = 1$; $P2_{Total} = 1$;

*Appendix 2—figure 6*

- $k_2 = 0.5$; $k_3 = 0.1$; $k_4 = 0.5$; $k_{b1} = 1$; $k_{b2} = 1$; $k_{b3} = 1$; $k_{b4} = 1$; $K_{Total} = 0.1$; $P_{Total} = 0.1$;

*Appendix 2—figure 7*

- (Left Panel): $k_1 = 30$; $k_2 = 2$; $k_3 = 0.3$; $k_4 = 20$; $a_1 = 0.3$; $a_2 = 20$; $a_3 = 30$; $a_4 = 2$; $k_{b1} = 100$; $k_{b3} = 100$; $k_{b4} = 0.1$; $a_{b1} = 100$; $a_{b2} = 0.1$; $a_{b3} = 100$; $K_{Total} = 1$; $P_{Total} = 1$;
- (Right Panel): $K_{Total} = 20$; $P_{Total} = 20$;

*Appendix 2—figure 8*

- $k_1 = 0.1$; $k_2 = 1.5$; $k_3 = 2$; $k_4 = 0.1$; $k_5 = 1.5$; $k_6 = 2$; $K_{Total} = 0.1$; $P_{Total} = 0.1$;

*Appendix 2—figure 9*

- $k_1 = 1$; $k_2 = 2$; $k_3 = 0.5$; $k_4 = 1.2$; $k_{b1} = 1$; $k_{b2} = 1$; $k_{b3} = 1$; $k_{b4} = 11$; $K_{Total} = 1$; $P_{Total} = 1$;

## (A) MIXED RANDOM DSP

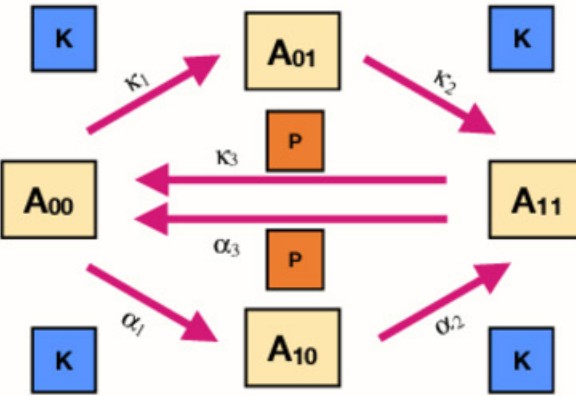

### Mixed Random 1
Common Kinase Common Phosphatase
Distributive Phosphorylation Processive Dephosphorylation

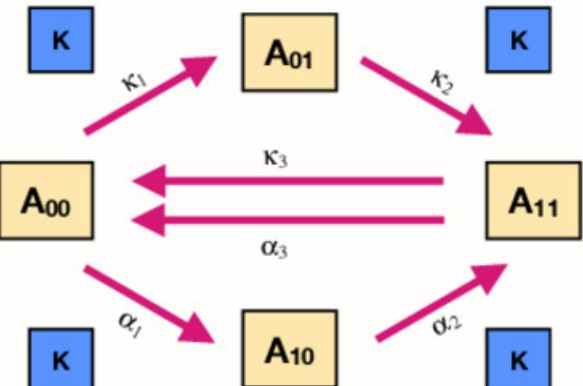

### Mixed Random 2a
Separate Kinase Common Phosphatase
Distributive Phosphorylation Processive Dephosphorylation
(Unsaturated)

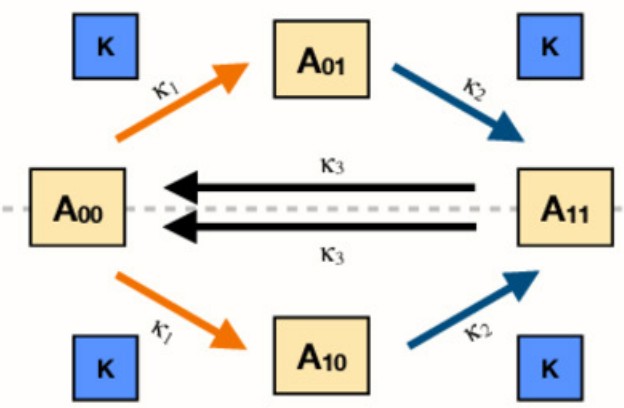

### (B) MIXED RANDOM 2a
Separate Kinase Common Phosphatase
Distributive Phosphorylation Processive Dephosphorylation
(Unsaturated)

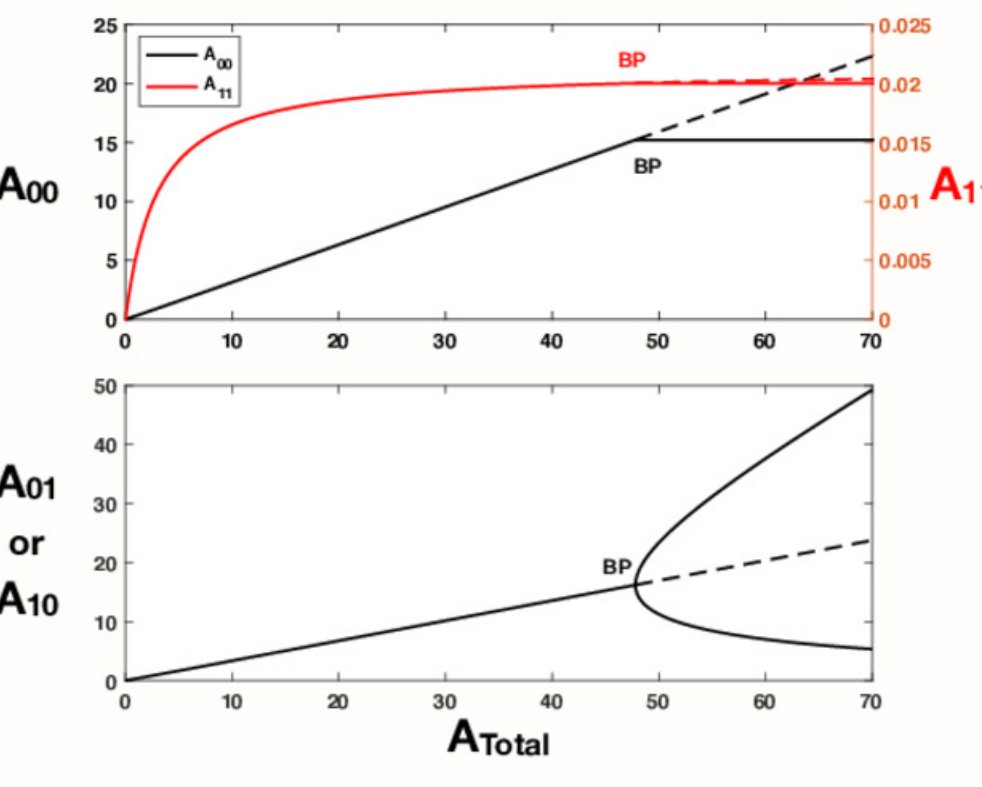

... of the mixed random ordered double-
... se and common phosphatase effecting
... horylation (mixed random 1), and the mixed
... nd common phosphatase effecting distributive
... the unsaturated regime (mixed random 2a).
... dom 2a network. Note that the concentration
... fixed in the asymmetric steady states. Dotted
... epresent stable steady states in the bifurcation

... code for appendix 2—

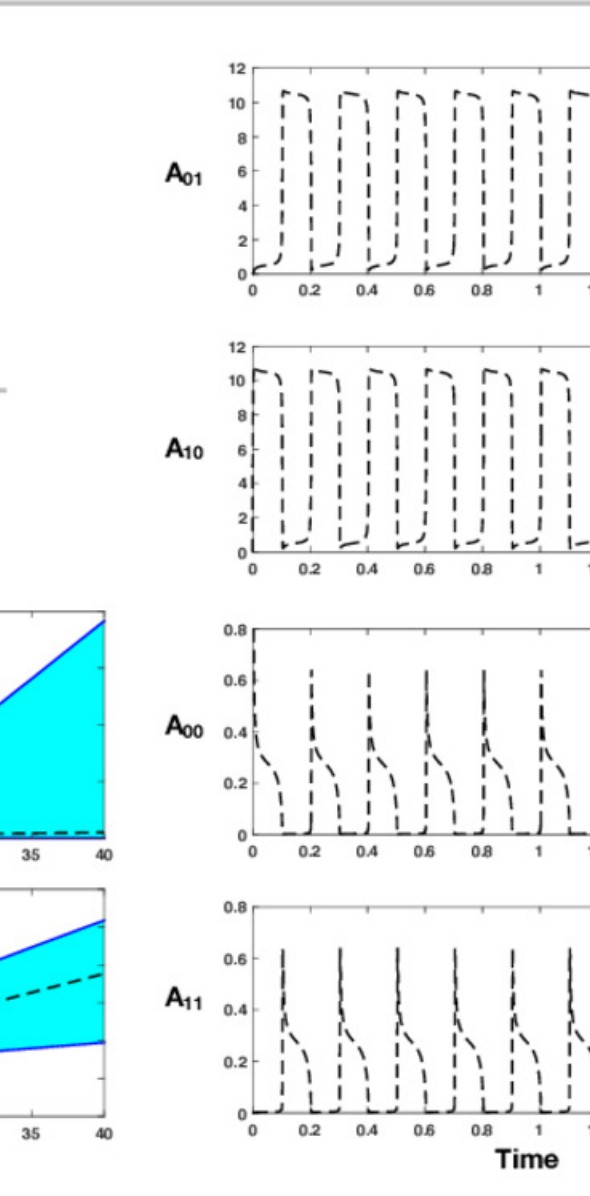

... random ordered double-site phosphorylation
... g on each modification site. Column 1 shows

*Appendix 2 continued on next page*

*Appendix 2 continued*

the presence of oscillations emerging through a Hopf bifurcation in the bifurcation diagram along $A_{Total}$. Column 2 shows long period oscillations in the system represented in *Figure 3A* from the main text (for a $A_{Total} = 12.95$). Such long period oscillations emerge when the oscillatory branch from the Hopf bifurcation approaches asymmetric stable steady states. Column 3 shows the presence of symmetry breaking through a supercritical pitchfork bifurcation. Note that the sum of concentrations of the partial substrates is conserved in the asymmetric steady states. This network is also capable of symmetry breaking through a subcritical pitchfork as seen in *Figure 3*. Dotted lines indicate unstable steady states, while solid lines represent stable steady states in the bifurcation diagram. Shaded regions in the bifurcation diagram indicate regions of oscillations, and the blue lines indicate bounds on concentrations during such oscillations. BP: pitchfork bifurcation; HP: Hopf bifurcation.

The online version of this article includes the following source code for appendix 2— figure 2:

- **Appendix 2—figure 2—source data 1.**

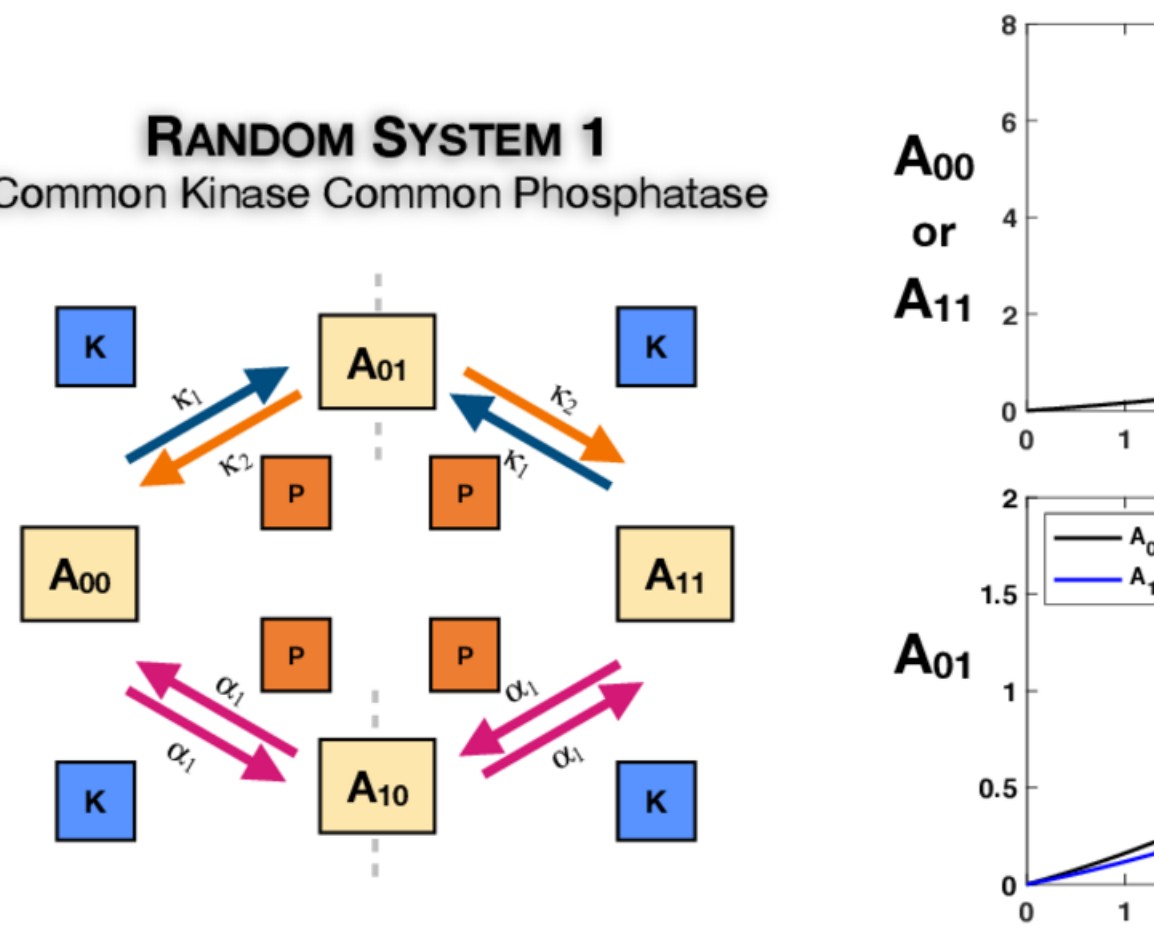

**Appendix 2—figure 3.** Case 1 symmetry breaking in the random ordered double-site phosphorylation with common kinase and common phosphatase acting on each modification site, even when one of the legs ($A_{00} \Longleftrightarrow A_{11}$) is incapable of breaking symmetry by itself (i.e. viewed as an ordered mechanism the kinetic parameters of the $A_{00} \Longleftrightarrow A_{11}$ leg forbid independent symmetry breaking; see main text and analytical work for discussion). Homeostasis (absolute concentration robustness) is observed in both partial substrates. Dotted lines indicate unstable steady states, while solid lines represent stable steady states in the bifurcation diagram. BP: pitchfork bifurcation.

The online version of this article includes the following source code for appendix 2—figure 3:

• **Appendix 2—figure 3—source data 1.**

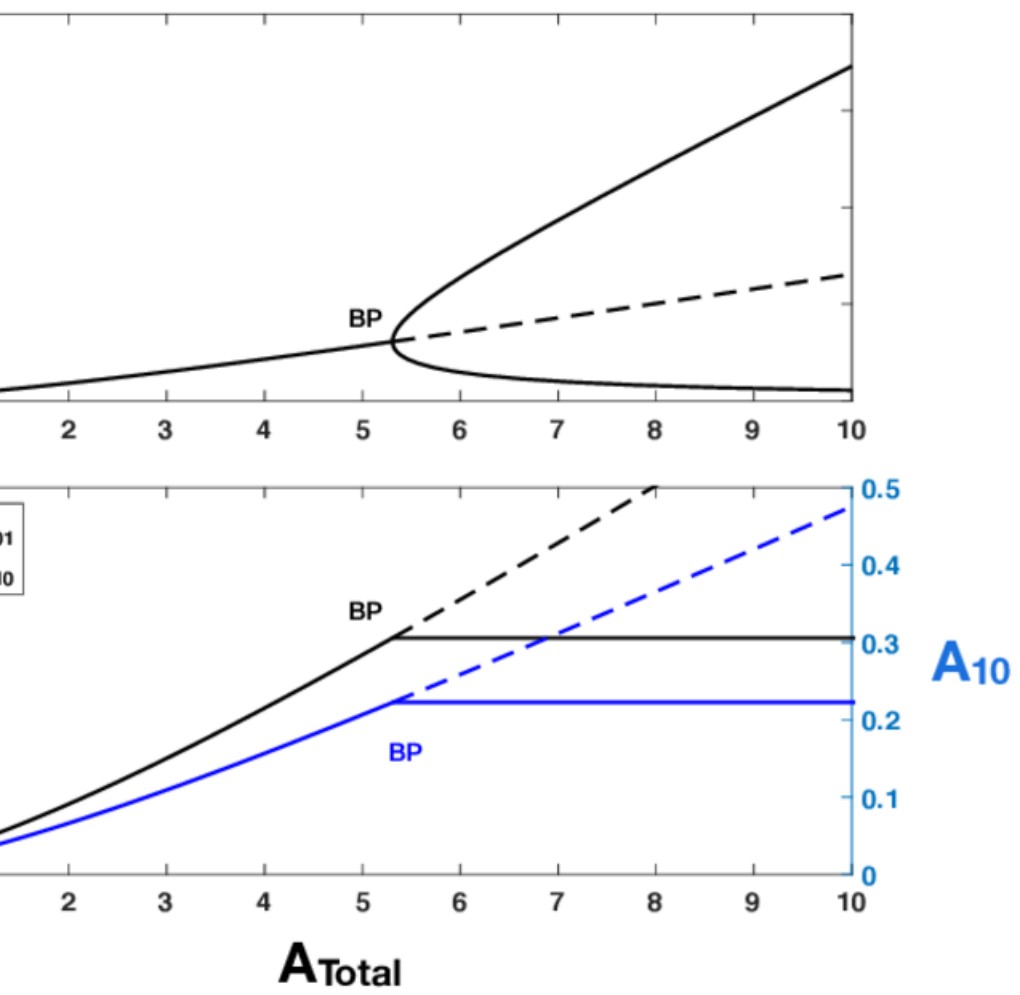

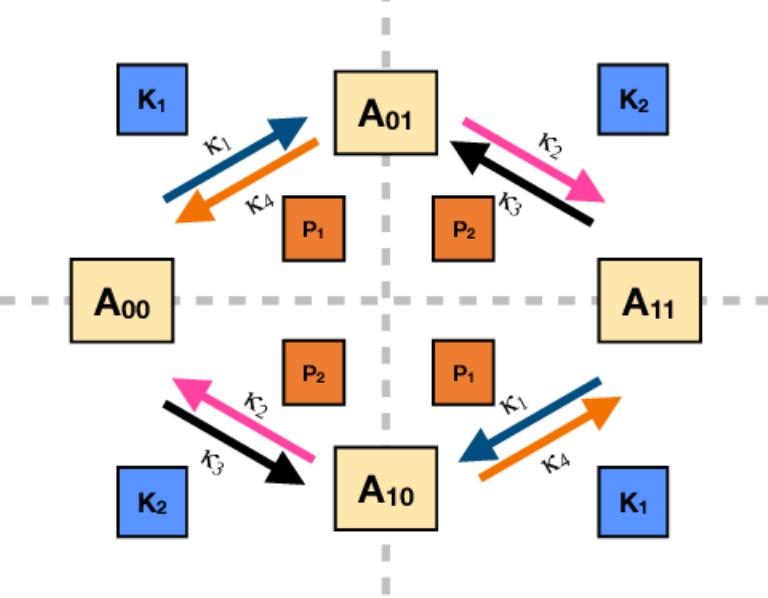

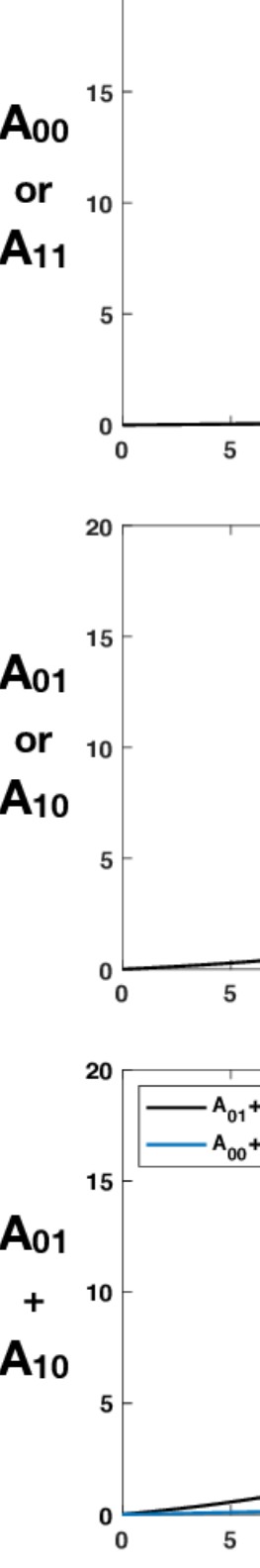

**Appendix 2—figure 4.** Case 3 symmetry breaking through a supercritical pitchfork bifurcation in the
*Appendix 2—figure 4 continued on next page*

*Appendix 2—figure 4 continued*

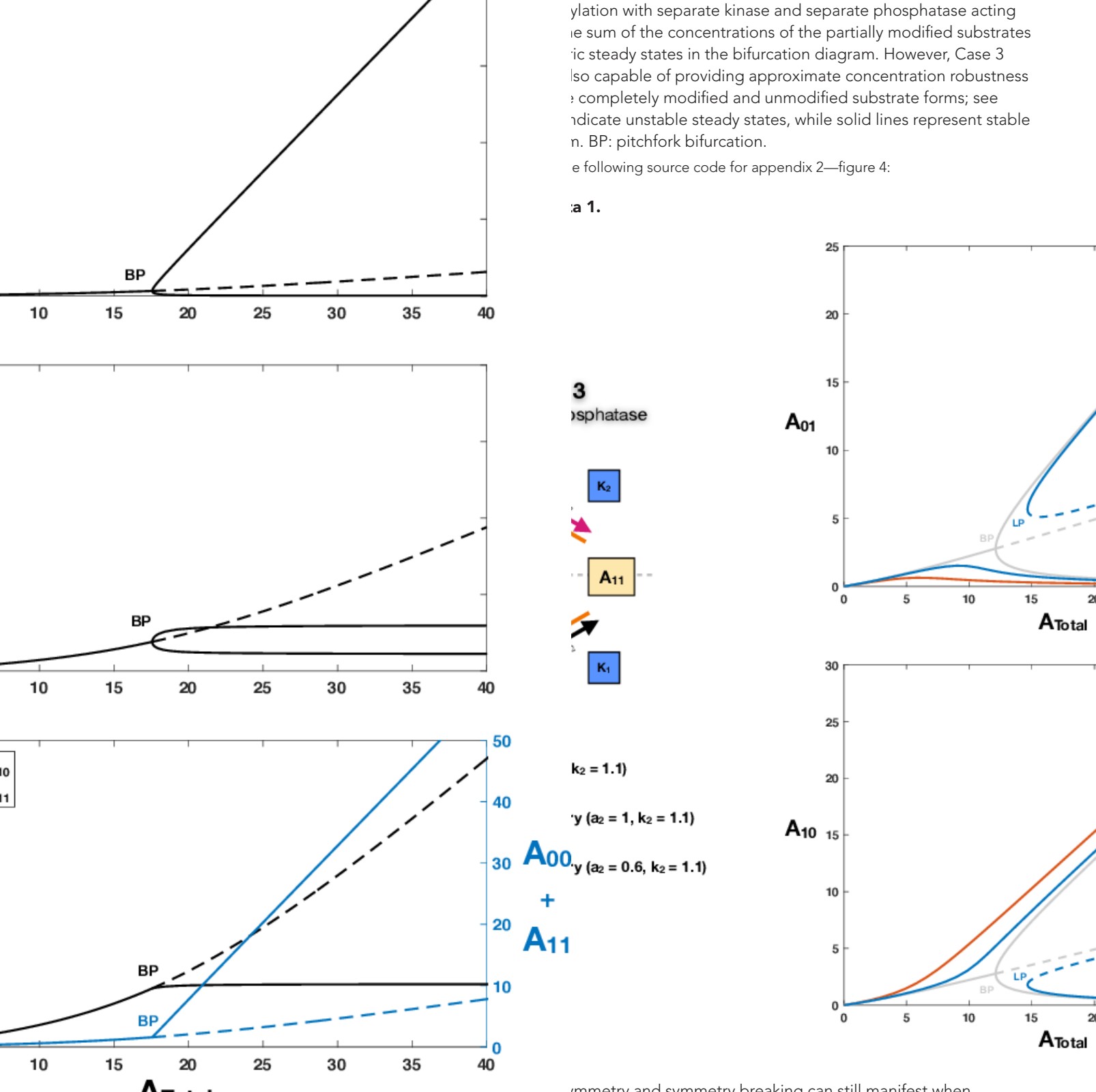

ylation with separate kinase and separate phosphatase acting
e sum of the concentrations of the partially modified substrates
ic steady states in the bifurcation diagram. However, Case 3
lso capable of providing approximate concentration robustness
completely modified and unmodified substrate forms; see
dicate unstable steady states, while solid lines represent stable
n. BP: pitchfork bifurcation.

e following source code for appendix 2—figure 4:

mmetry and symmetry breaking can still manifest when
the network is only approximately symmetric. This is represented through the example of Case 2
symmetry breaking in the distributive ordered double-sitephosphorylation with separate kinase and
separate phosphatase affecting phosphorylation and dephosphorylation, respectively. Dotted lines

*Appendix 2—Figure 5 continued on next page*

*Appendix 2—Figure 5 continued*
indicate unstable steady states, while solid lines represent stable steady states in the bifurcation diagram. BP: pitchfork bifurcation; LP: saddle node bifurcation.

The online version of this article includes the following source code for appendix 2—figure 5:

- **Appendix 2—figure 5—source data 1.**

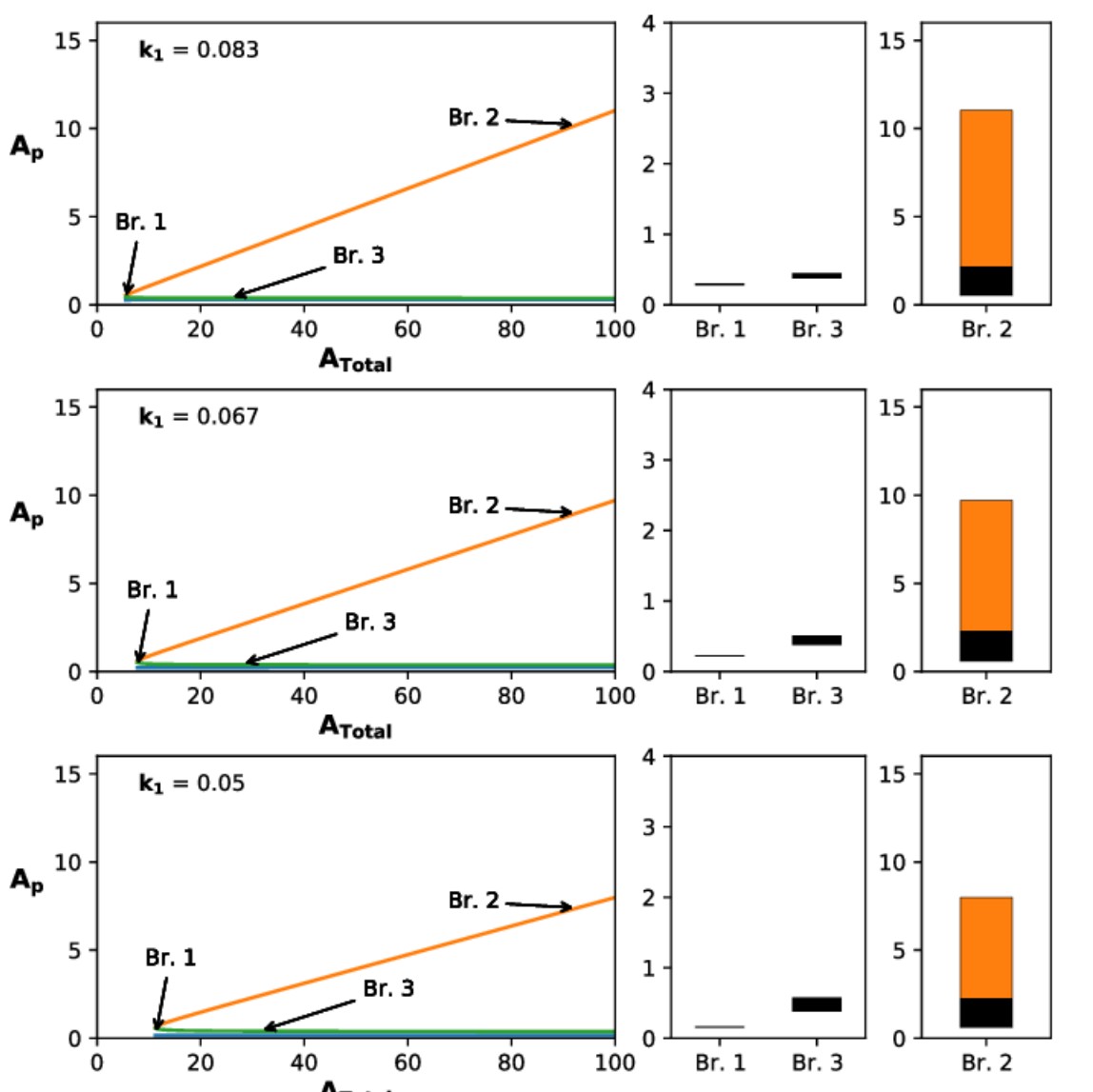

**Appendix 2—figure 6.** Approximate concentration robustness shown by systems that deviate from exact symmetry: Figure represents the approximate concentration robustness shown by $A_p$ in the ordered distributive double-site phosphorylation with common kinase and common phosphatase when the network is not symmetric. Note that the system is symmetric when $k_1 = 1$ and is presented in *Figure 2A*. In order to ascertain the behaviour of the system and in particular absolute concentration robustness (ACR) characteristics of $A_p$, we perturb the kinetics ($k_1$) at six values between 50% and 150% of the symmetric value (while keeping all other kinetics fixed) and present the result in six panels
*Appendix 2—figure 6 continued on next page*

*Appendix 2—figure 6 continued*
composed of three plots each; the first represents a bifurcation diagram showing the presence of
steady-state branches in a range of $A_{Total}$(0–100) where multistability is present. The other two are bar
graphs representing the norm of the concentration (max value – min value) of $A_p$ on the branches 1
and 3, and branch 2, respectively, across the entire range of variation ($A_{Total} = 0 - 100$). Note that in
the symmetric network we obtain a perfect pitchfork bifurcation; however when the system deviates
from exact symmetry, multistability is obtained through a saddle node bifurcation as shown here and
... smaller in magnitude on branches 1 and 3
...nches in the perfectly symmetric system)
...ogue of the symmetric branch). The norm
...e norm shown here is only for the range of
...arying negligibly with changing $A_{Total}$. This
...etween 50% to 150% and is again indicative
...nmetric systems for a large range of total

...de for appendix 2—figure 6:

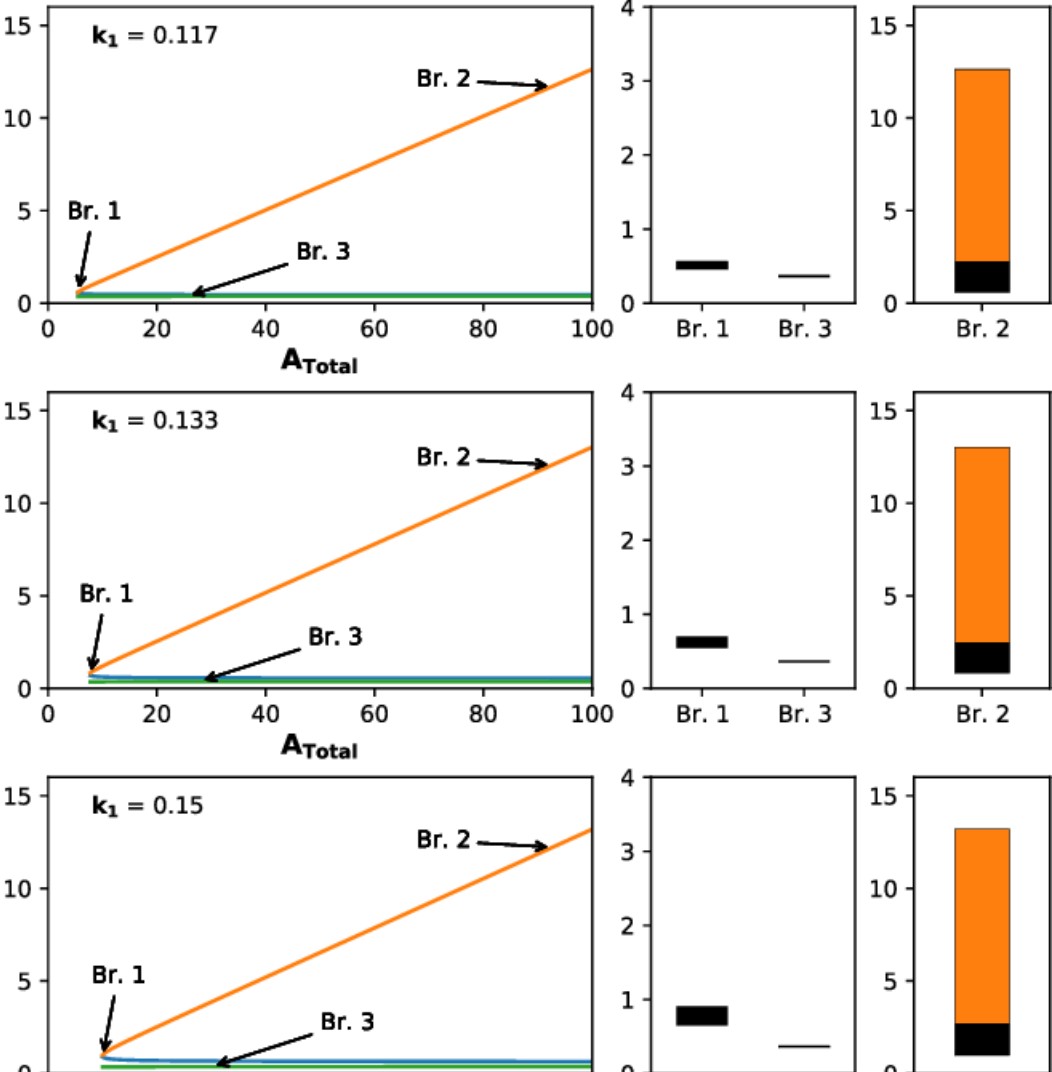

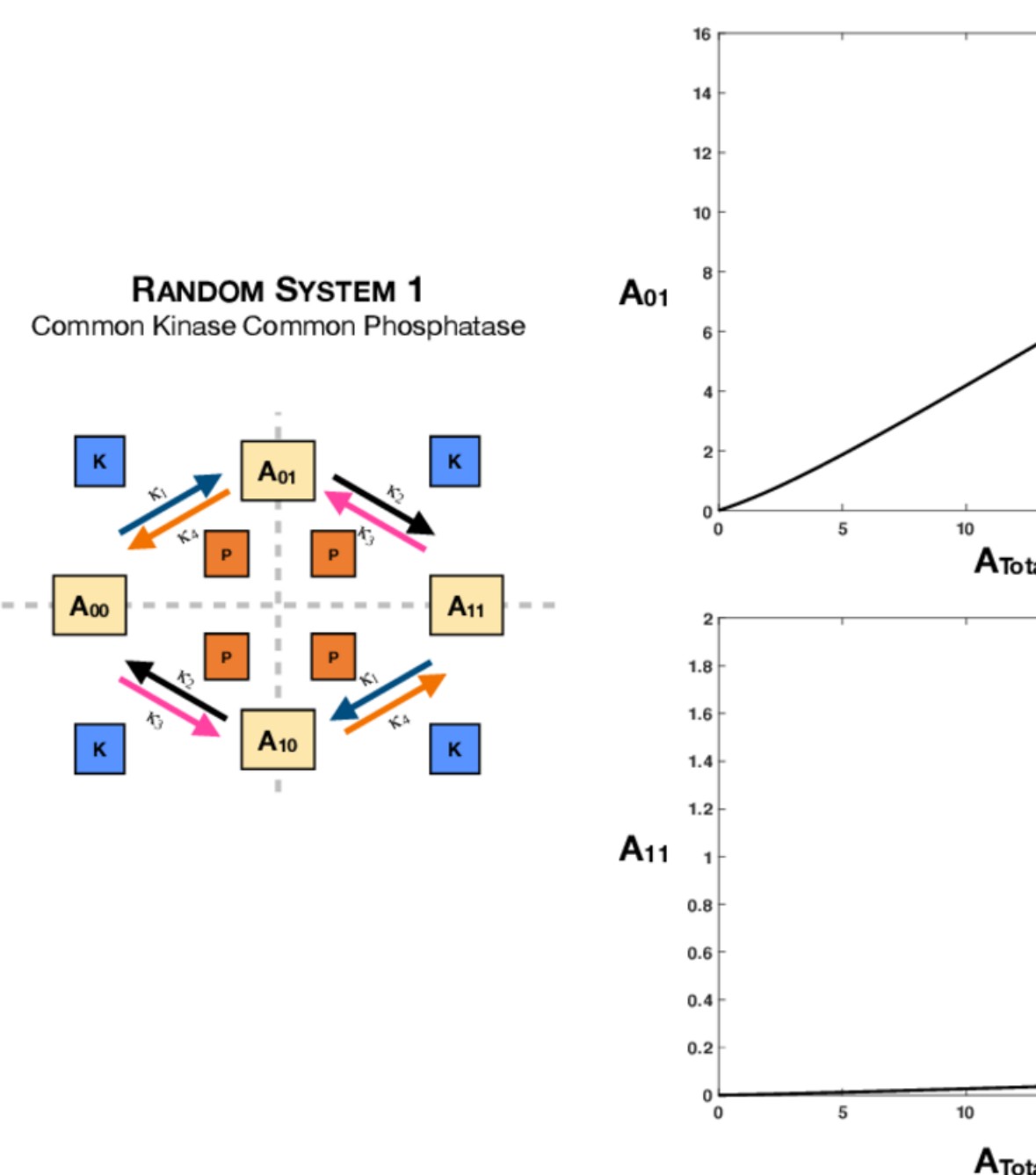

**Appendix 2—Figure 7.** Total enzyme concentrations are an additional lever (apart from basic network kinetics) that can independently tune symmetry-breaking behaviour in multisite phosphorylation networks. This is represented through the example of Case 3 symmetry breaking in the distributive random double-site phosphorylation with common kinase and common phosphatase). Panel 2 shows how increasing enzyme concentrations (left ⟶ right) can lead to loss of oscillatory behaviour in the network for the same basal kinetics parameters.

The online version of this article includes the following source code for appendix 2—figure 7:

- **Appendix 2—figure 7—source data 1.**

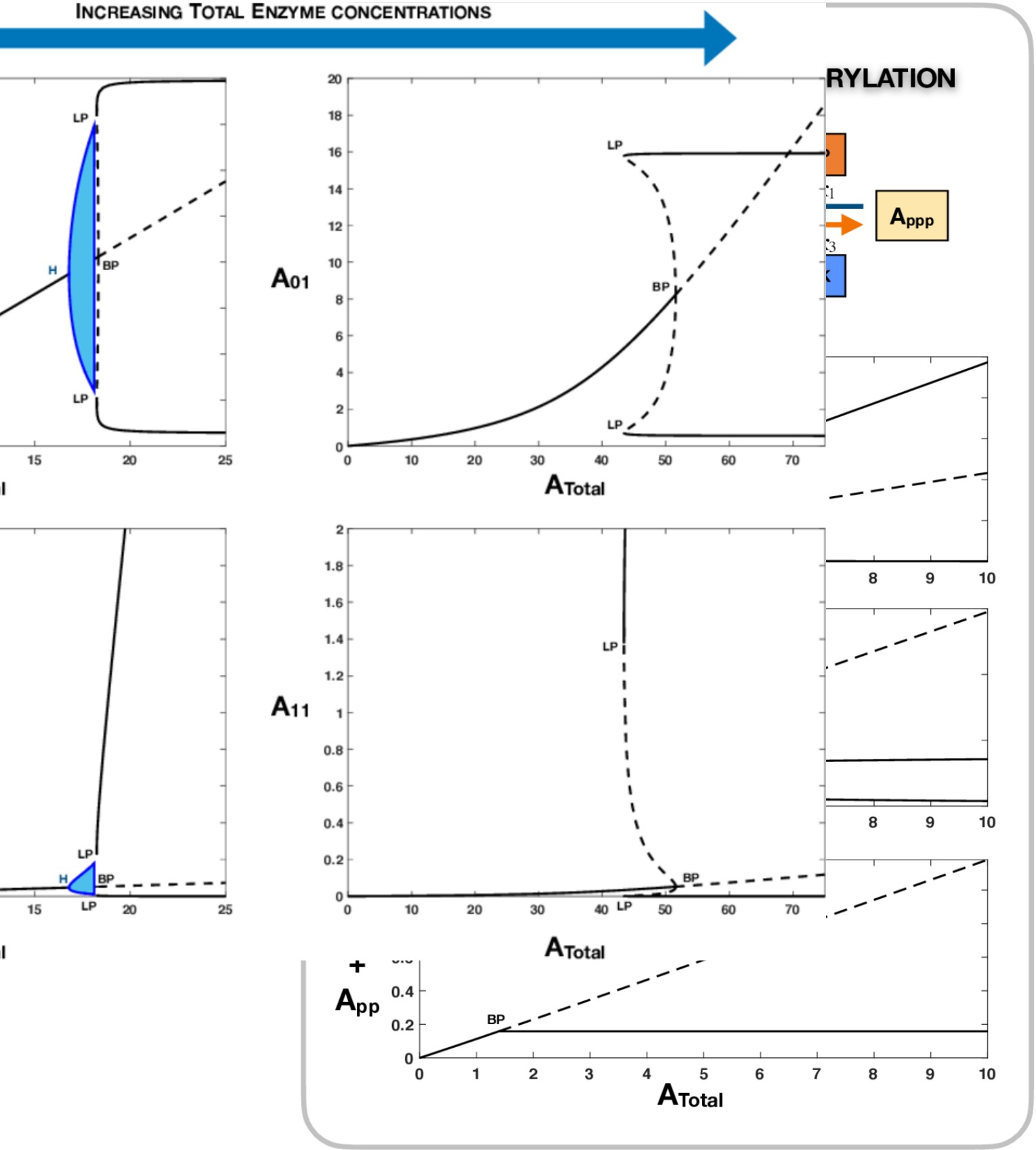

*Appendix 2—Figure 8 continued on next page*

*Appendix 2—Figure 8 continued*

**Appendix 2—Figure 8.** Case 1 symmetry breaking in the distributive ordered triple-site phosphorylation network. Here we observe absolute concentration robustness in the sum of the concentrations of the partially modified substrates. Dotted lines indicate unstable steady states, while solid lines represent stable steady states in the bifurcation diagram. BP: pitchfork bifurcation.

The online version of this article includes the following source code for appendix 2—figure 8:

- **Appendix 2—figure 8—source data 1.**

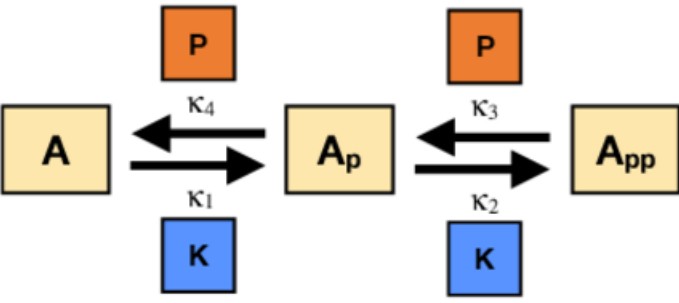

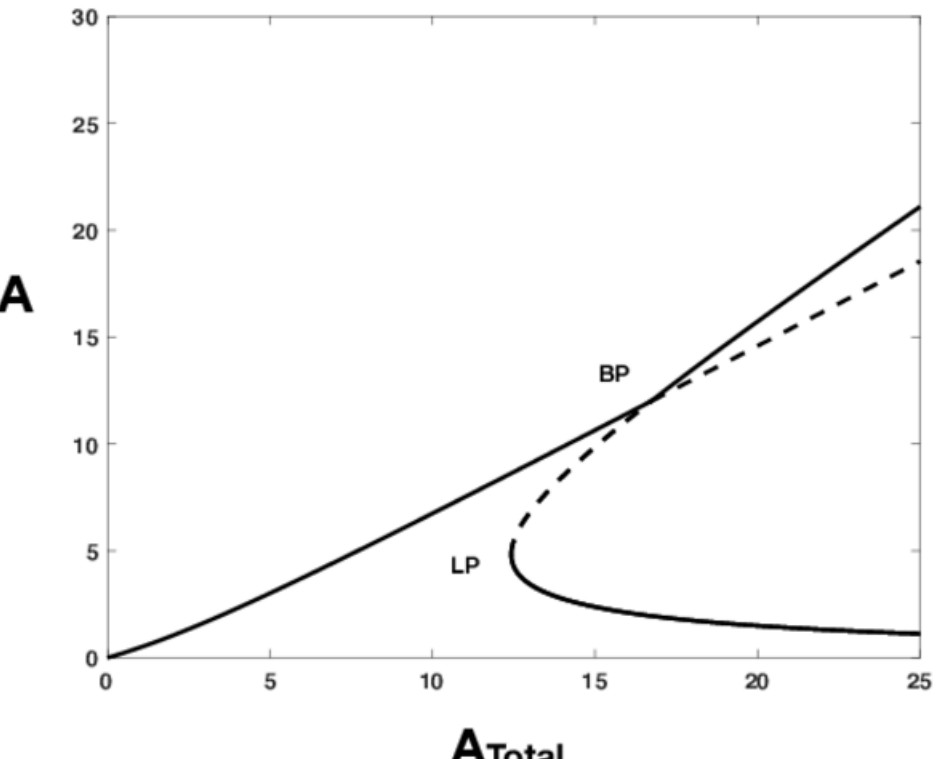

**Appendix 2—Figure 9.** Absolute concentration robustness (ACR) in ordered double-site phosphorylation: The ordered double site with common kinase common phosphatase is capable of exhibiting (exact) ACR in the partially modified substrate ($A_p$) with respect to changing total substrate concentration even in the absence of symmetry in the kinetics or total enzyme amounts (however, *Appendix 2—Figure 9 continued on next page*

*Appendix 2—Figure 9 continued*

a weaker constraint is required to enable this). The figure presents a computational example of this (complementing the discussion in the main text and Appendix 1). We observe a single non-ACR-exhibiting branch of steady states exchanging stability with branches of ACR-exhibiting steady states through a transcritical bifurcation (as opposed to a pitchfork bifurcation as seen in the symmetric examples earlier). The concentration of $A_p$ is fixed to be mathematically exact on these ACR branches as shown in the top-right plot. The unstable ACR branch emerging out of the transcritical bifurcation becomes stable through a saddle node bifurcation as shown. Dotted lines indicate unstable steady states, while solid lines represent stable steady states in the bifurcation diagram. BP: pitchfork bifurcation; LP: saddle node bifurcation.

The online version of this article includes the following source code for appendix 2—figure 9:

- **Appendix 2—figure 9—source data 1.**
- **Source code 1.** Maple document (Containing detailed proofs and parameter values used for the figures).

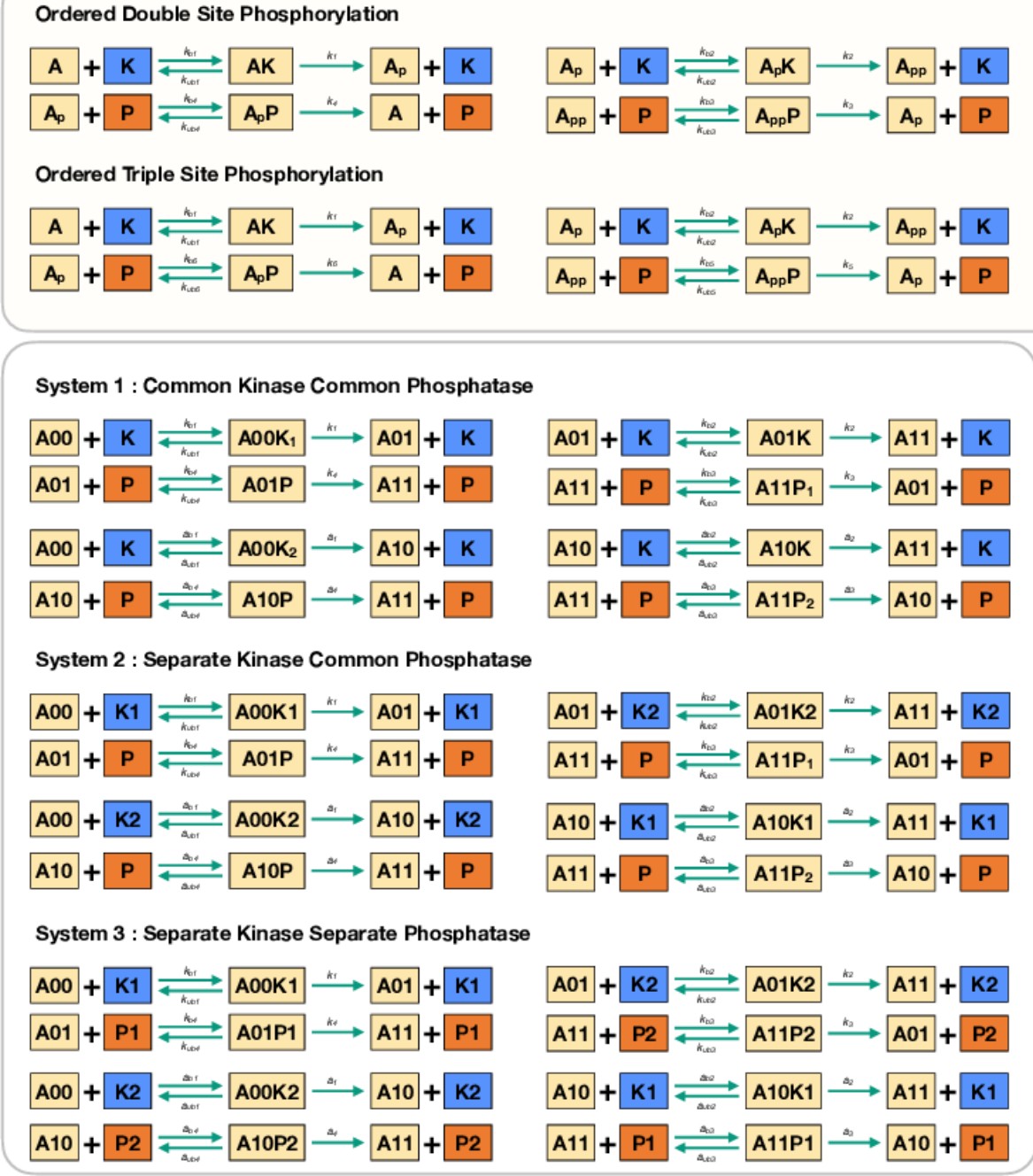

**Appendix 2—Figure 10.** Detailed model description of the various multisite phosphorylation networks used in this paper. The constituent binding, unbinding, and catalytic reactions of each modification step are described in detail and are modelled using mass kinetic description.

