## [Decision Letter]

**Acceptance summary:**

This paper proposes a new theoretical perspective to analyze multisite modification networks based on the symmetries that they display and the way in which these symmetries can be effectively broken. This perspective has several interesting implications, including for the analysis, synthetic design and evolution of these networks.

**Decision letter after peer review:**

Thank you for submitting your article "Symmetry breaking meets multisite modification" for consideration by *eLife*. Your article has been reviewed by 3 peer reviewers, one of whom is a member of our Board of Reviewing Editors, and the evaluation has been overseen Aleksandra Walczak as the Senior Editor. The reviewers have opted to remain anonymous.

Essential Revisions:

The three Reviewers concur to find the manuscript of interest but raise concerns related to its presentation and its relation to the previous literature. They welcome a resubmission provided that it addresses the points raised by the Reviewers and in particular the following essential points:

1) A clarification of the nature of the differences between the 'cases': to what extent are they really different? wouldn't it possible to have a more synthetic and unified presentation? all reviewers pointed out the manuscript in general and Figure 1D in particular would benefit from an effort of clarification

2) Relation to previous literature: to what extent are the results different from previous proposals, in particular zero-order ultrasensitivity, sequestration effects and previous analyses of symmetry breaking in multisite modification systems? (see in particular references given by Rev #2)

3) To what extent is symmetry breaking a necessary condition for the properties that it can induce, in particular absolute concentration robustness? and to what extent does symmetry need to exactly hold?

4) Make more explicit the evolutionary scenarios that are only briefly sketches: what are the assumptions on the symmetries that are 'naturally' expected to be present prior to any evolution?

5) Include mention of symmetries in other parameters, notably dissociation constants

*Reviewer #1 (Recommendations for the authors):*

Post-translational modifications of proteins at multiple sites, for instance by phosphorylation, play a key role in signal transduction and processing. Their system-level behaviors, which include bistability and oscillations, can be analyzed on a case-by-case basis with mathematical tools from dynamical systems. The main contribution of the paper is to propose a new perspective that goes beyond this case-by-case approach. This is achieved by pointing out that protein networks based on multisite modifications can display symmetries and that these symmetries can be broken to give rise to properties of biological relevance: ordering, directionality, concentration robustness,…

The paper includes on one side a mathematical analysis of two-site modifications systems, with an detailed analysis of the symmetries that they may display, the conditions for these symmetries to be present, and the conditions under which they may be broken. On the other side, the paper includes an extensive general discussion of the relevance of these analyses for actual biological system.

This is not a standard paper presenting a specific result or a novel technical method but a thought-provoking proposal revisiting the dynamics of multisite modification systems from a new perspective. While it is difficult to judge it by the usual standards, it should be of broad interest. For instance, no reference is made to any particular experimental system but the authors convincingly argue for the relevance of their point of view for interpreting natural systems and designing synthetic ones.

My main comment concerns the generality of the concept of symmetry: the paper advocates it in general but focuses on simple cases of reflection symmetry with 2-site modification systems. What other symmetries, if any, may be expected and of interest? How may the approach generalize beyond 2-site modification systems? Addressing exhaustively these questions may be outside the scope of a single article, but it would be worth fleshing out a few possible extensions.

– Figure 1D is key to define what is meant by symmetry but in my opinion not sufficiently informative. In particular the symmetry in kinetic structure is highlighted while the constraints on enzymes, which seem as essential, are not represented. For instance requiring k1=k3 is not sufficient, something like k1[K1]=k3[P2] is in fact needed? Also, as represented, Case 1 and Case 2 appear as formally equivalent (related through a 45 degree rotation). I was also wondering if more transparent notations could be used for ki, aj, which would reflect the (possible) symmetries?

– Another example of information processing at the molecular level where symmetry plays a key role is the MWC model of allostery. It may be worth mentioning it as, beyond the analogy, post-translational modifications can be associated with allosteric transitions.

– The two concepts of information processing and symmetry breaking bring to mind the possibility of encoding a bit of information in the symmetric states: could it be relevant in the context of post-translational modifications?

*Reviewer #2 (Recommendations for the authors):*

In this paper, the authors demonstrated that the symmetry in the concentration of different phospho-forms of a protein could be broken in multisite phosphorylation processes in proteins, even if the reaction's kinetic parameters to produce those are the same. They analyzed some different classes of symmetries in the kinetics for a protein with two phosphorylation sites and showed some of those shows the pitchfork bifurcation or the Hopf bifurcation. Also, they discussed that the relevance of the symmetry breaking for the absolute concentration robustness.

The analysis of classes of networks with different symmetries may be of use for future research in the field. Also, the relationship of the symmetry breaking to the absolute concentration robustness may be interesting. However, the discussion of whether symmetry breaking is a necessary condition for the absolute concentration robustness is still lacking. Moreover, the connection to previous studies has not been argued enough. For example, the symmetry-breaking mechanism may be related to the Goldbeter-Koshland-type zero-order ultrasensitivity and the enzyme competition. However, the connection has not been discussed sufficiently. Also, the symmetry breaking in the multisite modification has been studied for networks with feedback. However, mention of those studies is still lacking.

This paper still has several problems to be addressed, as described below.

1) The writing often lacks clarity and sharpness and is poorly organized. Although the authors showed their analyses for different symmetries in order from Case 1 to 3, the central mechanism of the symmetry breaking may be almost the same in these three classes. Moreover, the ordered distributive DSB seems sufficient to demonstrate the symmetry breaking by such a central mechanism. Hence, I recommend reorganizing the paper; to state detailed analyses for the ordered distributive DSB and an intuitive explanation at first, and then discuss three classes in the following sections.

2) The authors stated that Case 1 symmetry breaking was observed "only in the common kinase, common phosphatase case," and Case 2 symmetry breaking "is broken only for different kinase and different phosphatase case." It looks strange to me. Changes of variables, e.g., from A_00 to A_01, A_11 to A_10, k_1 to k_2, k_3 to a_2, and so on, can transform one case to the other. Thus, the two cases seem to be identical. Please explain why those two cases are different.

3) The mechanism of the symmetry breaking may be a combination of the Goldbeter-Koshland-type zero-order ultrasensitivity and a competition for the enzyme among different phospho-forms of the protein. However, there is no discussion about the relationship of symmetry breaking they found to zero-order ultrasensitivity and phenomena related to the enzyme competition.

4) The authors claimed the importance of the symmetry among phosphorylation and dephosphorylation speeds. However, even if the phosphorylation and dephosphorylation speeds are the same, the difference in dissociation constants can drastically change the actual reaction speed when substrates with different modification states compete for the limited amount of enzyme. (For example, Hatakeyama and Kaneko, PLoS Comput. Biol. (2014) and Hatakeyama Kaneko, Phys. Rev. Research (2020)). Hence, I recommend that the authors emphasize the importance of the symmetry in the dissociation constant as well as the modification speed, to make a reader pay attention.

5) The author demonstrated that if the network is symmetric and the symmetry in concentration is broken, the concentration of some phospho-form of the protein shows the absolute robustness. However, they did not show whether symmetry breaking is necessary for the absolute concentration robustness. Does a network without symmetry never show the absolute concentration robustness?

6) The authors cited papers only written by Ueda's group (Ode and Ueda, (2018), Jolly et al., (2012), Sugui et al., (2017), and Shinohara et al., (2017)) about the circadian clock as references. However, the circadian clock generated by the multisite modification has long been studied in cyanobacteria. Thus, the basic concept has already been stated in previous studies. For example, a generation of the oscillation by a protein with two phosphorylation sites was published in Rust et al., "Ordered phosphorylation governs oscillation of a three-protein circadian clock." (2007). The importance of enzyme sequestration for the oscillation was published in van Zon et al., "An allosteric model of circadian KaiC phosphorylation." (2007). Also, the mechanism of temperature compensation by the competition for the enzyme was published in Hatakeyama and Kaneko, "Generic temperature compensation of biological clocks by autonomous regulation of catalyst concentration." (2012). I recommend citing those papers in addition to Ueda's paper, for the healthy development of the scientific community.

7) The symmetry breaking in the multisite modification has been studied, at least for networks with explicit feedback. Hence, it may be better to tone down the title and some claims.

*Reviewer #4 (Recommendations for the authors):*

In this manuscript, Ramesh and Krishnan analyze on a theoretical level symmetry breaking in multisite models of covalent modification. The basic idea is that while the underlying network dynamics may preserve certain symmetries, these symmetries can be broken leading to asymmetric states, with for example unequal levels of doubly modified or unmodified phosphoforms. The symmetry breaking requires the presence of nonlinearities which fundamentally result from sequestration effects of the necessary enzymes/substrates. Overall, I found the paper to be a highly original contribution, with some very nice results that could open up new directions, for example in understanding the evolution of asymmetric networks. Their extensive analytical and numerical work clearly supports their conclusions, and suggestions for experimental signatures of broken symmetry are given. This work may have consequences for the fields of network evolution and synthetic biology.

In general, I found the manuscript to be well-written. Nevertheless, the paper is definitely not an easy read, despite clearly significant efforts from the authors due to the complexity of the subject matter.

One aspect that needs further attention is to what extent the system needs to be exactly symmetric to exhibit the behavior studied by the authors. This is an important point as some of the required symmetries are not natural: for example, in Case 1 of Figure 1D, k_1=k_3 which means that a phosphorylation and a dephosphorylation reaction must be precisely balanced. There is some effort towards analyzing this in Figure S5, but I think this aspect needs to be given much greater prominence and discussion also in the main text.

The authors also argue that evolution may have started with a symmetric network with broken symmetry as a starter towards asymmetric networks. This is an interesting idea but isn't it more likely that the networks were just asymmetric to begin with (which has a much bigger parameter space), and that was subsequently exaggerated, rather than starting from an a priori fine-tuned symmetric configuration? A similar issue arises with claims that an observed asymmetric network may not have biased dynamics but may instead simply be due to symmetry breaking within a symmetric network. Without more knowledge surely the former is still much more likely?

Key to the symmetry breaking is the nonlinearity introduced by sequestration effects without the need for explicit feedback. However, I think this needs an expanded discussion (probably with some equations) in the main text rather than being relegated to the (extremely large) methods. This is a central point, I think.

I didn't find the left side of Figure 1D very helpful for my understanding. Also, the explanation of the "square" network topology wasn't made very clear; I didn't find it more straightforward than the depictions in Figure 1A-C, for example.

---

## [Author Response]

Essential Revisions:The three Reviewers concur to find the manuscript of interest but raise concerns related to its presentation and its relation to the previous literature. They welcome a resubmission provided that it addresses the points raised by the Reviewers and in particular the following essential points:1) A clarification of the nature of the differences between the 'cases': to what extent are they really different? wouldn't it possible to have a more synthetic and unified presentation? all reviewers pointed out the manuscript in general and Figure 1D in particular would benefit from an effort of clarification

We explain here the logic for our arrangements, as well as the changes implemented to make this more clear.

The first point to note is that while we have presented the symmetries by referring to the topology of the underlying network, this has depicted only the substrates, with the enzymes being implicit in the arrows/connectors. In fact, for the symmetry to be present in the relevant network, not only should they be seen at the level of the substrate, but also with the associated enzymes. It is this feature which makes the network symmetric. When viewed in this light, the difference between Case 1 and Case 2 symmetry becomes clear: in the latter, symmetry involves the same modification rates and total amounts of enzyme pairs which are either kinases or phosphatases, whereas in Case 1 symmetry, the pairing is between a kinase and a phosphatase. Thus, in addition to the difference in implication (partial phosphoforms behaving identically—Case 2—or an absence of directionality—Case 1), the key ingredients of the symmetry when viewed from both substrates and enzymes together, is different.

We note here that if we had classified the symmetry only based on substrate, then in further analyzing the symmetry in detail, this would have naturally bifurcated into the two classes we have presented (they, being essentially different when viewed from both the full network perspective as well as the consequence).

The Associated changes in the manuscript to discuss this are

a) We have added multiple sentences in the paragraph “Associated Network Symmetries” in the Results section (Page 5, lines 166-184) to discuss this.

For instance – the following sentence is added in line 168.

“Note that in this depiction, the nodes of the network represent substrates, while the enzymes are implicitly present in the arrows: both substrates and enzymes together constitute an enzymatic reaction network of this type”

The following sentence is added in line 180.

“Thus enabling such symmetries establishes correspondences/constraints between different pairs of enzymes”

In addition, while introducing the symmetries in the paragraph “Network Symmetry meets multisite phosphorylation” (Page 6, line 195), we make this point clear, with the following change.

“The difference between them is what the symmetric nodes of the network correspond to in the context of multisite modification along with the fundamentally distinct pairings of enzymes in each case (discussed further below)”

Finally, after discussing Case 1 and Case 2 Symmetries, we have introduced a new paragraph (Page 6, lines 228-235):

“Difference between Case 1 and Case 2 Symmetries. Case 1 and Case 2 Symmetries involve different pairs of symmetric nodes. As noted earlier, the symmetries require both intrinsic rate constants and enzyme amounts to be equal for different pairs of enzymes. The essential difference between the two cases is the essentially different enzyme pairs associated with this. In Case 1 Symmetry, the pairing is between enzymes of different types (a kinase and a phosphatase), while in the Case 2, it is between enzymes of the same type (between kinases and between phosphatases). This is exactly why the Case 1 symmetry is not possible in the separate kinase common phosphatase network, while the Case 2 symmetry is.”

b) Figure 1(D) (Page 4), the LHS, has been altered to make clear the fact that the symmetries we present also imply a “symmetry” in arrows, which implicitly contain the enzymes.

Here we show how symmetry about one axis, involves the rates of corresponding production or removal reactions for the associated symmetric nodes to be the same (colour coded with the same colour) and then show that the like-coloured arrows are associated with enzymes of different types (Case 1) or of the same type (Case 2). In the RHS we have made explicit the requirements on enzymes by adding further information at the bottom of the RHS of 1(D). We have also removed some extraneous schematic graphical elements in this figure which were not serving an essential purpose.

c) The caption of Figure 1(D), (Page 4), has been updated to reflect this as well.

2) Relation to previous literature: to what extent are the results different from previous proposals, in particular zero-order ultrasensitivity, sequestration effects and previous analyses of symmetry breaking in multisite modification systems? (see in particular references given by Rev #2)

These results are essentially different for the following reasons

1. They focus on the intrinsic behaviour of the modification system, with no external feedback invoked

2. They consider a range of distinct symmetries

3. They examine together the cases of common/separate kinases and common/separate phosphatases

4. They are established explicitly analytically as well as computationally, along with an explicit analysis of the parametric dependence

5. These arise even in the deterministic setting, without invoking stochastic effects

Enzyme sequestration plays an essential role, as in its absence the system would be linear. Zero order ultrasensitivity can combine with enzyme sequestration/competition to generate interesting behaviour. However, in our case (i) we do not invoke zeroth order ultrasensitivity and (ii) it is not an essential ingredient. This is seen for instance, by looking at the analytical conditions enabling symmetry-breaking (which does not make any a priori restriction on the enzymatic regime), as well as computational results, which manifestly indicate that zeroth order ultrasensitivity is not needed. Thus, zeroth order ultrasensitivity, while capable of introducing interesting behaviour in combination with enzyme competition is neither necessary nor sufficient for what we discuss.

In response to this question, we make the following additions:

We have added a paragraph in the Discussion section on enzyme sequestration (Pages 16, lines 588-595). We write:

“Enzyme Sequestration. Enzyme sequestration (and competition) provides the key non-linearity for generating symmetry-breaking obviating the need for explicit feedback. Eliminating enzyme sequestration eliminates the possibility of symmetry-breaking.

Enzyme sequestration (and competition) is a key ingredient in multisite modification, and in general this could combine with zeroth order ultrasensitivity to generate new behaviour. However, the above behaviour does not require any explicit assumption on the kinetic regime of enzymatic action (as seen from the sufficient conditions we have obtained).”

We refer to a prior paper discussing explicit feedback with multisite modification (in a stochastic setting). In our discussion of Modularity in the Discussion section (Page 18, lines 704-710), we write:

“Interestingly an existing study Krishnamurthy et al.. (2007) examines sequential multisite modification with two explicit feedbacks: one from the maximally modified phosphoform increasing the probability of (every) modification and the other from the unmodified form increasing the probability of every de-modification. In a stochastic setting, this has been shown to result in breaking a symmetry between phosphorylation and dephosphorylation even with no enzyme sequestration. In contrast to this, all our studies are on the intrinsic behaviour of multisite modification, and in a deterministic setting.”

3) To what extent is symmetry breaking a necessary condition for the properties that it can induce, in particular absolute concentration robustness? and to what extent does symmetry need to exactly hold?

We first address the question: to what extent is symmetry-breaking necessary for absolute concentration robustness.

To do this we distinguish between two forms of absolute concentration robustness: exact and approximate. In the former the robustness is obtained (in a range of total concentration of substrate, say) exactly mathematically, with no assumptions on the regime of modification and no invocation of limiting regimes. In the second case, we see how absolute concentration robustness can emerge approximately.

Exact ACR: To address this question we focus on an ordered double site phosphorylation system. We address the questions:

1. Which substrates in this network are capable of exhibiting ACR?

2. Is ACR (where possible) only exhibited with respect to the (change of) total substrate concentration? Is ACR possible with changing total enzyme concentration?

3. Are there any additional constraints on the kinetic parameters required to observe ACR (where possible)?

4. What associated features (if any) are exhibited when ACR is observed in this network?

Our answers, obtained using a combination of analytical and computational work are as follows:

1. Our analysis reveals that ACR can only be exhibited by the partially modified substrate form (A_p_)

2. ACR in A_p_ is only possible with changing concentration of total substrate.3. Further analysis of the ACR in A_p_ reveals necessary (and sufficient) constraints on the kinetics and total concentrations of enzymes.k3PTotalc2(k2KTotal−k3PTotal)=k1KTotalc4(k4PTotal−k1KTotal)>04. Our analysis was able to further characterize associated features of this network when A_p_ exhibits ACR. In particular,

i. If this system exhibits ACR in A_p_ for a range of total substrate concentration, it is necessarily multistationary within that range with two steady states exhibiting ACR.

ii. There necessarily exists another steady state branch for all positive total substrate concentrations. This branch intersects one of the two ACR branches at some A_Total_ value (computationally found to be a transcritical bifurcation, refer Appendix 2 – Figure 9)

iii. This non-ACR branch state is characterized by a fixed ratio of free (unbound) Kinase to free (unbound) Phosphatase concentration for all positive total substrate concentrations, given byKP=KTotalPTotalThus, exact symmetry is not a requirement for (exact) ACR. In fact, a weaker condition can generate ACR (for A_p_). The symmetric case trivially satisfies this weaker condition. On the other hand, ACR of this type necessarily involves multiple steady state branches (with ACR).

Approximate ACR: We first note that deviations from the symmetric networks can result in approximate ACR (e.g., Appendix 2 – Figure 5, 6). It is easy to see that approximate ACR can also be achieved, by exploring different limiting regimes of enzymatic modification. We provide multiple explicit cases of this type:

In the common kinase common phosphatase case:

i) ACR in A_p_: Having phosphorylation of A being in the saturated regime, and phosphorylation of A_p_ in the unsaturated limit, while having phosphatase total amounts large relative to substrate. This ensures a zeroth order production of A_p_ balanced by a first order removal, providing the robustness

ii) ACR in A_pp_: A similar approach except that we require the phosphorylation of A_p_ to be saturated and that of A to be in the unsaturated limit.

iii) ACR in A: A similar approach to (ii) except for swapping the requirements/constraints on kinases and phosphatases

Note that such ACR does not require multiple steady states. Furthermore, it can be obtained even when the kinases and/or phosphatases are distinct

We also briefly discuss how the same approach can be used to obtain ACR for one species in a random modification network.

All in all, we can say that ACR can be obtained in different ways and does not require symmetry. However exact ACR of different types (with different combinations of variables or their sums), emerge “naturally” as a by-product of symmetry breaking.

In that sense, starting with a symmetric network (with some constraints to enable symmetry breaking), varying a single parameter can generate this, which is a noteworthy point is its own right: it does not require extensive local tinkering in the network. We also obtain ACR along with metastability, something not in the limiting regimes mentioned above.

Examining ACR mathematically in more detail and the requirements thereof in the various cases is well beyond the scope of the current work and is a separate investigation in its own right and will be studied subsequently.

Changes in the text:

i) We have added a section in the Appendix 1 discussing these issues in detail (Page 42, lines 1658) along with a new Figure, Appendix 2 – Figure 9, depicting presence of exact ACR in a non-symmetric ordered DSP with common kinase common phosphatase. A supporting Maple file (added in Source code file and Supplementary file 1) provides relevant information about the proofs (5.1).

ii) We have included two new paragraphs in the discussion on the origins of ACR (page 17).

We write (Page 17, line 658):

**“**The origins of Absolute Concentration Robustness. Based on the above, a natural question is which substrates could exhibit ACR and whether symmetry is a pre-requisite. We note that in the ACR we have made no assumption/restriction or invoked any particular kinetic regime for enzymatic action. We answer the questions (based on analytical work: see Appendix 1) relating to ACR in these terms, in the A ordered DSP network. (i) Only A_p_ can exhibit ACR, and this occurs only in response to A_Total_ (not K_Total_ or P_Total_). (ii) ACR necessarily requires multiple steady states, with two branches of steady states exhibiting ACR. There is another steady state branch which does not exhibit ACR but intersects one of the branches in what was computationally observed to be a transcritical bifurcation (iii) There is a constraint on parameters to enable this, which is weaker than the symmetry condition. (iv) In the case of symmetry, the two ACR branches are symmetric and intersect with the other branch in a pitchfork bifurcation.”

**“**Approximate ACR. We then investigate whether concentration robustness (approximate) could be obtained in specific kinetic parameter regimes. Here we find (i) A_pp_ and A could also exhibit concentration robustness. This can happen in a regime where the enzyme producing this from A_p_ acts in the saturated regime while the action of both enzymes on reactions not involving the species under consideration, act in the unsaturated limit (see Appendix 1). Here approximate ACR occurs without requiring multistability (ii) Similarly approximate ACR can occur in A_p_ without multistability by (for instance) having phosphorylation of A in the saturated regime, and phosphorylation of A_p_ and dephosphorylation of A_pp_ in the unsaturated limit. Similar insights can enable approximate ACR for one species in the corresponding random network: enzymatic modification leading to the creation of that species in the saturated limit (zeroth order) and all reactions not leading to production to or modification of the species under consideration acting in the unsaturated limit. Other analyses of exact and approximate ACR will be investigated in future work.”

Regarding exact vs Approximate Symmetry: We had briefly discussed this point in the first version. In the revised version we have included an additional supplementary figure (Appendix 2 – Figure 6) where we vary one parameter in a range (50% to 150% of the symmetric value) introducing an asymmetry. Our study indicates that the bifurcation structure is very similar to the case we had already showed (a perturbed pitchfork). Additionally, we also provide further evidence for an approximate ACR emerging in this case.

4) Make more explicit the evolutionary scenarios that are only briefly sketches: what are the assumptions on the symmetries that are 'naturally' expected to be present prior to any evolution?

We have made this more explicit. Regarding the assumptions of symmetry—if we assume a case 2 symmetry, we assume that the different phosphoforms behave essentially the same (and have similar dynamics). Symmetry breaking (or its analogue) could be effected by varying one simple to adjust parameter in the cell: expression level of substrate, which could enforce a significant degree of biasing of one pathway over another. Note that a given (asymmetric) steady state branch progressively accentuates the biasing. This means that in subsequent rounds of evolution, a cell by further increasing expression levels of a substrate could cause further biasing. This could also be reinforced by (rounds of) local tinkering, wherein further biasing could be introduced by decreasing/increasing rates of different reactions. The outcome of all of this would be a significant or even almost complete biasing of one pathway over the other.

We summarise this in the Discussion section with the addition of a new paragraph (Page 16, lines 622-631).

“Given a symmetric (Case 2 Symmetry) or close to symmetric network where different phosphoforms behave (essentially) the same, there are different ways in which evolution could lead to biasing of one modification pathway over the other. One is by effecting local changes in one of the pathways. Symmetry breaking allows for a distinct mechanism whereby changing one easy to manipulate parameter (expression level of substrate), a significant biasing of one pathway over the other is established. This could be further reinforced (if this is a desirable outcome) by local changes in the pathway or increasing substrate amounts further (which further accentuates the biasing). This can lead to either partial or even complete ordering subsequently. Thus, the mechanism could be seen as an efficient way of effecting a substantial change which could be reinforced and consolidated by further tinkering. It can also generate different robustness characteristics.”

5) Include mention of symmetries in other parameters, notably dissociation constants

This has been made clear in multiple places. We have explicitly mentioned that when we mention symmetries we are talking about all parameters: binding, dissociation and catalytic. This is emphasized in the main text (paragraph “Network Symmetry meets multisite modification” – Page 6, line 185), and caption of Figure 1 (Page 4).

This is reinforced in the Appendix 1 where the point about dissociation parameters is explicitly noted (as shown below – Page 32, lines 1186-1194).

“The symmetry in the context of our models is established through a strict kinetic structure and enforcement of total enzyme concentrations which ensures that certain pairs of kinetic terms are equal (Refer Figure 1). Note that this requires, in general, the binding, unbinding and catalytic constants to be the same. This ensures that starting with symmetric initial conditions for the appropriate variables (substrates and associated enzymes and complexes), the system evolves maintaining this symmetry. In this context we point out that all three constants could affect the modification rate, and in fact studies Hatakeyama and Kaneko (2014, 2020) show how even unbinding constants significantly affect efficacy of modification.”

Reviewer #1 (Recommendations for the authors):Post-translational modifications of proteins at multiple sites, for instance by phosphorylation, play a key role in signal transduction and processing. Their system-level behaviors, which include bistability and oscillations, can be analyzed on a case-by-case basis with mathematical tools from dynamical systems. The main contribution of the paper is to propose a new perspective that goes beyond this case-by-case approach. This is achieved by pointing out that protein networks based on multisite modifications can display symmetries and that these symmetries can be broken to give rise to properties of biological relevance: ordering, directionality, concentration robustness,…The paper includes on one side a mathematical analysis of two-site modifications systems, with an detailed analysis of the symmetries that they may display, the conditions for these symmetries to be present, and the conditions under which they may be broken. On the other side, the paper includes an extensive general discussion of the relevance of these analyses for actual biological system.This is not a standard paper presenting a specific result or a novel technical method but a thought-provoking proposal revisiting the dynamics of multisite modification systems from a new perspective. While it is difficult to judge it by the usual standards, it should be of broad interest. For instance, no reference is made to any particular experimental system but the authors convincingly argue for the relevance of their point of view for interpreting natural systems and designing synthetic ones.

We thank the reviewer for a careful reading of the manuscript and the comments. We are pleased to see that the reviewer finds the work thought-provoking and highlights the new perspective in this paper. We are also pleased to see that the reviewer notes that it should be of broad interest.

My main comment concerns the generality of the concept of symmetry: the paper advocates it in general but focuses on simple cases of reflection symmetry with 2-site modification systems. What other symmetries, if any, may be expected and of interest? How may the approach generalize beyond 2-site modification systems? Addressing exhaustively these questions may be outside the scope of a single article, but it would be worth fleshing out a few possible extensions.

Our paper has focussed on symmetry, keeping in mind not only what is possible but also where it might be natural in multisite modification. In the double-site modification, Case 2 Symmetry naturally follows from basic assumptions (associated with ordering), Case 1 is a basic symmetry based on directionality, and is also a constituent of Case 3 symmetry, which appears to be a focal point around which parameter sets enabling oscillations were found in a detailed parametric study.

When we proceed from 2 to a higher number of sites (in random modification networks), natural analogues of these symmetries exist. For instance, a symmetry in the direction of modification (analogue of Case 1) can be seen, as can the analogue of Case 2 symmetry (all phosphoforms of a certain number of modifications being the same). The analogue of Case 1 symmetry can be readily seen in ordered modification systems as well.

Additional possibilities arise due to the presence of multiple modification sites. For instance, it is possible that certain “legs” of the modification exhibit this symmetry but not others for instance, in a triple site modification, both pathways which follow from the one specific site of the three being modified first, behave identically. If the specific site being modified first is denoted by the first subscript, then we have A101=A110. But the symmetry/equality of partial phosphoforms may not extend to all modification “legs”.

Not surprisingly, increasing the number of modification sites further can create new behaviours as well. Based on our study of ordered Triple Site Phosphorylation (TSP) which was present in the original document, but only briefly referred to, we find that it is possible to have Case 1 Symmetry Breaking, with shared robustness, behaviour not encountered in the ordered Double Site Phosphorylation system (Appendix 2 – Figure 1B in the initial version, Appendix 2 – Figure 8 in the revised version).

Naturally when the number of sites increases, the number of partial phosphoforms increases exponentially. In addition, the number of possibilities significantly increases when one considers the different combinations of distinct and common enzymes which are possible (for instance that fact that there is a complete spectrum from all (de)modifications performed by one enzyme to every (de)modification performed by a distinct enzyme, to intermediate possibilities)

Analyzing this systematically needs a dedicated study of its own, for which the current study can serve as a platform. As part of this, a question to be addressed is whether there are symmetries which have no correspondence directly (fully or partially) or indirectly with the ones studied here (i.e., involving directionality or ordering), or combinations thereof. That being said, from the perspective of multisite modification, it appears that symmetries associated with directionality or ordering (or combinations thereof) appear to be the most natural ones to investigate.

A paragraph has been added in the Discussion section in relation to this (Page 19, lines 752-767). We write:

“It is worth examining implications and extensions of our study to a larger number of modification sites. Random networks lead to an exponential increase in the number of states, but in addition the modifications/demodifications can be effected by common enzymes (for all modifications), distinct enzymes (for every modification) or a combination thereof, leading to a combinatorial explosion in possibilities. Clearly direct analogues of the symmetry breaking seen here (for eg. Case 1 and Case 2) can be encountered here. In addition, new possibilities can emerge. In Case 2, for instance, in addition to the situation where all modification legs behave the same, we can have a situation where some modification legs are the same. Furthermore, not surprisingly, new behavioural characteristics can emerge. For instance in the ordered Triple Site Phosphorylation network (Case 1 Symmetry: Common Kinase Common Phosphatase), we find shared robustness (see Appendix 2 – Figure 8) as well as oscillations, not seen in the ordered double site modification. These aspects need a dedicated study of their own and will be studied in the future.”

– Figure 1D is key to define what is meant by symmetry but in my opinion not sufficiently informative. In particular the symmetry in kinetic structure is highlighted while the constraints on enzymes, which seem as essential, are not represented. For instance requiring k1=k3 is not sufficient, something like k1[K1]=k3[P2] is in fact needed? Also, as represented, Case 1 and Case 2 appear as formally equivalent (related through a 45 degree rotation). I was also wondering if more transparent notations could be used for ki, aj, which would reflect the (possible) symmetries?

Firstly, for the notation, we have modified the notation denoting the triplet of rate constants (binding/unbinding/catalytic), using k and a, to avoid confusion (See all figure schematics).

The point about the symmetries in the networks, and the importance of the enzymes (which really determine the distinction between Case 1 and Case 2 Symmetries) are discussed in Essential Revisions (response to Q1).

– Another example of information processing at the molecular level where symmetry plays a key role is the MWC model of allostery. It may be worth mentioning it as, beyond the analogy, post-translational modifications can be associated with allosteric transitions.

A sentence (along with associated reference) has been added to the Introduction (Page 2, line 71):

“Symmetry has also been invoked as a key ingredient in the development of the MWC model which has been used to explain allostery in biomolecular information processing Changeux (2012)”

– The two concepts of information processing and symmetry breaking bring to mind the possibility of encoding a bit of information in the symmetric states: could it be relevant in the context of post-translational modifications?

A new sentence has been added to Discussion section emphasizing this (Page 19, line 763).

“Viewed from the perspective of information storage, symmetry breaking suggests that a symmetric double-site modification network contains a bit of information. We emphasize however that the symmetric network encodes a richer set of information, such as simultaneously presenting homeostasis and multiple steady states”

Reviewer #2 (Recommendations for the authors):In this paper, the authors demonstrated that the symmetry in the concentration of different phospho-forms of a protein could be broken in multisite phosphorylation processes in proteins, even if the reaction's kinetic parameters to produce those are the same. They analyzed some different classes of symmetries in the kinetics for a protein with two phosphorylation sites and showed some of those shows the pitchfork bifurcation or the Hopf bifurcation. Also, they discussed that the relevance of the symmetry breaking for the absolute concentration robustness.The analysis of classes of networks with different symmetries may be of use for future research in the field. Also, the relationship of the symmetry breaking to the absolute concentration robustness may be interesting. However, the discussion of whether symmetry breaking is a necessary condition for the absolute concentration robustness is still lacking. Moreover, the connection to previous studies has not been argued enough. For example, the symmetry-breaking mechanism may be related to the Goldbeter-Koshland-type zero-order ultrasensitivity and the enzyme competition. However, the connection has not been discussed sufficiently. Also, the symmetry breaking in the multisite modification has been studied for networks with feedback. However, mention of those studies is still lacking.

We thank the reviewer for a careful reading of the manuscript and for the comments

This paper still has several problems to be addressed, as described below.1) The writing often lacks clarity and sharpness and is poorly organized. Although the authors showed their analyses for different symmetries in order from Case 1 to 3, the central mechanism of the symmetry breaking may be almost the same in these three classes. Moreover, the ordered distributive DSB seems sufficient to demonstrate the symmetry breaking by such a central mechanism. Hence, I recommend reorganizing the paper; to state detailed analyses for the ordered distributive DSB and an intuitive explanation at first, and then discuss three classes in the following sections.

We first make a point about the presentation. We have aimed to write for a broad audience and as a consequence have included an extra degree of summarizing in the conclusions for readers who may not be interested in reading through all the details, so that the essential points can be accessed in a self-contained way. This introduces a reduction in sharpness on account of this.

Regarding clarity (and the subsequent comment of the reviewer) we have made much more explicit the differences between the two cases of symmetries (discussed further below).

Now regarding the organization of the paper, we have organized the paper so that:

i. We discuss Case1, followed by Case 2, followed by Case 3.

ii. In each case we consider all combinations of common/separate kinases/phosphatases, where relevant.

iii. Case 1 symmetry is possible in an ordered mechanism as well. As a consequence, in the section on Case 1 symmetry, we present the ordered mechanism first, as it is simpler to understand before proceeding to the random mechanism. In fact we explicitly leverage our understanding of the ordered mechanism in analyzing the random mechanism.

That being said, we do not believe that the ordered mechanism (Case 1 symmetry breaking) represents the common underlying mechanism for all cases in the paper (other than the fact that it involves symmetry breaking via a pitchfork bifurcation). In fact, there is a difference in the Case 1 and Case 2 symmetry breaking, with further differences in implications; additionally, Case 2 as defined is not relevant in an ordered mechanism.

Finally, we point out that in the presentation of results, since Case 1 comes first, the ordered mechanism is the first non-trivial case to be presented anyway. We have also added a paragraph to explicitly clarify the organization of the paper.

We write (Page 7, line 259):

“We present the results for Case 1, Case 2 and Case 3 for the different random modification networks below. The ordered double-site modification network can exhibit Case 1 symmetry, as noted above. Therefore, in presenting the Case 1 symmetry, we start with this simpler network, before proceeding to the Random Modification Networks”

2) The authors stated that Case 1 symmetry breaking was observed "only in the common kinase, common phosphatase case," and Case 2 symmetry breaking "is broken only for different kinase and different phosphatase case." It looks strange to me. Changes of variables, e.g., from A_00 to A_01, A_11 to A_10, k_1 to k_2, k_3 to a_2, and so on, can transform one case to the other. Thus, the two cases seem to be identical. Please explain why those two cases are different.

We believe that the reviewer was referring to the third sentence in the paragraph “Which Symmetries can be broken” in the Discussion section “Case 1 Symmetry Breaking can be observed in a simple ordered DSP network (with only a single partial phosphoform), though only in the common kinase common phosphatase case” (in the initial version).

This appears to be a misunderstanding: we had stated that for ordered mechanisms, Case 1 symmetry breaking was observed only in the common kinase, common phosphatase case. The separate kinase separate phosphatase case of ordered double site modification does not possess sufficient non-linearity to enable multiple steady states, much less symmetry-breaking. Clearly if we consider random mechanisms, it can be seen in the separate kinase separate phosphatase case—and we have discussed it ourselves!

To avoid any potential confusion, the first line of the paragraph has been modified with the word “random” inserted (Page 15, line 546): it now reads

“The Case 1 Symmetry can be broken in all random modification networks where it exists”

As a broader point the Case 1 and Case 2 symmetries are different. This is discussed in the response to essential revisions Q1.

3) The mechanism of the symmetry breaking may be a combination of the Goldbeter-Koshland-type zero-order ultrasensitivity and a competition for the enzyme among different phospho-forms of the protein. However, there is no discussion about the relationship of symmetry breaking they found to zero-order ultrasensitivity and phenomena related to the enzyme competition.

We have studied symmetry-breaking in the different cases, and have found necessary conditions for the symmetry breaking, and shown further than this is in fact sufficient. In obtaining this, we have made no a priori assumptions on enzymatic regime.

We note two points: a degree of enzyme sequestration and enzyme competition is necessary for the requisite non-linearity—something seen across the studies in multisite modification.

With regard to zeroth order ultra-sensitivity, while this could contribute to the behaviour, this is not in any way a pre-requisite. Firstly, in examining our computational results, we find that there are instances which do not correspond to zeroth order ultrasensitivity. Secondly in examining the analytical requirements for symmetry breaking, the conditions do no enforce anything as specific as zeroth order ultrasensitivity. Thus, zeroth order ultrasensitivity is neither necessary nor sufficient.

We now have a paragraph in the discussion devoted in enzyme sequestration summarizing these points (Page 16, lines 588-595).

“Enzyme Sequestration. Enzyme sequestration (and competition) provides the key non-linearity for generating symmetry-breaking obviating the need for explicit feedback. Eliminating enzyme sequestration eliminates the possibility of symmetry-breaking.

Enzyme sequestration (and competition) is a key ingredient in multisite modification, and in general this could combine with zeroth order ultrasensitivity to generate new behaviour. However, the above behaviour does not require any explicit assumption on the kinetic regime of enzymatic action (as seen from the sufficient conditions we have obtained).”

4) The authors claimed the importance of the symmetry among phosphorylation and dephosphorylation speeds. However, even if the phosphorylation and dephosphorylation speeds are the same, the difference in dissociation constants can drastically change the actual reaction speed when substrates with different modification states compete for the limited amount of enzyme. (For example, Hatakeyama and Kaneko, PLoS Comput. Biol. (2014) and Hatakeyama Kaneko, Phys. Rev. Research (2020)). Hence, I recommend that the authors emphasize the importance of the symmetry in the dissociation constant as well as the modification speed, to make a reader pay attention.

When we impose symmetry, we are enforcing equality of the corresponding binding, dissociation and catalytic constants. This had been mentioned in the previous version and has been made more clear in the models section. Furthermore, the importance of all these constants in this regard is reinforced in the Appendix 1 (while discussing symmetry, Page 31, lines 1171-1176) and the relevant references are cited in the context of the dissociation constants.

We write (Page 32, lines 1186-1194):

“The symmetry in the context of our models is established through a strict kinetic structure and enforcement of total enzyme concentrations which ensures that certain pairs of kinetic terms are equal (Refer Figure 1). Note that this requires, in general, the binding, unbinding and catalytic constants to be the same. This ensures that starting with symmetric initial conditions for the appropriate variables (substrates and associated enzymes and complexes), the system evolves maintaining this symmetry. In this context we point out that all three constants could affect the modification rate, and in fact studies Hatakeyama and Kaneko (2014, 2020) show how even unbinding constants significantly affect efficacy of modification.”

5) The author demonstrated that if the network is symmetric and the symmetry in concentration is broken, the concentration of some phospho-form of the protein shows the absolute robustness. However, they did not show whether symmetry breaking is necessary for the absolute concentration robustness. Does a network without symmetry never show the absolute concentration robustness?

We had never claimed symmetry breaking was essential for concentration robustness. We have outlined additional analysis performed to address this question: this is presented in response to Essential Revisions Q 3.

6) The authors cited papers only written by Ueda's group (Ode and Ueda, (2018), Jolly et al., (2012), Sugui et al., (2017), and Shinohara et al., (2017)) about the circadian clock as references. However, the circadian clock generated by the multisite modification has long been studied in cyanobacteria. Thus, the basic concept has already been stated in previous studies. For example, a generation of the oscillation by a protein with two phosphorylation sites was published in Rust et al., "Ordered phosphorylation governs oscillation of a three-protein circadian clock." (2007). The importance of enzyme sequestration for the oscillation was published in van Zon et al., "An allosteric model of circadian KaiC phosphorylation." (2007). Also, the mechanism of temperature compensation by the competition for the enzyme was published in Hatakeyama and Kaneko, "Generic temperature compensation of biological clocks by autonomous regulation of catalyst concentration." (2012). I recommend citing those papers in addition to Ueda's paper, for the healthy development of the scientific community.

Thank you for suggesting these references. We had initially referenced what we thought were the most relevant references but agree that all these references are well worth including and represent a wider spread of investigations and groups. These have been included in the Discussion section in the paragraph on oscillations (Page 18, line 712).

7) The symmetry breaking in the multisite modification has been studied, at least for networks with explicit feedback. Hence, it may be better to tone down the title and some claims.

We have included a reference which involves multisite modification and explicit feedback, which includes symmetry breaking. We also discuss where our study is essentially different: we focus on the intrinsic dynamics of multisite modification without extrinsic feedback, and we perform studies in a deterministic setting for a broad range of basic multisite modification circuits (Page 18, line 704).

“Interestingly an existing study Krishnamurthy et al. (2007) examines sequential multisite modification with two explicit feedbacks: one from the maximally modified phosphoform increasing the probability of (every) modification and the other from the unmodified form increasing the probability of every de-modification. In a stochastic setting, this has been shown to result in breaking a symmetry between phosphorylation and dephosphorylation even with no enzyme sequestration. In contrast to this, all our studies are on the intrinsic behaviour of multisite modification, and in a deterministic setting.”

We did carefully consider the comment about toning down the title. Our title “Symmetry breaking meets multisite modification” simply reflects the fact that we investigate symmetry breaking across a broad suite of basic modification networks and focus on their symmetries and intrinsic capacity for symmetry breaking. In that sense we believe the title is justified. It is also crisper than something like “The interplay of symmetry-breaking and multisite modification” while conveying the same thing. Finally, we have not exaggerated any claims—all the technical claims are based on and supported by mathematical and computational analysis (a point referred to by Reviewer 4 as well).

Reviewer #4 (Recommendations for the authors):In this manuscript, Ramesh and Krishnan analyze on a theoretical level symmetry breaking in multisite models of covalent modification. The basic idea is that while the underlying network dynamics may preserve certain symmetries, these symmetries can be broken leading to asymmetric states, with for example unequal levels of doubly modified or unmodified phosphoforms. The symmetry breaking requires the presence of nonlinearities which fundamentally result from sequestration effects of the necessary enzymes/substrates. Overall, I found the paper to be a highly original contribution, with some very nice results that could open up new directions, for example in understanding the evolution of asymmetric networks. Their extensive analytical and numerical work clearly supports their conclusions, and suggestions for experimental signatures of broken symmetry are given. This work may have consequences for the fields of network evolution and synthetic biology.

We thank the reviewer for a careful reading of the manuscript: we are pleased to see the appreciative comments.

In general, I found the manuscript to be well-written. Nevertheless, the paper is definitely not an easy read, despite clearly significant efforts from the authors due to the complexity of the subject matter.One aspect that needs further attention is to what extent the system needs to be exactly symmetric to exhibit the behavior studied by the authors. This is an important point as some of the required symmetries are not natural: for example, in Case 1 of Figure 1D, k_1=k_3 which means that a phosphorylation and a dephosphorylation reaction must be precisely balanced. There is some effort towards analyzing this in Figure S5, but I think this aspect needs to be given much greater prominence and discussion also in the main text.

We thank the reviewer for recognizing the efforts in presenting the complexity of the matter for this audience. We have made a few changes in this version to help with the clarity: (i) A clearer delineation of the different symmetries, both in the main text and Figure 1(d): this is discussed in response to Essential revisions: point 1 (ii) We have included a paragraph in the results stating the organization of the results. We also have further discussion paragraphs in the conclusions, which help consolidate some of the themes better (e.g., “Enzyme Sequestration”, “The origins of Absolute Concentration Robustness” or a discussion of a greater number of modification sites).

Our discussion of symmetries and in fact our distinction between Case 1 and Case 2, makes the significance of the enzymes very explicit.

Regarding the inexact symmetries, we perform further analyses to show that echoes of all the principal features seen in the symmetric case are seen in cases where the system is not too far from symmetry. We expand on Appendix 2 – Figure 5 in the initial version (with an additional figure – Appendix 2 – Figure 5 and 6 in the revised version) and explore a range of parameters values of a given parameter (representing deviations from symmetric). As can be seen from this figure (and the metric we have used to analyze the results), echoes of the results seen in the symmetric case can be seen even when there is a clear deviation from the symmetric case.

While some symmetries appear more natural than others, consideration of them together provides a broader conceptual synthesis. We also point out that Case 1 symmetry is a subset of Case 3 symmetry which is scenario around which parameters enabling oscillations are clustered in computational parameter explorations in Jolly et al., 2012.

The authors also argue that evolution may have started with a symmetric network with broken symmetry as a starter towards asymmetric networks. This is an interesting idea but isn't it more likely that the networks were just asymmetric to begin with (which has a much bigger parameter space), and that was subsequently exaggerated, rather than starting from an a priori fine-tuned symmetric configuration? A similar issue arises with claims that an observed asymmetric network may not have biased dynamics but may instead simply be due to symmetry breaking within a symmetric network. Without more knowledge surely the former is still much more likely?

We note at the outset that symmetry breaking is not a pre-requisite for asymmetric networks—even if the networks were approximately symmetric to start with. Of course, certain networks could have started being significantly asymmetric. In considering evolutionary changes, evolutionary “tinkering” through local changes could have led to the biasing of one pathway over another, progressively. In that sense there could be many parameters which have been manipulated by a cell (though it is plausible that similar parameters/factors may have been tweaked in consecutive iterations).

We point out that (i) Symmetry breaking is a distinct mechanism which can cause significant biasing (ii) this could be effected by varying a simple to change factor—the expression level of a protein (iii) it could be progressively manipulated, and then other factors could have been tinkering with leading to significant/total ordering (for instance) (iv) Furthermore even when the system is reasonably close to the symmetric case, echoes of this are seen. (v) Certain symmetries—either exact or approximate are plausible presences in these networks.

In that sense this is a distinct mechanism with rich consequences, with relevance to both systems and synthetic biology.

Naturally there are other ways to, for instance, obtain ordered networks (for instance through manipulations at the molecular structure level) which could well be relevant in the existence/realization of asymmetric networks.

We had added a paragraph in the Discussion section (Page 16, lines 622-631):

“Given a symmetric (Case 2 Symmetry) or close to symmetric network where different phosphoforms behave (essentially) the same, there are different ways in which evolution could lead to biasing of one modification pathway over the other. One is by effecting local changes in one of the pathways. Symmetry breaking allows for a distinct mechanism whereby changing one easy to manipulate parameter (expression level of substrate), a significant biasing of one pathway over the other is established. This could be further reinforced (if this is a desirable outcome) by local changes in the pathway or increasing substrate amounts further (which further accentuates the biasing). This can lead to either partial or even complete ordering subsequently. Thus the mechanism could be seen as an efficient way of effecting a substantial change which could be reinforced and consolidated by further tinkering. It can also generate different robustness characteristics.”

Key to the symmetry breaking is the nonlinearity introduced by sequestration effects without the need for explicit feedback. However, I think this needs an expanded discussion (probably with some equations) in the main text rather than being relegated to the (extremely large) methods. This is a central point, I think.

We agree that this could have a clear presence in the main text. However, we felt that adding equations in the main text would disrupt the flow of the text, and so have retained those in Appendix 1. Enzyme sequestration is discussed in detail in the Discussion section with the following text addition (Page 16, lines 588-595).

“Enzyme Sequestration. Enzyme sequestration (and competition) provides the key non-linearity for generating symmetry-breaking obviating the need for explicit feedback. Eliminating enzyme sequestration eliminates the possibility of symmetry-breaking.

Enzyme sequestration (and competition) is a key ingredient in multisite modification, and in general this could combine with zeroth order ultrasensitivity to generate new behaviour. However, the above behaviour does not require any explicit assumption on the kinetic regime of enzymatic action (as seen from the sufficient conditions we have obtained).”

I didn't find the left side of Figure 1D very helpful for my understanding. Also, the explanation of the "square" network topology wasn't made very clear; I didn't find it more straightforward than the depictions in Figure 1A-C, for example.

The figure 1 (d) has been reworked and the caption is also made clearer. The importance of the enzymes in the symmetries is also made much more explicit in this revision. This has been discussed in detail in response to Essential Revisions Q1.